# Enhanced ice sheet melting driven by volcanic eruptions during the last deglaciation

Francesco Muschitiello[1,2,3], Francesco S.R. Pausata[4,5], James M. Lea [6], Douglas W.F. Mair[6] & Barbara Wohlfarth[3]

Volcanic eruptions can impact the mass balance of ice sheets through changes in climate and the radiative properties of the ice. Yet, empirical evidence highlighting the sensitivity of ancient ice sheets to volcanism is scarce. Here we present an exceptionally well-dated annual glacial varve chronology recording the melting history of the Fennoscandian Ice Sheet at the end of the last deglaciation (∼13,200–12,000 years ago). Our data indicate that abrupt ice melting events coincide with volcanogenic aerosol emissions recorded in Greenland ice cores. We suggest that enhanced ice sheet runoff is primarily associated with albedo effects due to deposition of ash sourced from high-latitude volcanic eruptions. Climate and snowpack mass-balance simulations show evidence for enhanced ice sheet runoff under volcanically forced conditions despite atmospheric cooling. The sensitivity of past ice sheets to volcanic ashfall highlights the need for an accurate coupling between atmosphere and ice sheet components in climate models.

[1] Lamont-Doherty Earth Observatory, Columbia University, Palisades, NY 10964, USA. [2] Uni Research Climate, Nygårdsgaten 112, 5008 Bergen, Norway. [3] Department of Geological Sciences and Bolin Centre for Climate Research, Stockholm University, SE106-91 Stockholm, Sweden. [4] Department of Earth and Atmospheric Sciences, University of Quebec in Montreal, Montreal, QC, Canada H3C 3P8. [5] Department of Meteorology and Bolin Centre for Climate Research, Stockholm University, SE106-91 Stockholm, Sweden. [6] Department of Geography and Planning, School of Environmental Sciences, University of Liverpool, Liverpool, Merseyside L69 72T, UK. Correspondence and requests for materials should be addressed to F.M. (email: fmuschit@ldeo.columbia.edu)

A better understanding of the direct and indirect effects of volcanogenic aerosols on ice sheets is critical in the context of future meltwater contributions to global sea-level rise and ocean circulation. For instance, deposition of aerosol particles on snow and ice can affect the surface energy balance by lowering the snow albedo, thereby accelerating meltwater runoff[1–4]. Moreover, volcanic eruptions can influence the mass balance of ice sheets and glaciers through changes in precipitation and surface ocean and air temperature[5–10].

Although the response of present-day ice sheets to volcanism can be well characterized through observations and modeling of their surface mass and energy balance, little is known about the sensitivity of ancient ice sheets to volcanism. Specifically, empirical evidence that directly highlights the response of ice-sheet melting to external forcing is still lacking.

Here we achieve this using a new precise 1257-year long chronology from a continuous sequence of annual glacial varves, which record changes in the melting rate of the Fennoscandian Ice Sheet (FIS) during the period ~13,200–12,000 years BP. Precise synchronization to the Greenland ice-core chronology allows for the first time comparison to ice-core volcanic records at an unprecedented precision and suggests a causal relationship between ice sheet melt events and volcanism.

## Results

**Varve chronology and melt events.** Glacial clay-varves are one of the very few archives that can both provide continuous chronologies and have the ability to resolve climatic information at annual or even sub-annual time scales. Our new varve chronology spans the period ~13,200–12,000 years BP and is based on statistically validated cross-matching of 57 overlapping glacial clay-varve sequences investigated in south-eastern Sweden and close to the former highest shoreline of the Baltic Ice Lake[11] in the provinces of Småland and Östergötland[12,13] (Fig. 1). The glacial varves consist of distinct summer and winter couplets that were formed during the seasonal accumulation of ice-distal sediment. Ice sheet runoff occurring during the summer season routed large amounts of subglacial, sediment-rich meltwater into the Baltic Ice Lake, which resulted in the deposition of silt to fine sand layers. The corresponding clay layer formed in winter when lake ice cover facilitated the deposition of suspended sediment material. One varve year is therefore composed of a silt and clay couplet and records the melting and non-melting season, respectively. Glacial varve thickness hence, provides a proxy that captures the first-order pattern of melting and subglacial sediment flux of the local FIS margin[14].

The glacial varve chronology is here synchronized to the Greenland Ice-Core Chronology 2005 (GICCO5)[15]—hereafter expressed as years before 1950 AD (BP)—using the Vedde Ash isochron[13] (12,121 ± 57 years BP; 1σ). The cumulative mismatch between the varve and the GICC05 time scales from the Vedde Ash to the end of Greenland Interstadial 1 (GI-1)/onset of Greenland Stadial 1 (GS-1) (726 GICC05 years) does not exceed 0.15%, with a difference of only 1 year[13]. In turn, this allows a confident annual-scale comparison of the two time scales (Methods section).

To ascertain anomalous ice-melt events, we focused on the varve thicknesses, identifying exceptionally thick glacial varves (ETV) (Methods section). The focus of our analysis leans on the older portion of the chronology (~13,200–12,300 years BP), which is composed of 56 (out of the total 57) overlapping varve diagrams[13], here presented as a unified mean varve thickness record (Fig. 2; Supplementary Data 1). The composite record allows minimizing the melting signal from random variability noise embedded in the individual diagrams. The older portion of

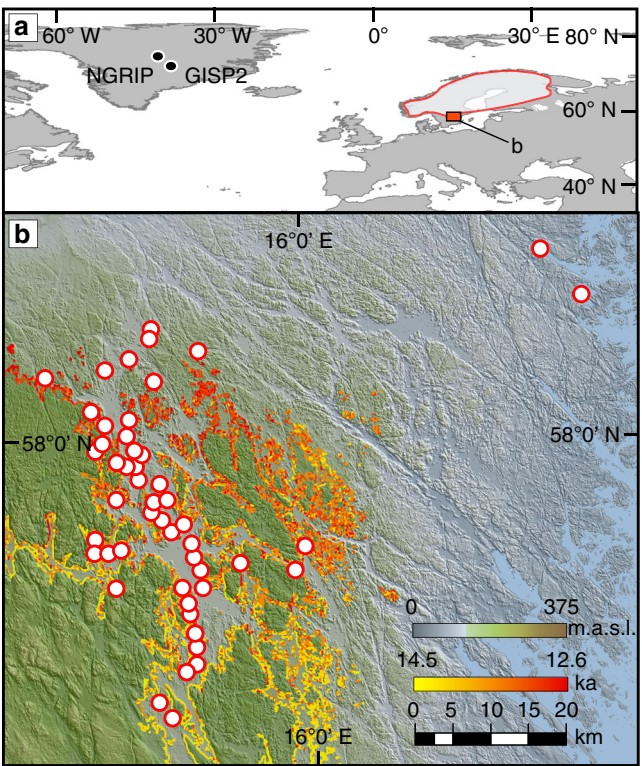

**Fig. 1** Location of the Swedish varve and Greenland ice-core records. **a** Study area with Younger Dryas ice-marginal line (red contour)[13,69] and the location of GISP2 and NGRIP ice cores in Greenland. **b** LiDAR-based topography showing the location of the sites used to construct the clay-varve chronology presented in this study (red circles). The highest shoreline of the Baltic Ice Lake is also displayed. Coastline data from the Geological Survey of Sweden colored according to present-day elevation (highest lake position was time-transgressive: yellow to red)

the chronology was therefore preferred to the younger, which is based on only one varve diagram[13].

**Ice-core records.** In this study we use the volcanic $SO_4^{2-}$ time series from the GISP2 ice core[16,17] (Methods section), which records past explosive and sulfur-rich volcanic activity (Fig. 2). The GISP2 record is synchronized to the GICC05 time scale via common volcanic markers[15,18,19] and exhibits a sampling resolution of 3–6 years per sample around the end of GI-1 and during the early part of GS-1. It should therefore be remembered that the frequency and magnitude of volcanic eruptions in this record is both under-represented and smoothed. Nonetheless, the record constitutes a valuable reconstruction of large-scale volcanogenic sulfate input to the Northern Hemisphere atmosphere, and is hitherto the only reliable record of this kind for Greenland.

Volcanic $SO_2$ is emitted into the troposphere and stratosphere, and progressively oxidized to $H_2SO_4$. Sulfates then precipitate on the Earth's surface via dry and wet deposition, and stratospheric sulfate aerosols are generally preserved in the ice-core stratigraphy in the form of sulfate and acidity peaks.

Electrical conductivity measurement data, which reflect the acidity of the ice, are also available for NGRIP2 ice cores[20] (Fig. 2) and have much higher resolution than the GISP2 record. However, these data can only be used as a complement to the GISP2 volcanic sulfate profile since high background alkaline dust levels during glacial conditions can suppress the acidity signal associated with potential volcanic eruptions[21].

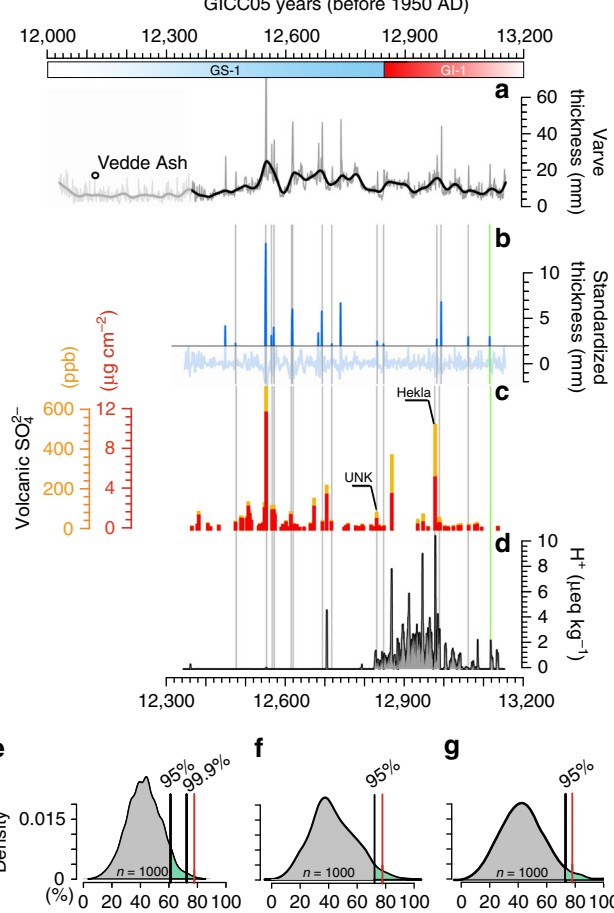

**Fig. 2** Comparison between the glacial clay-varve chronology and Greenland ice-core volcanic records. **a** Unified varve thickness diagram synchronized to the GICC05 time scale[13] and presented with a 10-year running mean smoothing line (black). The 'b2k' convention of the GICC05 time scale is here converted into BP (1950 years AD). The transition from Greenland Interstadial 1 (GI-1) to Greenland Stadial 1 (GS-1) is also displayed. **b** Varve thickness standardized anomalies of the portion of the varve chronology composed by 56 (out of 57) overlapping varve diagrams (see text for details). **c** Volcanic $SO_4^{2-}$ signal recorded in GISP2 expressed as absolute values (orange) and as flux (red). **d** Hydrogen ion measurements from the NGRIP2 ice-core reflecting the acidity of the ice[20]. This record is here used for reference as acidity peak heights can vary significantly between cores owing to differences in transport, deposition, and variations in background amount of alkaline dust[70]. Moreover, high dust levels during glacial/stadial conditions can make the ice alkaline, thereby suppressing the acidity signal[21]. Gray bars indicate exceptionally thick varve years coherent with anomalies in atmospheric volcanic sulfate. The green bar shows an additional match between an exceptionally thick varve year and an acidity peak in NGRIP2 records that has no counterpart in the GISP2 sulfate record. The thickness of the bars has been increased to improve readability. Tephra horizons identified within Greenland ice cores that correspond to volcanic sulfate peaks in GISP2 records are also labeled (Supplementary Table 2; UNK, unknown volcano). **e**–**g** Results from Monte Carlo significance tests of synchronicity between exceptionally thick varve years and volcanic eruptions. In **e**, synchronicity is tested using 1000 permutations of the varve thickness anomalies. In **f**, synchronicity is tested using 1000 individual realizations of the varve thickness record with similar red noise spectral characteristics. In **g**, synchronicity is tested similarly to **f** but using Gaussian white noise. The green area indicates the region above the 95% confidence level and the red line indicates the estimated coherency (%) between varve anomalies and volcanic eruptions (78%)

To ascertain tropical vs. high-latitude sources of volcanogenic sulfate injections in the GISP2 records, we also compare Greenlandic and Antarctic ice-core records of volcanism to identify tropical eruptions via the occurrence of volcanic isochrones in both hemispheres (Supplementary Fig. 1). The comparison shows a paucity of tropical eruptions over the period under investigation and generally a higher frequency of volcanic eruptions recorded in Greenland relative to Antarctica, suggesting a predominant Northern Hemisphere high-latitude source for these events.

**Data interpretation**. We identified 18 ETVs over the period 13,200–12,300 years BP and estimated that ∼80% of these events are synchronous with volcanic sulfate anomalies (Fig. 2; Supplementary Table 1). Three independent Monte Carlo tests (Methods section) indicate that the coherency is very unlikely to occur by chance ($p < 0.01$, $p < 0.05$, and $p < 0.05$, respectively; Fig. 2). We observe that the number of annual layers between the identified isochrones is consistent with the respective cumulative counting errors and that the ETVs fall well within the sampling resolution uncertainties associated with the GISP2 record (Supplementary Fig. 2). Furthermore, at least two isochrones correspond with the deposition of Icelandic tephra in NGRIP ice cores[22], which are associated with a large and a medium-sized sulfate peak, respectively, in GISP2 records (Fig. 2; Supplementary Table 2). Two consecutive and major ETVs occur within the span of a large excursion in the sulfate record (12,551 years BP) (Fig. 2). This could be due to the sampling resolution in the ice record, which under-represents the frequency of volcanic eruptions.

Volcanogenic sulfate records capture a signal that depends on a number of unknown factors. For instance, deposition of volcanic products in ice cores is influenced by: the distance from the source region of the volcanic eruption; the height of the volcanic plume, which influences the aerosol residence time in the atmosphere; the precipitation transport pathways and the season at which the eruption takes place; contribution of wet vs. dry deposition[23–25]; the amount of washout on the summit[26]; and the aerosol particle size and deposition efficiency[27].

Analogously, the composite varve thickness record captures a compounded ice-melt signal which integrates altogether: distance of the sampling site from the ice margin; meltwater pathways and sediment entrainment; and transport and deposition both subglacially and proglacially.

Climate metrics—and so the ice-sheet response to external perturbations—do not scale in a simple, linear fashion with volcanic aerosol forcing[10,28]. Therefore, due to the limitations associated with the proxy reconstructions and non-linear FIS response in terms of ice dynamics and hydrology, we do not expect a simple linear relationship between volcanic sulfate concentrations and annual varve thickness in our sedimentary archive. Nonetheless, the observed correspondence between volcanic sulfate anomalies and enhanced FIS meltwater runoff, corroborated by the significance tests, supports the hypotheses of a direct impact of volcanic eruptions on ice-sheet melting.

**Climate and runoff model simulations**. To explore the potential climate feedbacks on the ice sheet induced by volcanic eruptions, we turn to a set of climate simulations performed with two climate models to account for both the volcanic $SO_2$ and changes in boundary conditions (e.g., solar insolation) relative to present day (Methods section). The results are used to drive a field-validated physically based energy balance model incorporating snowpack/ice mass balance and to test the sensitivity of ice sheet runoff to ash deposition (Methods section). We simulate one of the largest

**Table 1 Modeled change in runoff in response to a summer high-latitude eruption**

| Altitude (m) | Runoff (cm w.e.)<br>$\alpha = 1$ ($\alpha_i$, $\alpha_s$) | Runoff (cm w.e.)<br>$\alpha = 0.95$ ($\alpha_i$, $\alpha_s$) | Runoff (cm w.e.)<br>$\alpha = 0.9$ ($\alpha_i$, $\alpha_s$) | Runoff (cm w.e.)<br>$\alpha = 0.85$ ($\alpha_i$, $\alpha_s$) |
|---|---|---|---|---|
| 0 | $-128.68 \pm 3.68$ | $-117.13 \pm 3.85$ | $-106.33 \pm 4.08$ | $-94.48 \pm 3.80$ |
| 500 | $-106.93 \pm 1.90$ | $-95.59 \pm 2.21$ | $-83.95 \pm 3.13$ | $-72.92 \pm 3.53$ |
| 1000 | $-81.18 \pm 1.36$ | $-72.15 \pm 1.25$ | $-63.08 \pm 1.64$ | $-52.44 \pm 3.29$ |
| 1500 | $-57.27 \pm 12.18$ | $-53.91 \pm 9.80$ | $-48.00 \pm 6.67$ | $-40.84 \pm 4.33$ |

Summary statistics of volcanically forced change in annual runoff model results (given in cm water equivalent, w.e.) and related standard deviations for a summer high-latitude volcanic eruption. The alpha value of the albedo refers to albedo of both snow and ice. The full simulation results are shown in Supplementary Fig. 3

**Table 2 Modeled change in runoff in response to a summer high-latitude eruption with unchanged SWRF**

| Altitude (m) | Runoff (cm w.e.)<br>$\alpha = 1$ ($\alpha_i$, $\alpha_s$) | Runoff (cm w.e.)<br>$\alpha = 0.95$ ($\alpha_i$, $\alpha_s$) | Runoff (cm w.e.)<br>$\alpha = 0.9$ ($\alpha_i$, $\alpha_s$) | Runoff (cm w.e.)<br>$\alpha = 0.85$ ($\alpha_i$, $\alpha_s$) |
|---|---|---|---|---|
| 0 | $-67.46 \pm 3.84$ | $-48.54 \pm 4.16$ | $-30.14 \pm 3.88$ | $-11.87 \pm 4.39$ |
| 500 | $-49.60 \pm 1.85$ | $-32.58 \pm 2.16$ | $-14.26 \pm 3.50$ | $4.03 \pm 3.79$ |
| 1000 | $-34.91 \pm 0.97$ | $-17.84 \pm 1.51$ | $-0.79 \pm 2.85$ | $16.12 \pm 3.63$ |
| 1500 | $-28.27 \pm 1.11$ | $-12.80 \pm 0.58$ | $2.82 \pm 1.45$ | $20.99 \pm 4.16$ |

Summary statistics of volcanically forced change in annual runoff model results (given in cm water equivalent, w.e.) and related standard deviations for a summer high-latitude volcanic eruption with SWRF left as if non-volcanically forced (i.e., a large eruption where there is insufficient sulfur emitted to alter SWRF). The alpha value of the albedo refers to albedo of both snow and ice. The full simulation results are shown in Supplementary Fig. 4

high-latitude volcanic eruptions in historical time—the Laki eruption (Iceland, 8 June 1783)—to test the effect of high-latitude eruptions on radiative forcing and climate (Methods section). To simulate the induced-climate impact of the volcanic eruption, we adopt present-day boundary conditions. Recent studies using an atmospheric general circulation model[29,30] suggest that the atmospheric circulation during the last deglaciation may have been similar to today over the North Atlantic, provided that the height of the Laurentide Ice sheet is lower than the Rocky Mountains, such as during the analyzed period[31]. Other studies[32,33] have also shown the predominant role of topography over sea surface temperature and sea-ice extent in altering North Atlantic atmospheric circulation. However, it is likely that sea surface temperature and sea-ice changes, together with a different strength of the Atlantic Meridional Overturning Circulation, had impacted atmospheric circulation primarily in winter, as suggested by some proxy data[34]. Our model experiments, on the other hand, focus primarily on the melt (summer) season (i.e., where temperatures are above 0 °C). Furthermore, to simulate changes in snowpack/ice mass balance in a more realistic fashion, we apply corrections to model output that allow us to drive our runoff model using Younger Dryas (YD) temperature, shortwave radiation, and precipitation conditions (Methods section). We also perform an additional experiment whereby we simulate an identical high-latitude eruption starting in winter and lasting for 4 months for comparison to the summer case where the eruption initiates midway through the year (Methods section).

Our climate model simulates a summer (JJA) cooling of approximately −3.5 °C over southern Scandinavia (55.8°–63.5° N, 5°–20° E) in response to the large summer high-latitude volcanic eruption, whereas no significant cooling is observed in the simulation where the eruption occurs in winter. In large part this difference in cooling is due to the weak solar insolation during winter. This not only leads to a reduction in the net shortwave radiative forcing (SWRF)[35] but it also limits the chemical reactions that form sulphate particles, which further decrease the SWRF. However, the winter experiment highlights that global climate altering sulfate emissions may not have occurred for all eruptions, though maintain the possibility of significant distal ashfall[36].

Where a summer eruption impacts SWRF this leads to a reduction in runoff which is not consistent with the formation of an ETV (Table 1; Supplementary Fig. 3). However, for runoff simulations driven by volcanically forced climate but with SWRF unchanged (i.e., if ashfall onto the ice sheet occurred within an exceptionally cool year within the range of natural variability), modest changes in albedo driven by volcanic ash deposition ($\Delta \alpha = -15\%$) can lead to increases in runoff that still more than offset these low temperatures at high elevations (Table 2; Supplementary Fig. 4). Furthermore, if a high-latitude eruption had no climate impact (for example occurring in winter or as most contemporary Icelandic eruptions) and only resulted in ashfall over FIS, this would result in significant increase in runoff over all elevations for small decreases in albedo (Table 3; Supplementary Figs. 5 and 6).

For eruptions that initiate during winter, where the impact of sulfate emission into the atmosphere predicted by the climate model is notably smaller relative to a summer eruption, large amounts of runoff can be triggered by even smaller changes in albedo (Table 4; Supplementary Fig. 7).

## Discussion

Greenland records of volcanism are particularly sensitive to high-latitude eruptions as compared to tropical eruptions owing to the close proximity to the source region. As such, high-latitude volcanic sulfate signatures are better represented in the ice than their tropical counterparts (Supplementary Fig. 8). In particular, Icelandic volcanoes remain the dominant source of volcanogenic aerosols in Greenland ice cores due to their relative proximity and high eruptive frequency[37].

The frequency of volcanic eruptions was considerably higher during the last deglaciation as compared to the last few hundred years[16] and most of these originated in formerly glaciated high-latitude regions[38]. This increased volcanic activity has been attributed to glacio-isostatic rebound that accompanied the retreat of Northern Hemisphere ice sheets[38–40]. Empirical studies support this hypothesis[41] and show that volcanic eruptions on Iceland were up to 50 times more frequent during the last deglaciation than during recent times. The highest eruption rates

**Table 3 Modeled change in runoff in response to ash deposition**

| Altitude (m) | Runoff (cm w.e.) | Runoff (cm w.e.) | Runoff (cm w.e.) | Runoff (cm w.e.) |
|---|---|---|---|---|
| | $\alpha = 1$ ($\alpha_{ir}$, $\alpha_s$) | $\alpha = 0.95$ ($\alpha_{ir}$, $\alpha_s$) | $\alpha = 0.9$ ($\alpha_{ir}$, $\alpha_s$) | $\alpha = 0.85$ ($\alpha_{ir}$, $\alpha_s$) |
| 0 | $0 \pm 0$ | $19.85 \pm 4.64$ | $39.32 \pm 4.61$ | $59.16 \pm 4.97$ |
| 500 | $0 \pm 0$ | $18.87 \pm 2.29$ | $39.51 \pm 3.34$ | $59.25 \pm 3.79$ |
| 1000 | $0 \pm 0$ | $18.76 \pm 2.15$ | $37.23 \pm 2.96$ | $56.51 \pm 4.29$ |
| 1500 | $0 \pm 0$ | $16.80 \pm 0.51$ | $36.04 \pm 3.42$ | $54.73 \pm 4.80$ |

Summary statistics of non-volcanically forced change in annual runoff (given in cm water equivalent, w.e.) and related standard deviations where only the effect of surface albedo changes due to ashfall are evaluated. The full simulation results are shown in Supplementary Fig. 5

on Iceland were recorded between ∼13,000 and 11,000 years BP[41], in contrast to a dramatic decrease in eruptive activity in Northwestern America[42].

Hence, it is likely that the majority of the volcanic eruptions recorded in the GISP2 record during the interval discussed in this study have an Icelandic origin. Moreover, we find no clear correspondence between the GISP2 volcanic sulfate anomalies associated with ETVs and Antarctic ice-core records that could indicate contributions from tropical volcanoes at times of increased FIS melting (Supplementary Fig. 1). Furthermore, there is a paucity of many large tropical eruptions over the time window ∼13,200–12,300 years BP and generally a higher frequency of volcanic eruptions recorded in Greenland as compared to Antarctica.

Critically, recent studies have shown that even moderate-size Alaskan[43,44] and Icelandic[36] eruptions (with associated moderate sulfate emissions) can result in a significant volcanic ash distribution over the Atlantic region, both easily reaching Northern Europe and Scandinavia. New findings[45] also indicate that volcanic ash clouds from historical mid-size eruptions of North American and Icelandic volcanoes have resulted in frequent ash fallouts over Northern Europe at intervals of <50 years, which are also not necessarily associated with climate altering sulfate emission. More importantly, instrumental observations show that such eruptions can be a major agent of glacier melting via tephra-induced surface albedo changes[3,4,46]. Equatorial eruptions on the other hand, even though they can have a global climate impact, result in very little (if any) ash deposition in Greenland due to their latitude.

Altogether, more frequent high-latitude volcanism during the final stage of the last deglaciation likely resulted in direct ash deposition across the FIS—predominantly from Icelandic sources just upwind of the ice sheet (due to the prevailing westerly flow). Moreover, these eruptions were likely to have occurred sub-glacially[22], which implies that were more likely to be explosive due to interaction with water, and therefore ash will have been transported over relatively longer distances. Finally, reconstructions[47] show that such explosive eruptions were mainly associated with mafic events that produced very dark tephra that are more effective at decreasing ice sheet surface albedo.

Results from snowpack simulations demonstrate that significant increases in summer and winter runoff can result from even a slight reduction in snow/ice albedo (Tables 2 and 3; Supplementary Figs. 4, 5 and 7). However, the seasonal timing and style of these eruptions will have been critical to how the ice sheet responded (Tables 1–4). Where runoff is enhanced, this will have had pronounced implications for the subglacial hydrological system of the FIS and sediment delivery to the Baltic Ice Lake, especially given the increased area of the ablation zone that is implied by the results. Comparing this scenario of an expanded ablation zone to contemporary ice sheet settings such as Greenland, greater runoff over a melt season has been linked to the formation of more extensive and efficient subglacial hydrological networks[48], which are then more likely to access and evacuate untapped sediment stores underneath the ice sheet[48].

In addition, extra runoff at high elevations will have increased the likelihood of supraglacial lakes forming and draining to the ice sheet bed[49]. These drainages are known to cause stepwise expansions in the subglacial hydrological network in an upglacier to downglacier direction, causing transient but substantial peaks in the suspended sediment concentration of proglacial meltwater once the supraglacial water connects to the pre-existing network[50]. This implies that enhanced runoff would allow a more extensive, efficient subglacial hydrological system to form, and hence supply extra suspended sediment to the Baltic Ice Lake to create an ETV.

By placing runoff response to volcanic forcing into the context of contemporary ice sheet mass balance and subglacial hydrological processes, this also suggests that the timing, style, and duration of eruptive events during the melt season will be crucial as to whether they are recorded in the varve record or not. For example, eruptions that are sustained throughout the melt season are more likely to produce the largest cumulative runoff response (Supplementary Fig. 7), and therefore the thickest varves. However, where sulfate emission from eruptions led to reductions in SWRF, this could have suppressed both runoff and the formation of an ETV. These factors provide a further explanation why the magnitude of sulfate and varve peaks are not scalable, while some apparently large volcanic eruptions are absent from the varve record.

The albedo mechanism (Fig. 3) is consistent with new evidence of enhanced short-term glacier melting in response to regional volcanism and ash-driven ice darkening during the early deglaciation in Alaska[42]. It is also consistent with the occurrence of a thick varve layer that is coeval with the deposition of the Vedde Ash in the youngest varve sequence of our chronology[51].

Further support to our interpretation of a dynamical mechanism behind the formation of ETVs is provided by the identification of Icelandic tephra horizons in NGRIP ice cores in relation with two sulfate-ETV isochrones[22] (Fig. 2; Supplementary Table 2). Although only two tephra layers have been identified over the period under investigation, this is likely an expression of the selective sampling approach that has been undertaken until recently[37,52], rather than a lack of ash deposition in ice cores[43,53,54].

As to the temporal relationship between sulfate and tephra deposition in Greenland ice cores, it has been shown[37] that aerosols and tephra are not always stratigraphically coeval. This stratigraphic offset, which does not exceed ±1 year, has been identified in GISP2 records[37]. The lead/lag generally arises when soluble and insoluble components are transported via different atmospheric pathways, especially in association with long-lasting eruptions. Under the assumption that the ETVs mainly reflect ash depositional events on the FIS, we observe no systematic lead/lag between aerosols and tephra deposition (Supplementary Fig. 2). However, this estimate is hindered by the fact that the GISP2 record cannot resolve potential offsets of the order of one year or less.

As mentioned above, volcanic aerosol can also have a cooling effect on climate[7,55,56], influencing atmospheric and ocean

**Table 4 Modeled change in runoff in response to a winter high-latitude eruption**

| Altitude (m) | Runoff (cm w.e.) $\alpha = 1$ ($\alpha_i$, $\alpha_s$) | Runoff (cm w.e.) $\alpha = 0.95$ ($\alpha_i$, $\alpha_s$) | Runoff (cm w.e.) $\alpha = 0.9$ ($\alpha_i$, $\alpha_s$) | Runoff (cm w.e.) $\alpha = 0.85$ ($\alpha_i$, $\alpha_s$) |
|---|---|---|---|---|
| 0 | $4.38 \pm 4.64$ | $37.56 \pm 5.40$ | $72.79 \pm 6.00$ | $105.46 \pm 6.57$ |
| 500 | $4.61 \pm 2.07$ | $36.18 \pm 3.43$ | $67.00 \pm 5.09$ | $98.45 \pm 6.29$ |
| 1000 | $2.06 \pm 0.82$ | $30.65 \pm 2.22$ | $59.12 \pm 3.55$ | $87.67 \pm 4.67$ |
| 1500 | $0.40 \pm 0.40$ | $28.54 \pm 1.60$ | $55.92 \pm 3.68$ | $82.19 \pm 4.99$ |

Summary statistics of volcanically forced change in annual runoff model results (given in cm water equivalent, w.e.) and related standard deviations for a winter high-latitude volcanic eruption. The full simulation results are shown in Supplementary Fig. 7

circulation on sub-decadal time scales[8] that in isolation suppress ice-sheet and glacier melt[57,58]. Given that our data suggest that enhanced runoff is a transient response to ash deposition, this does not contradict the notion of glacier expansion in response to volcanic aerosol cooling occurring over time scales longer than 1 melt season[59]. Conversely, the results of Table 1 (Supplementary Fig. 3) are in agreement with this where eruptions can have significant climate impacts. However, as our runoff experiments show, albedo change can more than cancel out the runoff response to cooling if SWRF is not significantly impacted (Table 2; Supplementary Fig. 4).

Recently, it has been hypothesized that millennial-scale Southern Hemisphere volcanism triggered an asymmetric warming in the Northern Hemisphere during the last deglaciation[60]. This warming mechanism could provide a driving force for Northern Hemisphere ice sheet melt and thereby for enhanced volcanism via isostatic rebound. However, no large eruptions have been detected in Antarctic ice-core records during the period investigated in our study. Moreover, there is no evidence for short- or long-term warming events between the end of GI-1 and the first half of GS-1. On the contrary, it has been shown that this period was characterized by gradually colder summer conditions in Northern Europe and especially in southern Scandinavia[61,62]. Therefore, we also dismiss the hypothesis that enhanced short-term ablation rates of the FIS are attributable to long-term regional warming.

As a final remark, we observe that one ETV corresponds with the catastrophic drainage of the Baltic Ice Lake (12,847 years BP)[11,13], which is likely responsible for the abrupt hydro-climate shifts captured in Greenland ice cores and associated with the initiation of the YD stadial—GS-1 (12,846 years BP). Although speculative at this point, a causative link cannot be ruled out. Thus, further work is required to verify the impact of volcanic aerosol forcing and ice surface albedo changes on the recession of the FIS beyond the spillway threshold in south-central Sweden.

In conclusion, evidence from our glacial varve records indicate that during the last deglaciation volcanism caused more extensive melt and enhanced levels of runoff from the FIS. We suggest that this was primarily an expression of decreased ice sheet albedo owing to deposition of dark volcanic ash. Our results highlight the sensitivity of ancient ice sheets to volcanogenic aerosols and the necessity to employ dynamic and interactive ice-sheet components in climate models that reproduce ice configuration changes in response to external forcing. This study also provides motivation for further investigations with regard to the mechanisms behind catastrophic freshwater surges of the past and their pivotal role on rapid climate change.

## Methods

**Varve chronology.** The chronology provides an annually resolved and continuous record where the potential problem of missing varve years is overcome by cross-dating several overlapping varve-thickness records. The precision of the chronology is verified by the (i) general lack of disturbed layers and good preservation of the varves in all the sequences that compose the unified chronology, (ii) the evenly

high lateral consistency of numerous adjacent—and distal—cross-correlated sequences[12], and (iii) the internal chronological consistency verified by statistical analysis, independent[14]C dating, and well-defined biostratigraphic marker horizons, respectively[12,13]. An overall uncertainty (entailing precision and accuracy) of $\pm 0.5\%$ ($2\sigma$) has been assigned to the varve chronology[13]. However, this should be considered as a highly conservative estimate.

**Statistical analysis.** The varve thickness record was filtered using a low-pass spline to remove harmonic functions below the 20th degree, which is similar to dividing the time series by a low-pass filter with a cut-off frequency of 1/25 year. The spline was preferred to other filters based on the fitting performance and we observe that the following results are independent of the type of filter used. The time series was then divided by its standard deviation. Finally, we identified anomalous melt events as years characterized by an annual varve thickness that exceeds $+2\sigma$ (defined as exceptionally thick clay varves—ETV).

Prior to comparison with ice-core volcanic records, we linearly interpolated the volcanic sulfate time series to annual resolution to avoid smoothing of the ETV record. This allows the sulfate peaks to accommodate a large part of the age uncertainty associated with each potential ETV isochrone ($\pm 1\sigma$; $\pm 1.75$ years), which is thus directly accounted for when testing the significance of the correlation using the Monte Carlo tests described below. A maximum cumulative mismatch of 0.15% between the Vedde Ash and the end of GI-1 is inferred between the glacial varve chronology and the GICC05 chronology[13]. This suggests that the two records are evenly synchronous during the interval under analysis and that the age uncertainty due to sampling resolution, and allocated to each $SO_4^{2-}$ sample, is not higher than the relative mismatch between the two chronologies.

To evaluate the significance of the correlation between volcanic eruptions and ETVs, we employed three independent Monte Carlo tests. The first approach is a simple permutation analysis whereby the number of original ETVs is randomly shuffled within the time frame of the chronology for 1000 times and for each realization the coherence with the maximum resampled volcanic sulfate anomalies is evaluated. In the second approach the significance of the coherence is inferred via comparison of the GISP2 data to 1000 individual realizations of the varve thickness record with similar red noise spectral characteristics. In the third test the coherence is inferred via comparison of the GISP2 data to 1000 individual Gaussian white noise realizations of the varve thickness record.

**Climate model simulations.** We use the Norwegian Earth System Model (NorESM1-M)[63,64] to simulate an extreme high-latitude multistage eruption under present day conditions. NorESM1-M has a horizontal resolution for the atmosphere of 1.9° (latitude) × 2.5° (longitude) and 26 vertical levels. NorESM1-M uses a modified version of Community Atmospheric Model version 4 (CAM4)[65], CAM4–Oslo with the updated aerosol module, simulating the life cycle of aerosol particles, primary and secondary organics. The atmospheric model is coupled to the Miami Isopycnic Coordinate Ocean Model—MICOM. A detailed description of the model used in this study can be found in Bentsen et al.[63], Iversen et al.[64], Kirkevåg et al.[66], and Pausata et al.[9]. The model performance has also been evaluated in these studies from a basic validation of the physical climate[63], to the climate response to future climate scnarios[64], to aerosol-climate interactions[66], and as well as to high-latitude Laki-type volcanic eruptions[9].

We mimic the largest high-latitude volcanic eruption in recent history—the Laki (Iceland, 8 June 1783) eruption—to simulate the effect of high-latitude eruptions on radiative forcing and climate. We simulate the high-latitude eruption by injecting 100 Tg of $SO_2$ and dust (median radius = 0.22 μm in accumulation mode), as an analog for ash, mostly into the upper-troposphere/lower stratosphere over a 4-month period. We start the eruption on 1st June in order to replicate the original Laki eruption (details regarding the set-up are provided in refs. [9, 10]). We also perform an identical Northern Hemisphere high-latitude eruption, but with start date on 1st December and lasting the full duration of the melt season in order to appreciate the climate effect of such eruption in winter and compare it to the summer case.

We analyze an ensemble of 20 simulations with each member starting from a different year selected from a transient historical simulation (1901–1960). The no-volcano ensemble is obtained by simply considering each of the unperturbed

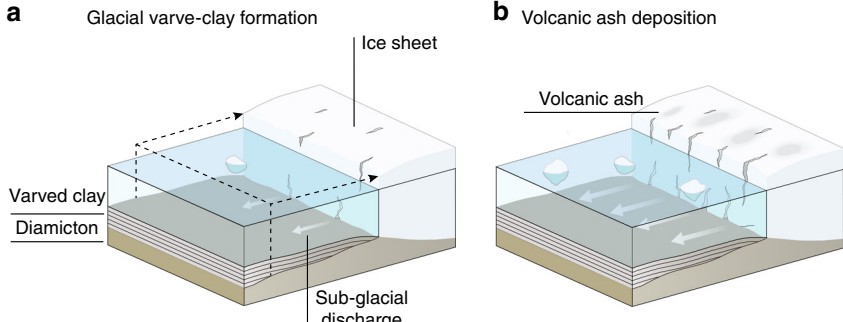

**Fig. 3** Illustration showing the formation of glacial varved clay in response to volcanic eruptions during the last deglaciation of the Fennoscandian Ice Sheet. **a** Melting of the ice in the ablation zone of the ice sheet during the summer season contributes large volumes of subglacial meltwater with high sediment load to the Baltic Ice Lake. This results in lake-bottom currents and deposition of fine sand and silt layers. The clay layer formed in winter when lake ice covered the Baltic Ice Lake. The two layers or couplets are thus associated with the melting and non-melting season, respectively, and form one varve. **b** Volcanic eruptions result in ash deposition on the ice-ablation zone, which enhances melting and thus subglacial sediment discharge by lowering the ice albedo

years from the historical simulation corresponding to the same years of the perturbed case.

To account for the changes in boundary and climate conditions in terms of solar insolation, surface temperature, and precipitation during the late phase of the deglaciation (YD) over southern Scandinavia (55.8°–63.5° N, 5°–20° E), we use the output from the simulations performed with the Community Atmospheric Model version 3 (ref. [67]). We use these data as no-volcano control experiment to force the runoff model, described in the section below. Then we imposed on the YD control simulation the area-averaged volcanically induced anomalies extrapolated from the NorESM coupled simulations. The volcanically induced anomalies inferred using this approach are only meant to be indicative of the climate anomalies during the late deglaciation.

**Runoff model simulations**. We apply a field-validated physically based one-dimensional energy balance model of melt, refreezing, and runoff processes that occur within a given snowpack/ice column[68]. The runoff model has previously been field validated for Devon Ice Cap, Nunavut, Canada, where it was shown to be capable of fully accounting for density, temperature, and albedo evolution of the snow/ice column[68]. It is able to account for feedbacks associated with melt, refreezing within the snowpack/ice and runoff, and therefore provides more physically realistic runoff estimates than simpler degree day models of surface mass balance. The model uses hourly values of air temperature, cloud cover, precipitation, relative humidity, and incoming shortwave radiation flux (SWRF) that are interpolated to 15 min timesteps. A full description of the model is in Morris et al.[68], however a modification in this study is that we account for changes in cloudiness, whereas constant conditions were assumed previously.

Runoff is determined by calculating both energy and mass balances for a 9.5 m column of ice/firn/snow at 1 cm intervals, and determining the potential for melt, refreezing, and percolation through to a dynamically determined impermeable ice layer (i.e., where density is that of solid ice, or where it has increased to achieve that value due to refreezing within the firn/snow). For each timestep, melt that does not refreeze within the snow/ice column is lost as runoff.

The ability of the model to account for melt, percolation, energy balance, and snow/firn density changes through refreezing represents a much more realistic way of simulating runoff compared to models that calculate melt only. For this study, the distinction between melt and runoff is important, since refreezing within the snowpack means that the former does not necessarily translate to the latter. Similarly, the model accounts for the fact that capacity for refreezing is not constant through time, and will evolve due to density and englacial temperature changes (e.g., Supplementary Figs. 3–5 and 7). The runoff values generated are therefore physically robust estimates of meltwater that would be potentially available to access the subglacial hydrological system of FIS, and therefore entrain suspended sediment that could contribute towards forming varves.

Climate model outputs were used to drive the runoff model, though given that the majority of the relevant output for the former is at daily or monthly resolution, it was necessary to add climate model constrained sub-daily variability to the original air temperature, SWRF, and relative humidity values. A description of how each runoff model input was derived at hourly intervals is outlined below.

Daily minimum, mean, and maximum air temperature values are generated by the climate model. A simplifying assumption is made that these temperatures occur at 00 h, 06 h/18 h, and 12 h respectively. Values for the intervening hours are based on a piecewise cubic interpolation of the values available. Transformation of these values to temperatures typical of the YD were undertaken by applying a piecewise cubic interpolated monthly mean correction to all values.

The climate model provides monthly mean cloud cover values. These are interpolated to hourly values using piecewise cubic interpolation.

Values for precipitation are provided at daily resolution from the climate model. These daily totals are divided evenly through that particular day (i.e., a day with 24 mm of precipitation would be included within the model as having 1 mm/h of precipitation). A threshold temperature of 0 °C is set, at which or below precipitation will fall as snow and contribute to the snowpack rather than fall as rain where it can contribute to percolation, and provide extra thermal energy to the snowpack/ice, freeze, and/or runoff.

Daily mean absolute humidity values are provided by the climate model that are interpolated to hourly observations before being converted to relative humidity values using the air temperature values used to drive the runoff model.

The climate model provides monthly mean values for SWRF. However, this value will vary substantially over the course of a day, and is non-trivial in providing energy to the snow/ice surface for melting. As such, it was necessary to recalculate SWRF outside of the climate model to capture its diurnal variability. This was achieved using the solar insolation tool of ArcGIS (ESRI) to calculate the daily insolation for a topographically unshielded point at 60° N for a uniform sky. The diffuse fraction value was calculated using the cloud cover values from the climate model, by assuming that at the extremes of cloudiness, zero cloud cover equated to 20% diffuse contribution, while total cloud cover equated to 70% diffuse contribution (http://desktop.arcgis.com/en/arcmap/10.3/tools/spatial-analyst-toolbox/points-solar-radiation.htm). This provided hourly SWRF values that are consistent with the time of day, time of year, and the cloudiness conditions given by the climate model. Volcanically forced SWRF is calculated by multiplying the above by the fractional difference between monthly non-volcanic and volcanic SWRF output given by the climate model. To ensure the smooth evolution of SWRF for use in the runoff model, the monthly fractional differences between non-volcanic/volcanic climate model outputs are interpolated to the same temporal resolution as runoff model inputs. This ensures that inputs to the runoff model capture both diurnal variability of SWRF and any effects of volcanic forcing predicted by the climate model. The differences between non-volcanically and volcanically forced SWRF are shown in Supplementary Fig. 6.

Idealised simulations for a range of snowpack/ice conditions are conducted for both volcanic and non-volcanically forced conditions, as determined by climate model output and prescribed albedo changes due to ash fall. Specifically, simulations were undertaken using temperature, SWRF, and precipitation conditions using climate model output adjusted to represent YD conditions (non-volcanic control) and those altered by a Laki-type eruption (volcanically forced) occurring (i) in summer (volcanic forcing initiated on 1st June), and (ii) winter (volcanic forcing initiated on 1st December). Each set of simulations had spin-up periods preceding the initiation of the eruption, with the summer eruption simulation beginning on 1st January, and winter eruption on 1st October. These initiation dates were chosen to allow (i) sufficient time for the ice column temperature respond to the atmospheric conditions determined by the climate model output, and (ii) not to include any period of time from the preceding melt season (i.e., where temperature was >0 °C). Consequently no melt and no density changes to the column occurred during the spin-up period. Precipitation was also reduced to zero until the initiation of volcanic forcing or the first day of positive temperatures (whichever occurred earlier). This allowed ice column conditions for volcanically and non-volcanically forced simulations to be directly comparable.

There are many uncertainties associated with modeling the mass balance of palaeo ice sheets. Rather than avoiding them, our modeling approach seeks to explore these uncertainties in order to characterise the potential range of response for the volcanic and non-volcanically forced scenarios. The scenarios tested involve applying an atmospheric lapse rate of −5.4 °C km$^{-1}$ to evaluate runoff response at sea level, 500, 1000, and 1500 m elevation, in addition to four different albedo forcing scenarios simulating the effect of ash fall for the volcanically forced scenarios only. The range of albedo changes applied (0, 5, 10, and 15% reductions

in the albedo of both snow and ice) represent modest absolute reductions compared to initial values that are derived values from Devon Ice Cap, Nunavut, Canada (alpha_s = 0.81, alpha_i = 0.65). Consequently, each suite of simulations comprises of 16 different volcanically forced scenarios (varying elevation and albedo), and four different non-volcanically forced scenarios (varying elevation only).

The large range of uncertainty in initial snow/ice column conditions for each scenario is also explored systematically. This is achieved by running the model for a range of initial snow/firn thicknesses, simulating 0–100 cm thicknesses of each at 10 cm intervals. Each individual scenario is therefore tested for 121 different sets of initial conditions. Consequently, the runoff potential of each ensemble of scenarios is evaluated for 2420 unique combinations of conditions (i.e., 4840 individual simulation scenarios). This allows runoff response to be considered across an elevation range of the FIS, and against different albedo forcing scenarios for a wide range of potential initial conditions.

It is also possible to assess the relative contributions to runoff due to each environmental variable (changes in volcanically forced temperatures, cloudiness, precipitation, and albedo) to be evaluated in combination and/or isolation, through comparison to a non-volcanically forced ensemble of simulations. The results of simulations where the effects of temperature and cloudiness are evaluated in isolation are shown in Supplementary Fig. 9.

Finally, to test the significance of the variability of cloudiness in controlling runoff, we have also conducted an ensemble of melt/runoff simulations where we add random noise (at a daily timescale) to the cloudiness data. This aims to evaluate the impact of short-term changes in this input on the overall trends and magnitudes of the runoff results generated by the runoff model. The results of these simulations are presented as the difference between the simulations where the noise has been added to the cloudiness data (Supplementary Figs. 10 and 11) and the original simulations. The original cloudiness values were obtained by interpolating monthly mean cloudiness values from climate model output as described above, while the maximum magnitude of the noise added to the original cloud cover fraction data is ±0.3. The same pattern of noise was added to both the volcanical and non-volcanically driven simulations. This ensures that both simulations are consistent, and only the impact of cloud cover variability on the overall results is evaluated. The monthly means of the cloudiness data with noise added are consistent with those of the original simulations.

In addition, adding noise to the cloudiness values will also change the shortwave radiation fluxes compared to the original simulations. Consequently, for the new simulations the SWRF is recalculated following the same method described above. All remaining data used to drive the new simulations are consistent with those of the original simulations. Supplementary Fig. 11 shows the difference between volcanic and non-volcanically forced simulations with noise added to the cloudiness data compared to those using the monthly data. These results show that introducing daily timescale white noise to the cloudiness input data leads to negligible differences in runoff between the two sets of simulations (Supplementary Fig. 11). Consequently the impact of introducing noise to the cloudiness inputs is relatively small where it is applied to both the vocanically and non-volcanically forced scenarios, and where it does arise is likely due to feedbacks due to differences in refreezing of melt within the snowpack. The full ensemble of simulations conducted therefore represents a comprehensive analysis of the runoff response to volcanic forcing for a full range of potential snow/ice conditions at different elevations of the FIS. Given the uncertainties in simulating runoff for a palaeo ice sheet (e.g., initial snowpack conditions, ice sheet profile, and equilibrium line altitude), the absolute values given for each individual simulation by the runoff model should be treated with caution, though are likely to fall within the range of values within scenarios tested. Consequently, the direction and relative magnitude of runoff response should be taken to provide meaningful relative indication of the sensitivity of FIS runoff to volcanic forcing.

**Data availability**. The varve chronology presented in this study is available along the online version of this article on the publisher's web-site. All the model data and runoff model codes are available from the authors upon request.

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

## Acknowledgements

F.M. is funded through a Lamont-Doherty Earth Observatory Postdoctoral Fellowship grant. F.S.R.P. is funded by the Swedish Vetenskapsrådet as part of the MILEX project. This work is a contribution to the INTIMATE project and the ERC-Synergy granted ice2ice project.

## Author contributions

F.M. conceived the study, performed the statistical analysis, and wrote the first draft of the manuscript. F.S.R.P. designed and performed the climate model experiments. J.M.L. designed and performed the runoff model experiments. D.W.F.M. provided the runoff model code. B.W. provided the clay varve data sets. All authors contributed the interpretation of the results and editing of the manuscript.

## Additional information

**Competing interests:** The authors declare no competing financial interests

