## [Peer Review File · Nature Communications]

Reviewers' comments:

Reviewer #1 (Remarks to the Author):

Corresponding Author: Francesco Muschitiello

Title: Impact of volcanic aerosols on Fennoscandian Ice Sheet melting during the last deglaciation

A. Summary of the key results

The authors use a recently updated glacial varve chronology from Southern Sweden together with an ice-core sulphate record from Greenland to compare and discuss the timing of anomalous ice melting events from the retreating Fennoscandian Ice Sheet (indicated by exceptionally thick clay varves) with increased volcanic activity during the last deglaciation (12.4-13.2 ka BP). Thick clay varves (i.e. strong melting) are found to have occurred often synchronous to increased sulphate deposition in Greenland (i.e. volcanic eruptions), with this observed coherence being stronger as expected by chance on the basis of Monte Carlo tests. Taking into account results from climate model simulations for modern analogues of volcanic eruptions (Tambora, 1815, Laki 1783) the authors hypothesize that large volcanic eruptions during this time strongly contributed to the Fennoscandian Ice Sheet melting by 1) increasing the cloud cover over the ice-sheet and 2) changing the snow albedo through deposition of ash on the ice surface.

B. Originality and significance: if not novel, please include reference

The deglacial climate is marked by rapid warming events in the North Atlantic, atmospheric circulation changes, three abrupt CO₂ pulses, melting ice-sheets and rising global sea-levels. Rapid thermal changes originating in the North Atlantic may have caused re-organization of the atmospheric circulation on timescales as low as 3 years [Steffensen et al., 2008]. At the same time geologic records of volcanic eruptions show that volcanism increased during the deglaciation above Holocene and Glacial background levels [Huybers & Langmuir 2009]. Fully disentangling cause and effects of these changes requires integration of well synchronized proxy records and climate model results. However, there is a paucity of long, well-dated and high-resolution records to explore potential connections between volcanism and large ice-sheets. The current study tries to fill this gap by using two annually dated and partly synchronized proxy records of subaerial volcanism and rapid ice-sheet melting events during the deglaciation.

Previously, volcanism was hypothesized to have had a positive climate forcing during the deglaciation by increased CO₂ emissions [Huybers & Langmuir 2009]. Here, increased cloudiness and decreased albedo through ash deposition are proposed for the first time as forcing mechanism explaining the rapid ice-sheet melting pulses deduced from the varve record. The hypothesized positive volcanic contribution to increased ice-sheet melting is a novel concept and in opposition to the well-known radiative cooling effects of historic volcanic eruptions [Stoffel et al., 2015] that are often associated with increased glacier growth during volcanic active periods (e.g, Little Ice Age). Volcanic forcing is considered as one of the most important decadal-scale drivers of glacier advances during the Neoglacial [Le Roy et al., 2015; Solomina et al., 2015; Miller et al., 2012].

C. Data & methodology: validity of approach, quality of data, quality of presentation

The previously published glacial varve record is of high quality regarding resolution and age-control. The statistically derived coherence between subaerial volcanism and rapid glacier melting itself is intriguing, however, the approach applied for analyzing the potential causal links to past volcanic activity uses some very simplified assumptions that in my view are not substantial enough to derive a causal nexus between volcanism and melting via a positive cloud radiative forcing and ash induced albedo changes on the ice-sheets surface.

To further disentangle potential leads and lags in the relationship between volcanism and ice melting, it would be helpful to take full advantage that both proxy records are based on annually dated chronologies and to also discuss if the number of annual-layers between the potential

isochrones were consistent with the respective dating uncertainties of the chronological models. The mechanisms suggested to be involved in the positive climate forcing that caused the melting are based strongly on the model output of a single climate-aerosol model, and are shown here only for single month (June). Driven by the aerosol forcing of two of the largest eruptions of known history the simulations are performed under present-day boundary conditions and show only relative moderate cloud cover changes of 15% (predominantly for high-latitude eruptions). Under the assumptions that during deglaciation 1) atmospheric volcanic aerosol content was much larger, 2) volcanic eruptions occurred more frequently and 3) may have involved also deposition of light-absorbing ash on the ice sheet, these model results are extrapolated to deduce an important role of volcanism on contributing to increased ice-sheet melting via a positive cloud and albedo feedback. The validity of these assumptions are, however, not always supported by strong empirical evidence.

Questions not sufficiently addressed also include:

- 1) What is the sensitivity of the simulated cloud cover increases to the choice of the aerosol/climate model and the respective month? Do other models show comparable results for Scandinavia?
- 2) Are simulations under present-day conditions suitable to deduce the climatic response during such a rapidly changing climate state as during the last deglaciation?
- 3) Can you better characterize the volcanic eruptions that correspond to your ETVs with other proxy evidence from ice cores (e.g. tephra)? Are these ash-rich events, high- or low-latitude eruptions? Dynamical and radiative climate effects are quite different for these eruptions.
- 4) Radiative cooling is among the better understood volcanic effects and is believed to have caused increased glacier mass balances as observed and reconstructed after historic events. Thus your proposed positive forcing needs, in addition, to compensate for these cooling effects in order to contribute significantly to the ice sheet melting.

D. Conclusions: robustness, validity, reliability

The well-dated high-resolution glacial varve record shows a strong coherence with volcanic aerosol concentrations, suggesting that extreme melting events may be somehow interlinked with the volcanic activity. The datasets used and the analysis performed, however, do not fully support the conclusion that the melting events had been directly forced by volcanic eruptions. On the basis of the presented data it cannot be ruled out that both processes are interlinked via a third process that is driving ice-sheet melting as well as increased volcanism. Alternative causes for rapid changes in climate and atmospheric circulation that seemed to have occurred on very short timescale [Steffensen et al. 2008; Baldini et al., 2015a] should therefore be discussed as well (including for example alternative hypotheses involving volcanic eruptions e.g. Baldini et al., [2015b])

E. Suggested improvements: experiments, data for possible revision

Any experiment or dataset helping to answer the questions raised under C. will strongly benefit the interpretation of these datasets. With 30% of all samples in the time window covered by this study labeled as "volcanic" [Zielinski et al., 1996] the coherency analysis would also strongly benefit from a higher resolved and/or more precisely dated volcano record to avoid spurious correlations caused by the low sampling resolution. The NGRIP record appears higher resolved but seems not to capture volcanism during the GS-1. Is there any chance that the new NEEM record (your reference 23) can improve resolution and dating precision of the volcanic sulphate record?

F. References: appropriate credit to previous work?

Previous work is credited, but some references suggesting opposing (i.e. cooling) effects of volcanism or the importance of asymmetric forcing for the dynamical climate response are missing in the discussion (e.g. post volcanic runoff-reduction in Greenland; glacier growth in volcanic active periods; changes in atmospheric circulation after volcanic eruptions and during the Younger Dryas; alternative mechanism suggested to link volcanoes with North Atlantic climate variability)

-

Additional Comments:

L. 31: "...aerosol particles deposited on snow can alter..."

L. 35: Many citations are given for the various volcanic effects on various climatic parameters; some are redundant e.g. 7,12,15; Citation 9 is showing that volcanic cooling in polar regions was suppressing ice sheet melt after some historic eruptions, which is the opposite of the proposed mechanism in this paper. A negative forcing and growth of ice caps and mountain glaciers during volcanic active periods is also supported by other studies (see references above). Delete reference with super-eruption (ref. 11) because no such eruption took place during the deglaciation.

L. 37: Similar simulations from Oman et al. [2005, 2006] using GISS Model E for Katmai and Laki show only minimal changes in cloud cover over Scandinavia (2%) but roughly 1°C surface cooling.

L. 37: "Precise synchronization to a(n annual-layer dated) Greenland ice-core chronology"; there are several chronologies available for Greenland ice cores.

L48: Isn't precision more important than accuracy for your type of analysis? The absolute accuracy is of limited importance because you have a common stratigraphic time marker and two annually laminated records.

L. 62: What is the age of the Vedde Ash? Please add this information here.

L65-66: Given this impressive agreement in independently derived numbers of layers between the two marker horizons I was expecting a more detailed comparison of the volcanic marker events and the proposed candidate melt events taking into account estimated uncertainties of the underlying records respectively (see for example Fig. 7 in Svenssen et al., 2013).

L. 71: add time, e.g. older 12,400 years BP

L. 80: Consider replacing the judgment "established record" with something like "...contains an imprint of past volcanic activity" or "can contain the fallout of past volcanic activity"

L. 86: Replace ref. 26 with a more recent publication from Seiderstadt et al., [2014]

L. 90: Define "in phase"! I assume you use the original volcanic attribution from Zielinski et al., [1996]? According to their record roughly 1/3 of their total sulphate samples (12,400-13,200 yr BP) are attributed to volcanic eruptions. Depending on the threshold for a match one always must expect a high number of ETVs occurring in phase. How many SO₄ anomalies are reported for the according time period? If you align and compose the varve segments relative to the volcanic sulphate peaks do you derive a clear melting signal relative to the background?

L. 94: How do you define the years according to the volcanic events? Year of maximum SO₄ concentration? Year of maximum annual resampled anomalies? Start of increased SO₄ deposition? Some volcanic signals last up to 18 years, and also the one discussed here is dated 12553-12545 yr BP.

For this example the proposed forcing occurred 10-20 years later than the proposed ice-sheet response. Is such an offset still compatible with the age uncertainties for the varve record and the ice-core chronology?

L. 96: Can the higher resolved ECM record provide information if two adjacent eruptions contributed to the broad volcanic sulphate signal?

L. 100-101: Why is that? The ice-core record should record high-latitude events as well as tropical

eruptions, but one cannot tell on the basis of GISP2 alone if individual signals are from tropical eruptions or high-latitude events.

L. 101-110: This is all true, but I cannot see the full relevance of the discussion here. Do you want to imply that the sulphate record from GISP2 is incomplete and may miss some eruptions? Or have false positives, such as SO₄ anomalies that are not of volcanic origin?

L. 109: Which volcanic markers? Tephra for sure, but sulphate is deposited widespread and over longer time and therefore very unlikely to be missing.

L. 110: "...can even be missing in single/individual ice core records".

L. 114-115: This is likely true. Moreover, the different volcanic effects on climate discussed earlier also operate on very different temporal scales.

L. 118: "Magnitude of atmospheric SO₄ input" is a metric not known. Volcanic and total sulphate concentrations are the only metrics provided for GISP2.

L. 121: The correspondence may also be a consequence that both, volcanism and rapid ice-sheet melting, are both responding to the same events (i.e. rapid surface warming).

L. 130: Are the results of these simulations consistent with those from other models? I believe a large number of other models have performed simulations for this two specific, most recent, extreme events.

L. 131: An increase of 15% in cloud cover does not appear very much. Is this strong enough to cause extreme ice-melting, considering that the 30% increase in precipitation (some of which may be snowfall?) over the ice-sheet will compensate for some of the loss.

L. 138: I would find it more convincing if you could provide (in addition to the MC simulations) for the 16 ETVs and their associated age uncertainties the ages and age uncertainties of the candidate volcanic events. This would allow to addressing if the response is lagging/leading the proposed forcing and if the association is consistent with the individual age models.

L. 141-144: These effects need to be able to compensate opposing and relatively well documented and modeled effects such as summer cooling resulting in a positive glacier mass balance [Hanna et al., 2005, your ref. 9; Solomina et al., 2015].

L. 148: If cloud-induced runoff with respect to clear skies is only +30%, then the runoff increase after a simulated +15% cloud cover would only be in the order of 5% (assuming a linear relationship).

L. 154-156: Please specify which kind eruptions (tropical, extratropical, glaciated or non- glaciated areas) experienced an increase. What are the metrics and empirical evidence for the statement that eruptions were "larger"? Reference 35 only discusses frequencies. Figures 2 and 4 also don't allow to drawing conclusions about the magnitudes and trends of past volcanism relative to historic analogues because injection mass and ice concentrations are different parameters (and units).

L. 159-161: This comparison is not valid as the data resolution and annual accumulation rates are very different between historic and the late glacial events. If you were to compare these events you would need to use the same metrics for a comparison (e.g. thinning corrected volcanic sulphate deposition flux).

This also applies to Figure 2. There is no axis given showing the SO₂ injection for the three historic events. It almost appear like the arrows are meant to correspond to the y-axis showing the GISP2 volcanic sulphate concentrations (in ppb), which would be wrong as SO₂ injection does not scale

linearly to volcanic sulphate concentrations in the GISP2 ice. It may well be correct that some of the deglacial eruptions injected large amounts of SO₂ causing huge sulphate spikes in GISP2, but this has not yet been demonstrated.

L. 164-172: The statement that sulphate originating from N-American eruptions may be stronger represented in an ice core from Greenland compared to sulphate from eruptions downwind (e.g. from Iceland), but again the relative sulphate amplitudes of Katmai and St. Helens relative to Laki is not a valid metric to demonstrate this, because a large but unknown fraction of the total sulphate from these recent events is caused by anthropogenic sulphate emissions [Fyfe et al., 2013].

L. 173-178: The intensity estimated as the summed volume of erupted magma over time is strongly biased by few magma producing large eruptions forming vast lava shields in contrast to the fissure eruptions during the Holocene period [Maclennan et al. 2002, your ref. 38]. It is unlikely that the increased intensity of Icelandic volcanism scales in a linear way with the atmospheric aerosol content. Using the GISP2 SO₄ record Zielinski et al. [1996] estimated the frequency of (all) eruptions around 12 ka BP at 20-25 eruptions per ka, which is only 2 to 2.5 times higher than the overall average frequency; and about 4-5 times higher than the Holocene frequency. Given that Laki and Tambora are among the largest events of the past 2,000 years it seems unlikely that the volcanic eruptions between 12-13.2 ka were on average larger than these two extreme events.

L. 179-182: Is there any empirical evidence of increased ash generation and widespread deposition during these time periods? Tephrochronologists have been working on these ash layers in Greenland ice cores for years (e.g., Abbott and Davies [2012]). Is there an increase in the frequencies of ash layers during the deglaciation? Or is there any evidence of increased melt in the varve record following the Vedde Ash, for which the spatial extent towards the FIC is well known? Any change of the surface albedo would only result in increased melt during the spring/summer season and only until the ash is buried by subsequent snowfall [Hansen and Nazarenko 2004].

L. 183-195: You may want to discuss here also competing hypotheses. For example, Baldini et al., [2015b] recently suggested that asymmetric forcing resulting from Southern hemisphere and Northern hemisphere eruptions changed the ITCZ which may have triggered rapid warming/cooling events in the North Atlantic. While I am not convinced that this hypothesis is strongly supported by their low-resolution geological records, their analysis highlights at least the complexity of the dynamical climatic response w.r.t. to the location of the eruption which is also reflected by observations and climate models from more recent eruptions [Haywood et al., 2013; Ridley et al., 2015]. On the basis of increased sulphate deposition coeval with melting of the FIS only, the suggested important impact of cloud and ash albedo forcing on the ice sheet finds, in my view, only little support. If the authors could demonstrate that high-latitude eruptions or such eruptions that produced widespread ash deposits were preferentially contributing to these melt-events, these would strengthen the case for a direct causal relationship. With the available data, the statistically significant relationship between volcanic eruptions and melt events may as well be driven by a common third driver, such as rapid temperature rises, increasing ice unloading and subaerial emissions in the volcanic source regions more or less synchronous to the melting events of the FIC.

L. 201: Can you give an estimate of the precision of the annual-layer counted chronology over the 1257-year long chronology? The "Maximum Counting Error" for GICC05 for example over this section is increasing from 110 to 145 years suggesting a relative uncertainty of 35 years for the full record. Is the varve record comparable? Or even better?

L. 216-226: As pointed out earlier, a table with the relative (to Vedde Ash) age uncertainties for the candidate match events of the 16 ETVs for both records would be helpful (in supplement).

L. 248: Which size mode? How different would the results be without the "dust"?

L. 261-269: The statement that large-scale atmospheric circulation should have been similar to modern conditions on the basis of a single regional modeling study from the Laurentide Ice Sheet comes as a surprise. Simulations and proxy records both show the deglaciation had a strong influence on the regional structures and storm tracks of the Arctic and Europe, resulting for example in the migration of the westerly wind position, changes in the sea-ice extent and subsequent atmospheric reorganization, or changes of the AMOC [Baldini et al., 2015a; Ullman et al., 2014; Dethloff et al., 2004; Steffensen et al., 2008, Chen et al., 2015]. Given the amplitudes and rates of changes occurring during this time period, especially in the North Atlantic sector, the simplified assumption used for the climate-aerosol simulations are unlikely representing realistic boundary conditions.

Fig. 2:

Add separate axis for three historic eruptions. What happened with the "acidity" during GS-1?

Fig. 3:

What about July, August? Did you only show June because Laki started in June? If there is no cloud increase in the other summer months, how does the forcing work? One cannot expect that all past eruptions occurred in June.

Fig. 4:

Labeled as Fig. 3; Laki was in 1783 not 1781

Although labeled "volcanic sulphate" by Zielinski et al. [1996], it is clear that the sulphate record after about 1900 is strongly driven by anthropogenic emissions superimposed on the volcanic emissions. Thus a direct comparison of the "volcanic sulphate" levels of GISP2 between Laki and Mt. St. Helens (to some extent also Katmai) will lead to wrong results. With both "upwind" eruptions falling in the 20th century, the interpretation of the dominance of the "upwind" eruption is also not supported by this data.

References:

Abbott, P. M. & Davies, S. M. Volcanism and the Greenland ice-cores: the tephra record. *Earth-Sci Rev* 115, 173-191, doi:10.1016/j.earscirev.2012.09.001 (2012).

Baldini, L. M. et al. Regional temperature, atmospheric circulation, and sea-ice variability within the Younger Dryas Event constrained using a speleothem from northern Iberia. *Earth Planet Sc Lett* 419, 101-110, doi:10.1016/j.epsl.2015.03.015 (2015a).

Baldini, J. U. L., Brown, R. J. & McElwaine, J. N. Was millennial scale climate change during the Last Glacial triggered by explosive volcanism? *Sci Rep-Uk* 5, doi:ARTN 17442 10.1038/srep17442 (2015b).

Chen, T. Y. et al. Synchronous centennial abrupt events in the ocean and atmosphere during the last deglaciation. *Science* 349, 1537-1541, doi:10.1126/science.aac6159 (2015).

Dethloff, K. et al. The impact of Greenland's deglaciation on the Arctic circulation. *Geophys Res Lett* 31, doi:Artn L19201 10.1029/2004gl020714 (2004).

Fyfe, J. C., K. von Salzen, N. P. Gillett, V. K. Arora, G. M. Flato, and J. R. McConnell (2013), One hundred years of Arctic surface temperature variation due to anthropogenic influence, *Sci Rep-Uk*, 3, doi:Artn 2645 Doi 10.1038/Srep02645.

Hansen, J., and L. Nazarenko (2004), Soot climate forcing via snow and ice albedos, *P Natl Acad Sci USA*, 101(2), 423-428, doi:10.1073/pnas.2237157100.

- Haywood, J. M., A. Jones, N. Bellouin, and D. Stephenson (2013), Asymmetric forcing from stratospheric aerosols impacts Sahelian rainfall, *Nat Clim Change*, 3(7), 660-665, doi:10.1038/Nclimate1857.
- Huybers, P., and C. Langmuir (2009), Feedback between deglaciation, volcanism, and atmospheric CO₂, *Earth Planet Sc Lett*, 286(3-4), 479-491, doi:10.1016/j.epsl.2009.07.014.
- Le Roy, M. et al. Calendar-dated glacier variations in the western European Alps during the Neoglacial: the Mer de Glace record, Mont Blanc massif. *Quaternary Sci Rev* 108, 1-22, doi:10.1016/j.quascirev.2014.10.033 (2015).
- Miller, G. H. et al. Abrupt onset of the Little Ice Age triggered by volcanism and sustained by sea-ice/ocean feedbacks. *Geophys Res Lett* 39, doi:Artn L02708 10.1029/2011gl050168 (2012).
- Oman, L., A. Robock, G. Stenchikov, G. A. Schmidt, and R. Ruedy (2005), Climatic response to high-latitude volcanic eruptions, *J Geophys Res-Atmos*, 110(D13), doi:Artn D13103 Doi 10.1029/2004jd005487.
- Oman, L., A. Robock, G. L. Stenchikov, and T. Thordarson (2006), High-latitude eruptions cast shadow over the African monsoon and the flow of the Nile, *Geophys Res Lett*, 33(18), doi:Artn L18711.
- Ridley, H. E., et al. (2015), Aerosol forcing of the position of the intertropical convergence zone since AD 1550, *Nat Geosci*, 8(3), 195-200, doi:10.1038/Ngeo2353.
- Seierstad, I. K. et al. Consistently dated records from the Greenland GRIP, GISP2 and NGRIP ice cores for the past 104 ka reveal regional millennial-scale delta O-18 gradients with possible Heinrich event imprint. *Quaternary Sci Rev* 106, 29-46, doi:10.1016/j.quascirev.2014.10.032 (2014).
- Solomina, O. N. et al. Holocene glacier fluctuations. *Quaternary Sci Rev* 111, 9-34, doi:10.1016/j.quascirev.2014.11.018 (2015).
- Steffensen, J. P. et al. High-resolution Greenland Ice Core data show abrupt climate change happens in few years. *Science* 321, 680-684, doi:DOI 10.1126/science.1157707 (2008).
- Svensson, A., et al. (2013), Direct linking of Greenland and Antarctic ice cores at the Toba eruption (74 ka BP), *Clim Past*, 9(2), 749-766, doi:10.5194/cp-9-749-2013.
- Ullman, D. J., LeGrande, A. N., Carlson, A. E., Anslow, F. S. & Licciardi, J. M. Assessing the impact of Laurentide Ice Sheet topography on glacial climate. *Clim Past* 10, 487-507, doi:10.5194/cp-10-487-2014 (2014).
- Stoffel, M., et al. (2015), Estimates of volcanic-induced cooling in the Northern Hemisphere over the past 1,500 years, *Nat Geosci*, doi:10.1038/Ngeo2526.
- Zielinski, G. A., P. A. Mayewski, L. D. Meeker, S. Whitlow, and M. S. Twickler (1996), A 110,000-yr record of explosive volcanism from the GISP2 (Greenland) ice core, *Quaternary Res*, 45(2), 109-118, doi:DOI 10.1006/qres.1996.0013.

Reviewer #2 (Remarks to the Author):

A. To my knowledge this is the first time a link has been found between volcanic activity and ancient ice sheet mass balance. This paper for the first time takes us out of the theoretical realm of what we infer should happen to Pleistocene ice sheets to what actually happened during eruptive events. It also side steps using modern analogs to infer the behavior of ancient ice sheets that were in a significantly different climatic regime than modern glaciers. The paper makes use of late Holocene examples but only to strengthen the reasoning that eruptions lead to melting events.

Overall the paper makes a strong case for the connection between volcanic eruptions and melting events on the FIS but I am less convinced of the mechanism.

B. This paper will find a broad readership among Quaternary geologists and paleoclimatologists and has some relevance to modern climate change and sea level should a large volcanic eruption occur today. We tend to think of eruptions producing cold events, as in the "Year (1816) without summer" triggered by Tambora but the authors show that there is some evidence to suggest large scale melting events.

C&D. The handling of the varve data seems to be done well. The authors take many overlapping records and use a composite varve record that eliminates local variability. Although not an expected result given the historic record of climate change (1815-1819) in Europe related to the Tambora eruption it does appear that there is a correspondence of melting events and large volcanic events.

My issue with the paper is that I think the authors have overstated the eruptive influences on cloud cover and precipitation on Figure 3. This is not a trivial matter since the simulations are intended to document volcanic influences for two modern analogs in a situation where increased glacial melting is what is being proposed. Here is the way I read Figure 3 vs. the paper text:

1. line 131-133 - The paper claims that over the FIS and Nordic Seas summer cloud cover for a high latitude eruption increased by 15% and precipitation by 30%. This seems to stretch what actually appears on Figure 3. Looking at the figure it would appear to me that in only one small area of the Baltic Sea does an increase in cloud cover approach 15% (orange and yellow colors signify less than 15%) and across much of the FIS and Nordic Seas precipitation increases by substantially less than 30%. It is not clear how the authors arrived at the 15% and 30% values that they portray as the average change for a high altitude eruption.

2. line 133-135 - The paper claims that over the FIS and Nordic Seas summer cloud cover for a tropical eruption increased by 15%. But this looks much less convincing than for the high latitude eruption simulation. This claim very much stretches what appears to be the case on Figure 3. Looking at the figure it would appear to me that white to blue areas would indicate decreases or an unchanged situation and I think a claim of 15% increase is unwarranted.

As a side note I think what happens to west Greenland is impressive (dramatic decrease in cloud cover and precipitation during both high latitude and tropical eruptive events).

E&F. Issues outlined above should be addressed.

G. The paper is clearly and concisely written in a format that is very easy to follow. The capture for figure 4 should say Figure 4 and not Figure 3.

Reviewer #3 (Remarks to the Author):

I recommend this paper be rejected or sent back for major revisions. There are a number of issues with the claims in the paper and the explanations of the relationship they claim between volcanic eruptions and ice sheets. These are all explained below.

1. The fundamental results are presented in Fig. 2, but I do not understand how the correlation between volcanic eruptions and varve thickness is calculated. The figure shows two time series

with spikes, but because they claim simultaneous annual spikes and 900 years are plotted, it is impossible to see whether the thick spikes plotted line up or not. There are circles that are unexplained. They need to provide a list of the actual year of each spike and the ones from each time series that they claim are connected. They also need a better statistical test of whether they correspond or not. The plots at the bottom obscure the data and do not show the actual agreements.

2. The authors do not understand feedbacks and forcing in the climate system. Their statements about them in the abstract and introduction are completely wrong. And those claims are not even discussed in the paper, and have no business being in the abstract.

3. The authors never explain what glacial varves are and what the physical mechanism would be linking volcanic eruptions and thick varves. Are they at the edge of the ice sheet? Do they represent progressive shrinkage of the ice sheet? Why are there only winter and summer layers and not for fall and spring? In lines 140-144 they explain how a volcanic eruption could affect the ice sheet, but what does this have to do with varves?

4. There are serious questions about the timing of the volcanic deposition vs. the varve record. The date of a volcanic sulfate layer is not the same as that of the eruption. It takes 1-2 years for the stratospheric sulfate to make it from the stratosphere to the ice for tropical eruptions. For high-latitude eruptions, the time is shorter, and the fraction of sulfate that ends up there is much larger, and not representative of the climate forcing.

5. The description of the climate model results (lines 131-135) are wrong. There are no significant responses over the FIS region for the tropical simulations, and only cloud but not precipitation responses for the high latitude eruption.

6. The claim that most of the volcanic eruptions during this period were from high-altitude eruptions (I'm not sure what that even means - do they mean high latitude?) has no evidence to support it. Ice cores do not tell you where the eruption was.

7. There are two Figure 3s.

8. The second Fig. 3 (Fig. 4?) shows only one ice core record, yet it is well known that any individual core produces a very noisy volcanic signal. Why not use a record from multiple ice cores, such as Gao et al., Crowley and Unterman, or the new Sigl record, all of which produce a much better Greenland record, as they account for the different deposition patterns and average out the noise?

9. The caption for the second Fig. 3 (Fig. 4?) mixes up SO₂ and SO₄, and does not give the units for SO₄. Furthermore, the estimates of SO₂ emissions from the different eruptions are out-of-date. Use the most recent ice core data, from Sigl et al. In any case, the numbers plotted on the figure do not agree with those in Textor et al., Table 3.

10. The altitude of the Tambora simulations was too low. The tropopause is at about 18 km in the tropics. The emissions should have been above 20 km. Otherwise, most of the SO₂ would have been removed in the troposphere.

11. We know what the climate response to the 1783 Laki and 1815 Tambora eruptions was. The authors need to evaluate these simulations with all the data we have in terms of the regional patterns of temperature and precipitation, as well as global average temperature change so that we have a way of validating the simulations, before we are expected to accept that the regional results presented are accurate.

12. All the figures have issues, as indicated on the attached annotated manuscript, which also has

a number of other comments, all of which need to be addressed.

First, we would like to thank the reviewers for their time and effort in providing us with such a highly detailed, lengthy and constructive review of our paper. Please find below our responses to the points and comments raised by the reviewers.

Reviewer#1

C. Data & methodology: validity of approach, quality of data, quality of presentation

The previously published glacial varve record is of high quality regarding resolution and age-control. The statistically derived coherence between subaerial volcanism and rapid glacier melting itself is intriguing, however, the approach applied for analyzing the potential causal links to past volcanic activity uses some very simplified assumptions that in my view are not substantial enough to derive a causal nexus between volcanism and melting via a positive cloud radiative forcing and ash induced albedo changes on the ice-sheets surface.

To further disentangle potential leads and lags in the relationship between volcanism and ice melting, it would be helpful to take full advantage that both proxy records are based on annually dated chronologies and to also discuss if the number of annual-layers between the potential isochrones were consistent with the respective dating uncertainties of the chronological models.

We agree with Reviewer#1 and we have now provided a detailed analysis that elucidates the time lags between the potential isochrones (new Fig. S2 and Table S1). We note that all the ETVs fall within the sampling uncertainty of the GISP2 sulfate record (3-6 years), even accounting for possible cumulative counting errors. This is now mentioned in the text (new lines 116-119).

The mechanisms suggested to be involved in the positive climate forcing that caused the melting are based strongly on the model output of a single climate-aerosol model, and are shown here only for single month (June). Driven by the aerosol forcing of two of the largest eruptions of known history the simulations are performed under present-day boundary conditions and show only relative moderate cloud cover changes of 15% (predominantly for high-latitude eruptions). Under the assumptions that during deglaciation 1) atmospheric volcanic aerosol content was much larger, 2) volcanic eruptions occurred more frequently and 3) may have involved also deposition of light-absorbing ash on the ice sheet, these model results are extrapolated to deduce an important role of volcanism on contributing to increased ice-sheet melting via a positive cloud and albedo feedback. The validity of these assumptions are, however, not always supported by strong empirical evidence. Questions not sufficiently addressed also include: 1) What is the sensitivity of the simulated cloud cover increases to the choice of the aerosol/climate model and the

respective month? Do other models show comparable results for Scandinavia?

Regarding high-latitude eruptions, which appear to be the one that most affect the cloud cover and also the one that most likely have shown an increase in number during the deglaciation, there is only one simulation available that investigates cloud cover changes (Oman et al., 2006a). In this study the authors also attempt to simulate the Laki eruption using the atmospheric-chemistry GISS model E and find a significant increase in cloudiness over Scandinavia of the order of 2-6% (see Fig. 2a), while no changes in precipitation are simulated. This information has been included in the text (new lines 249-252). Furthermore, we would like to point out that we have considered the entire summer season (June to August) rather than just June.

2) Are simulations under present-day conditions suitable to deduce the climatic response during such a rapidly changing climate state as during the last deglaciation?

This is an important point and we had discussed in the Methods that our simulations are here merely used to investigate potential mechanisms. We had also highlighted that recent studies have demonstrated that large-scale atmospheric circulation during glacial and deglacial conditions was broadly similar to present-day (Löffverström et al., 2016, 2014).

We agree with the reviewer that anyway the climate was rapidly changing at that time, but given the fact that the overall general circulation may have been similar, we expect that our simulations are able to still provide a qualitative indication of circulation changes and hence cloud cover and precipitation associated with the enhanced volcanic eruptions during the deglaciation.

We now discuss these issues in the main text (new lines 154-167).

3) Can you better characterize the volcanic eruptions that correspond to your ETVs with other proxy evidence from ice cores (e.g. tephra)? Are these ash-rich events, high- or low-latitude eruptions? Dynamical and radiative climate effects are quite different for these eruptions.

Thank you for bringing this up. We acknowledge that more work was necessary to characterize the source of the volcanic eruptions recorded in the GISP2 records. We have thoroughly scrutinized the literature and have identified at least two tephra horizons in NGRIP ice cores that can be associated with peaks in volcanic sulfates in GISP2 ice cores. This information is now presented in the new Figure S2 and reported in Table S2. The tephra layers correspond to one large and one medium-sized sulphate peak in GISP2. The first marker is associated with an eruption from the Hekla volcano. The second has an unknown origin but the chemistry clearly argues for an Icelandic origin (Mortensen et al., 2005).

Furthermore, we are aware of an additional tephra horizon that has been recently identified in NEEM ice cores and corresponding to the very large GISP2 sulfate peak observed around 12,550 years BP (Siwan Davies, Swansea University, personal communication). This horizon is associated with another

Icelandic eruption, but regrettably we are not permitted to include this information in the present manuscript, as it is not published yet.

Furthermore, we now provide a detailed comparison between Greenland and Antarctic ice-core records of volcanism for the period under study (new Figure S1). Specifically, we have looked at sulfate and electrical conductivity profiles from EDML, EDC and TD Antarctic ice cores to provide an adequate representation of Southern Hemisphere eruptions. These ice cores provide high-resolution record of past volcanism and are underpinned by exceptionally well-constrained chronologies relative to the Greenland ice-core time scale throughout the later stage of deglaciation. In fact, the Antarctic ice cores were synchronized to Greenland via a number of volcanic tie points (during Early Holocene and HS1), gas stratigraphic markers (methane synchronization) and corroborated by annual-layer counting (Bazin et al., 2013; Veres et al., 2013). The records also offer an adequate spatial coverage of the Antarctic continent to investigate the potential co-occurrence of volcanic events in both hemispheres and thereby allowing identifying tropical eruptions. We conclude that there are no clear signals of Southern Hemisphere eruptions or synchronous events recorded in both Greenlandic and Antarctic ice cores during the interval 13,200-12,300 years BP that would suggest the occurrence of tropical eruptions at this time.

Therefore, based on: 1) the presence of three tephra layers in association with the two largest sulphate peaks and one medium-sized sulfate peak in GISP2 records, and 2) the lack of tropical eruptions inferable from comparing ice cores from both hemispheres, we conclude that the events recorded in the GISP2 volcanic sulfate profile can be primarily ascribed to high-latitude (mainly Icelandic) eruptions.

Unfortunately, only three tephra out of 18 potential ETV-sulfate isochrones have been found in ice cores so far. Though, as also claimed by Abbott and Davies, (2012), the shortage of tephra horizons identified in Greenland during deglaciation does not arise from a lack of eruptions. It is rather an expression of the sampling approach that has been undertaken so far in ice cores, whereby only specific intervals have been investigated. All these points have now been discussed in the main text (new lines 119-122 and new Discussion).

We would also like to point out that early Holocene tephras from Alaskan sources have recently been identified in lake sediment records just south of the Scandinavian ice sheet margin (Muschitiello et al., in preparation) as well as in Svalbard (Willem van der Bilt, University of Bergen, personal communication). This highlights that potentially also ash plumes from North American volcanoes can easily reach as far as Northern Europe.

4) Radiative cooling is among the better understood volcanic effects and is believed to have caused increased glacier mass balances as observed and reconstructed after historic events. Thus your proposed positive forcing needs, in addition, to compensate for these cooling effects in order to contribute significantly to the ice sheet melting.

This is a very good point that we had not addressed in the previous version of the manuscript. We have now included a detailed discussion on the effects of

volcanism on climate cooling with appropriate references and also addressed the counter effect of ice albedo changes (new lines 253-264).

We argue that our results are not at odds with the notion of post-volcanic cooling, but they just indicate that during deglaciation the darkening of the ice from ash may have counteracted and outpaced the cooling effects. Furthermore, the summer solar insolation during the deglaciation was much higher than today ($>50 \text{ W/m}^2$), thus enhancing the albedo feedback. The response of the ice sheet to this albedo effect is probably very rapid and therefore outpaces the effect of changes in accumulation rates. On the other hand, increased precipitation would lead to an increase in ash deposition, which would further decrease the albedo.

Even though some of the precipitation may fall as snow in the ablation zone due to the cooling, large part of it will still be rain, which would act as an additional melting agent through a pathway that is detailed in the text. Finally, we have also highlighted that enhanced glacier melting in conjunction with volcanic eruptions has also been inferred from other records from Alaska during the Bølling interstadial (Praetorius et al., 2016).

D. Conclusions: robustness, validity, reliability. The well-dated high-resolution glacial varve record shows a strong coherence with volcanic aerosol concentrations, suggesting that extreme melting events may be somehow interlinked with the volcanic activity. The datasets used and the analysis performed, however, do not fully support the conclusion that the melting events had been directly forced by volcanic eruptions. On the basis of the presented data it cannot be ruled out that both processes are interlinked via a third process that is driving ice-sheet melting as well as increased volcanism. Alternative causes for rapid changes in climate and atmospheric circulation that seemed to have occurred on very short timescale [Steffensen et al. 2008; Baldini et al., 2015a] should therefore be discussed as well (including for example alternative hypotheses involving volcanic eruptions e.g. Baldini et al., [2015b])

We deem the warming effect of Southern Hemisphere volcanic forcing upon the Northern Hemisphere very unlikely and operating on time scales (centennial/millennial) that are very different from those discussed in our work (annual). However, this issue is worth discussing. We have included in the Discussion (new lines 265-276) a long section where we argue that: 1) there are no clear signs in Antarctic ice cores that would suggest increased volcanic activity in the Southern Hemisphere during the period under consideration (see new Figure S1); 2) there are compelling evidence showing that Northern Europe and particularly southern Sweden experienced a gradual summer cooling starting at the end of GI-1 (ca. 13,200 years BP) which led to the Younger Dryas stadial (Muschitiello et al., 2015; Muschitiello and Wohlfarth, 2015). In particular, regional summer cooling persisted until at least the start of the Holocene. Altogether, we conclude that the ETVs could not have been caused by regional warming, let alone by Southern Hemisphere volcanism.

E. Suggested improvements: experiments, data for possible revision
Any experiment or dataset helping to answer the questions raised under C. will strongly benefit the interpretation of these datasets. With 30% of all

samples in the time window covered by this study labeled as "volcanic" [Zielinski et al., 1996] the coherency analysis would also strongly benefit from a higher resolved and/or more precisely dated volcano record to avoid spurious correlations caused by the low sampling resolution. The NGRIP record appears higher resolved but seems not to capture volcanism during the GS-1. Is there any chance that the new NEEM record (your reference 23) can improve resolution and dating precision of the volcanic sulphate record?

We agree with Reviewer #1 that better resolved records are needed to constrain the timing of volcanic events during deglaciation. However, at present the GISP2 volcanic sulfate record is the only Greenlandic record available of its kind (new lines 97-98). Other records, such as electric conductivity data, can only serve as complementary profiles since conductivity is potentially biased by alkaline dust (Wolff et al., 2005), which is substantially higher during glacials and deglaciations (Ruth et al., 2003). This is the reason why the electrical conductivity data are not discussed and only presented as complementary information.

We have now clarified these points in the main text (new lines 103-106, 599-604) and in the caption of Figure 2.

F. References: appropriate credit to previous work? Previous work is credited, but some references suggesting opposing (i.e. cooling) effects of volcanism or the importance of asymmetric forcing for the dynamical climate response are missing in the discussion (e.g. post volcanic runoff-reduction in Greenland; glacier growth in volcanic active periods; changes in atmospheric circulation after volcanic eruptions and during the Younger Dryas; alternative mechanism suggested to link volcanoes with North Atlantic climate variability).

As mentioned above, we have now more thoroughly discussed the role of volcanic cooling and the related feedbacks on the mass balance of ice sheets and glaciers (new lines 253-255). We have included a number of relevant references and stressed the potential role of volcanism on atmospheric and ocean circulation. We have also introduced the hypothesis by Baldini et al. (2015) and provided an extensive discussion on why an asymmetrical warming mechanism between hemispheres can be ruled out (new lines 265-276).

Additional Comments:

L. 31: "...aerosol particles deposited on snow can alter..."

This has been changed (new lines 34-35).

L. 35: Many citations are given for the various volcanic effects on various climatic parameters; some are redundant e.g. 7,12,15; Citation 9 is showing that volcanic cooling in polar regions was suppressing ice sheet melt after some historic eruptions, which is the opposite of the proposed mechanism in this paper. A negative forcing and growth of ice caps and mountain

glaciers during volcanic active periods is also supported by other studies (see references above). Delete reference with super-eruption (ref. 11) because no such eruption took place during the deglaciation.

References 7, 9, 12 and 15 have been removed from the introduction and are now appropriately cited in the Discussion (new lines 253-255). Reference 11 has been removed.

L 37: Similar simulations from Oman et al. [2005, 2006] using GISS Model E for Katmai and Laki show only minimal changes in cloud cover over Scandinavia (2%) but roughly 1{degree sign}C surface cooling.

We have now mentioned the results from Oman et al. (2006a) in the text. Reviewer #1 will also find in the new version of the manuscript that the cloud radiative effect has been toned down and that our model simulations are only presented as qualitative experiments to test potential volcanism-climate interactions and transient sensitivity (new lines 165-167).

L 37: "Precise synchronization to an annual-layer dated Greenland ice-core chronology"; there are several chronologies available for Greenland ice cores.

This has been changed (new lines 49, 72-73)

L48: Isn't precision more important than accuracy for your type of analysis? The absolute accuracy is of limited importance because you have a common stratigraphic time marker and two annually laminated records.

That is correct. Thank you for pointing this out. This has been changed now (new line 51).

L. 62: What is the age of the Vedde Ash? Please add this information here.

This has been added (new line 74).

L65-66: Given this impressive agreement in independently derived numbers of layers between the two marker horizons I was expecting a more detailed comparison of the volcanic marker events and the proposed candidate melt events taking into account estimated uncertainties of the underlying records respectively (see for example Fig. 7 in Svenssen et al., 2013).

As mentioned earlier, we have now provided a more thorough statistical comparison between the GISP2 and our varve records, which is now summarised in Figure S2 of the Supplementary Material. We have also changed Figure 2 to better highlight the potential isochrones.

L. 71: add time, e.g. older 12,400 years BP.

This has been added (new lines 82-83).

L. 80: Consider replacing the judgment "established record" with something like "...contains an imprint of past volcanic activity" or "can contain the fallout of past volcanic activity".

This has been changed (new lines 91-92).

L. 86: Replace ref. 26 with a more recent publication from Seiderstadt et al., [2014]

The synchronization of GISP2 and NGRIP records throughout GI-1 and GS-1 has not changed from the first study published by Rasmussen et al. (2006) but we have now also cited Seierstad et al. (2014).

L. 90: Define "in phase"! I assume you use the original volcanic attribution from Zielinski et al., [1996]? According to their record roughly 1/3 of their total sulphate samples (12,400-13,200 yr BP) are attributed to volcanic eruptions. Depending on the threshold for a match one always must expect a high number of ETVs occurring in phase. How many SO₄ anomalies are reported for the according time period? If you align and compose the varve segments relative to the volcanic sulphate peaks do you derive a clear melting signal relative to the background?

The terminology has been changed (new lines 113-114). As previously mentioned, we deem the sulphate anomalies to actually under-represent the true volcanic forcing owing to the fact the many eruptions probably occurred sub-glacially and only a portion of the aerosols was released to the atmosphere. We therefore argue that even small-amplitude sulfate anomalies can be associated with volcanic eruptions during deglaciation.

Moreover, we provide a cross-correlation coherency analysis between the ETVs and the GISP2 sulfate data over a window that encompasses more than the maximum cumulative uncertainty predicted by the respective chronologies (Figure S2). The results of this analysis show that the coherency between ETVs and sulphate peaks is only significant for the placement presented in this study. Therefore, despite there might be a high number of ETV-sulfate isochrones for other placements of the varve record on the GICC05 chronology – due to the large number of small sulphate peaks – none of these are statistically significant.

L 94: How do you define the years according to the volcanic events? Year of maximum SO₄ concentration? Year of maximum annual resampled anomalies? Start of increased SO₄ deposition? Some volcanic signals last up to 18 years, and also the one discussed here is dated 12553-12545 yr BP.

The eruption years are defined based on the maximum resampled anomalies. This has now been clarified in the text (new line 336). We now report the age of the volcanic events as well as their varve counterparts for the reader's reference (Table S1).

For this example the proposed forcing occurred 10-20 years later than the proposed ice-sheet response. Is such an offset still compatible with the age uncertainties for the varve record and the ice-core chronology?

Based on a more detailed comparison between ETVs and sulphate peaks (Table S1), we argue that there is hardly any offset between the proposed eruption time and the melt event.

L 96: Can the higher resolved ECM record provide information if two adjacent eruptions contributed to the broad volcanic sulphate signal?

Theoretically yes, but for the reasons explained above (high alkaline dust background damping the volcanic-derived acidity signal) the *ecm* profile should be taken with a pinch of salt. This has been explained in the text (see earlier reply).

L 100-101: Why is that? The ice-core record should record high-latitude events as well as tropical eruptions, but one cannot tell on the basis of GISP2 alone if individual signals are from tropical eruptions or high-latitude events.

We agree with Reviewer#1. This has been removed.

L. 101-110: This is all true, but I cannot see the full relevance of the discussion here. Do you want to imply that the sulphate record from GISP2 is incomplete and may miss some eruptions? Or have false positives, such as SO₄ anomalies that are not of volcanic origin?

The point of this discussion is to warn the reader from all the potential complications that contribute to both the volcanic sulphate signal in the GISP2 records and to our varve reconstruction. This is important, as one could wonder why there is no apparent linear correlation between the amplitude of the sulphate peaks and the thickness of the varve isochrones. We deem this as part of a honest description of the proxy data. However, we leave to the editor deciding if this section should be removed.

L. 109: Which volcanic markers? Tephra for sure, but sulphate is deposited widespread and over longer time and therefore very unlikely to be missing.

We meant tephra. This statement was confusing and we have therefore removed it.

L. 110: "...can even be missing in single/individual ice core records".

Please see previous reply.

L. 114-115: This is likely true. Moreover, the different volcanic effects on

climate discussed earlier also operate on very different temporal scales.

True. We have now mentioned the time scale over which these effects operate (new line 254).

L 118: "Magnitude of atmospheric SO₄ input" is a metric not known. Volcanic and total sulphate concentrations are the only metrics provided for GISP2.

This has been changed (new line 142).

L. 121: The correspondence may also be a consequence that both, volcanism and rapid ice-sheet melting, are both responding to the same events (i.e. rapid surface warming).

As mentioned in a previous reply, we have now provided an extensive discussion whereby we dismiss the potential role of Southern Hemisphere volcanism on Northern Hemisphere warming likewise the role of regional climate warming (new lines 265-276).

L. 130: Are the results of these simulations consistent with those from other models? I believe a large number of other models have performed simulations for this two specific, most recent, extreme events.

Besides the studies of Pausata et al. (2015a, 2015b), there are only a handful of studies investigating the climate impacts of the Laki eruption and high-latitude eruptions in general (Highwood and Stevenson, 2003; Kravitz and Robock, 2011; Oman et al., 2006a, 2006b, 2005; Schmidt et al., 2012), using only two different models: the GISS model and the Reading Intermediate General Circulation Model (IGCM). Similar results have been found by Oman et al. (2006a), who simulated a significant increase in cloud cover over Scandinavia in response to the Laki eruption using the atmospheric-chemistry GISS model E. This information has been included in the text (new lines 249-252). Unfortunately, no information is available regarding IGCM.

L. 131: An increase of 15% in cloud cover does not appear very much. Is this strong enough to cause extreme ice-melting, considering that the 30% increase in precipitation (some of which may be snowfall?) over the ice-sheet will compensate for some of the loss.

We have provided the simulated average cloudiness, precipitation and temperature change over southern Scandinavia for reference (new lines 168-176).

Our data suggest that despite the regional cooling in response to high-latitude eruptions that may have led to increased snowfall, the albedo and probably the cloud/precipitation feedback described in the text must have counterweighed it. Moreover, the increased rainfall over the ablation zones may deposit a large

amount of tephra, thus further increasing the melting. We think that both the increased cloudiness and albedo play together in accelerating the melting of the Fennoscandian Ice Sheet.

Nonetheless, we have now toned down the role of cloud-radiative effects.

L. 138: I would find it more convincing if you could provide (in addition to the MC simulations) for the 16 ETVs and their associated age uncertainties the ages and age uncertainties of the candidate volcanic events. This would allow to addressing if the response is lagging/leading the proposed forcing and if the association is consistent with the individual age models.

As suggested by the reviewer we now provide all the ages of the ETVs and the potential volcanic isochrones in Table S1. We have also estimated the presence of systematic leads/lags between the varve and the volcanic events (see Figure S2). We have found no significant offset between the ETVs and the associated maximum resampled volcanic sulphate peaks. This is now mentioned in the text (new lines 116-119, 226-230).

L. 141-144: These effects need to be able to compensate opposing and relatively well documented and modeled effects such as summer cooling resulting in a positive glacier mass balance [Hanna et al., 2005, your ref. 9; Solomina et al., 2015].

All these factors have been addressed and examined in the discussion (see previous replies). The paper by Solomina et al. (2015) has also been cited.

L. 148: If cloud-induced runoff with respect to clear skies is only +30%, then the runoff increase after a simulated +15% cloud cover would only be in the order of 5% (assuming a linear relationship).

This is correct, but we deem this would still be a significant aspect. In addition, as we mentioned earlier and stated in the text (new lines 165-167), our simulations are only meant to provide clues on the potential mechanisms at play and a real quantification of the cloud-induced runoff is beyond the scope of this study.

L. 154-156: Please specify which kind eruptions (tropical, extratropical, glaciated or non- glaciated areas) experienced an increase. What are the metrics and empirical evidence for the statement that eruptions were "larger"? Reference 35 only discusses frequencies. Figures 2 and 4 also don't allow to drawing conclusions about the magnitudes and trends of past volcanism relative to historic analogues because injection mass and ice concentrations are different parameters (and units).

We agree with Reviewr#1. We have now removed this statement. Figure 2 and Figure 4 have also been changed.

L. 159-161: This comparison is not valid as the data resolution and annual accumulation rates are very different between historic and the late glacial

events. If you were to compare these events you would need to use the same metrics for a comparison (e.g. thinning corrected volcanic sulphate deposition flux).

We now show in Figure 4 the thinning-corrected historical volcanic sulphate deposition data from Sigl et al. (2015).

This also applies to Figure 2. There is no axis given showing the SO₂ injection for the three historic events. It almost appear like the arrows are meant to correspond to the y-axis showing the GISP2 volcanic sulphate concentrations (in ppb), which would be wrong as SO₂ injection does not scale linearly to volcanic sulphate concentrations in the GISP2 ice. It may well be correct that some of the deglacial eruptions injected large amounts of SO₂ causing huge sulphate spikes in GISP2, but this has not yet been demonstrated.

Figure 2 has been changed and the reference to historical eruptions removed.

L. 164-172: The statement that sulphate originating from N-American eruptions may be stronger represented in an ice core from Greenland compared to sulphate from eruptions downwind (e.g. from Iceland), but again the relative sulphate amplitudes of Katmai and St. Helens relative to Laki is not a valid metric to demonstrate this, because a large but unknown fraction of the total sulphate from these recent events is caused by anthropogenic sulphate emissions [Fyfe et al., 2013].

We have addressed this problem by presenting the new volcanic data from Sigl. et al. (2015), which combines the volcanic signal (thinning corrected) recorded in several Greenlandic ice cores. Furthermore, the discussion regarding the sulphate anomalies of the most recent eruptions has been removed. In the light of the more reliable data from Sigl et al. (2015), we have also reconsidered, and ultimately removed from the text, our discussion about the eruption sources (North America versus Iceland) (see new Discussion).

L. 173-178: The intensity estimated as the summed volume of erupted magma over time is strongly biased by few magma producing large eruptions forming vast lava shields in contrast to the fissure eruptions during the Holocene period [Maclennan et al. 2002, your ref. 38]. It is unlikely that the increased intensity of Icelandic volcanism scales in a linear way with the atmospheric aerosol content. Using the GISP2 SO₄ record Zielinski et al. [1996] estimated the frequency of (all) eruptions around 12 ka BP at 20-25 eruptions per ka, which is only 2 to 2.5 times higher than the overall average frequency; and about 4-5 times higher than the Holocene frequency. Given that Laki and Tambora are among the largest events of the past 2,000 years it seems unlikely that the volcanic eruptions between 12-13.2 ka were on average larger than these two extreme events.

We have now removed any references to the magnitude of deglacial eruptions and shifted the focus of our discussion towards the frequency and the sources (see new Discussion). Also, given the comparison between Sigl. et al.'s data and the GISP2 record, we agree with Reviewer#1 that it is unlikely that deglacial eruptions were on average larger than events such Laki and Tambora. However, it should be born in mind that some eruption could be under-represented in GISP2 due to the sub-glacial nature of the Icelandic eruptions.

L. 179-182: Is there any empirical evidence of increased ash generation and widespread deposition during these time periods? Tephrochronologists have been working on these ash layers in Greenland ice cores for years (e.g., Abbott and Davies [2012]). Is there an increase in the frequencies of ash layers during the deglaciation? Or is there any evidence of increased melt in the varve record following the Vedde Ash, for which the spatial extent towards the FIC is well known? Any change of the surface albedo would only result in increased melt during the spring/summer season and only until the ash is buried by subsequent snowfall [Hansen and Nazarenko 2004].

It is hard to say, as the investigation of tephra layers in Greenland ice cores is still a work-in-progress. Abbott and Davies (2012) write: "[...] it must be stressed that the tracing of tephra horizons in the Greenland ice cores has only recently begun and the current status reflect the selective sampling approach adopted".

As yet, only two tephra horizons have been identified during the interval investigated in our study and (Table S2). Notably, these tephra originated from Icelandic volcanoes (Mortensen et al., 2005) and can be in fact assigned to two ETVs observed in our varve records. As also discussed further above, we are aware of at least another Icelandic tephra that can be associated with a large sulphate peak in the GISP2 profile and a large ETV in our record.

Reviewer#1 is right to bring up the Vedde Ash and we confirm that this tephra marker is indeed accompanied by a relatively thick varve layer (MacLeod et al., 2014).

We would also like to stress that we now provide more compelling evidence suggesting that the events recorded in the GISP2 sulfate record are predominantly originated at high northern latitudes (see comparison between Antarctic and Greenlandic ice cores – Figure S1) and primarily from Iceland. In fact a new study has shown a substantial decrease in volcanic activity in North America starting around 13.2-13.1 ka BP (Praetorius et al., 2016). This implies that, given the general paucity of tropical eruptions after ca. 13 ka BP, Iceland must have been the largest contributor in terms of aerosol deposition in GISP2 ice cores.

All these points are now presented in the new Discussion.

L. 183-195: You may want to discuss here also competing hypotheses. For example, Baldini et al., [2015b] recently suggested that asymmetric forcing resulting from Southern hemisphere and Northern hemisphere eruptions changed the ITCZ which may have triggered rapid warming/cooling events in the North Atlantic. While I am not convinced that this hypothesis is strongly supported by their low-resolution geological records, their

analysis highlights at least the complexity of the dynamical climatic response w.r.t. to the location of the eruption which is also reflected by observations and climate models from more recent eruptions [Haywood et al., 2013; Ridley et al., 2015]. On the basis of increased sulphate deposition coeval with melting of the FIS only, the suggested important impact of cloud and ash albedo forcing on the ice sheet finds, in my view, only little support. If the authors could demonstrate that high-latitude eruptions or such eruptions that produced widespread ash deposits were preferentially contributing to these melt-events, these would strengthen the case for a direct causal relationship. With the available data, the statistically significant relationship between volcanic eruptions and melt events may as well be driven by a common third driver, such as rapid temperature rises, increasing ice unloading and subaerial emissions in the volcanic source regions more or less synchronous to the melting events of the FIC.

We have now provided an extensive discussion addressing the hypothesis of Baldini et al. (2015) as well as the possibility of Northern Hemisphere warming (new lines 265-276).

L. 201: Can you give an estimate of the precision of the annual-layer counted chronology over the 1257-year long chronology? The "Maximum Counting Error" for GICC05 for example over this section is increasing from 110 to 145 years suggesting a relative uncertainty of 35 years for the full record. Is the varve record comparable? Or even better?

This information has been included in the text (new lines 307-309) and shown in Figure S2.

L. 216-226: As pointed out earlier, a table with the relative (to Vedde Ash) age uncertainties for the candidate match events of the 16 ETVs for both records would be helpful (in supplement).

We now provide all this information in a supplementary table (Table S1).

L. 248: Which size mode? How different would the results be without the "dust"?

The median radius of the dust in accumulation mode is 0.22 μm (new line 353). We have not performed an experiment without including the dust. However, the dust aerosol is removed very quickly in a matter of a month, therefore it is likely not to have an impact on the simulated climate.

L. 261-269: The statement that large-scale atmospheric circulation should have been similar to modern conditions on the basis of a single regional modeling study from the Laurentide Ice Sheet comes as a surprise. Simulations and proxy records both show the deglaciation had a strong influence on the regional structures and storm tracks of the Arctic and Europe, resulting for example in the migration of the westerly wind position, changes in the sea-ice extent and subsequent atmospheric

reorganization, or changes of the AMOC [Baldini et al., 2015a; Ullman et al., 2014; Dethloff et al., 2004; Steffensen et al., 2008, Chen et al., 2015]. Given the amplitudes and rates of changes occurring during this time period, especially in the North Atlantic sector, the simplified assumption used for the climate-aerosol simulations are unlikely representing realistic boundary conditions.

We agree with Reviewr#1 that the model boundary conditions are not the most appropriate to quantify the climate change induced by volcanic eruptions during the last deglaciation. However, Pausata et al. (2011, 2009) have shown using different climate models that the topography has a dominant role in affecting North Atlantic atmospheric circulation compared to sea surface temperature and sea ice. Furthermore, the impact of a more extensive sea-ice cover during the Younger Dryas may be larger in winter than in summer. Unfortunately, running with different boundary conditions would require the model to be span-up (at least a thousand or more years), and at present we can't afford it. In the revised version of the manuscript we invite the readers to take the model results as qualitatively, stating the limitation of our model design.

Fig. 2: Add separate axis for three historic eruptions. What happened with the "acidity" during GS-1?

Increased dust loading during stadials makes the ice alkaline (Ruth et al., 2003; Wolff et al., 2005), thus damping potential acidity signals in response to volcanism (the dust load increases drastically at the start of the Younger Dryas Stadial – e.g. Fischer et al., 2007). This is a critical issue that prevents us from using the acidity record stand-alone. This is the reason why we focused on the GISP2 sulfate record and only employed the NGRIP2 acidity profile as a complementary record. We have now discussed these complications in the main text, in the caption of Figure 2 and included appropriate references.

Fig. 3: What about July, August? Did you only show June because Laki started in June? If there is no cloud increase in the other summer months, how does the forcing work? One cannot expect that all past eruptions occurred in June.

This has been incorrectly reported in the caption. The estimates do refer to the whole summer period (JJA). We have now clarified this and apologies for the mistake.

Fig. 4: Labeled as Fig. 3; Laki was in 1783 not 1781 Although labeled "volcanic sulphate" by Zielinski et al. [1996], it is clear that the sulphate record after about 1900 is strongly driven by anthropogenic emissions superimposed on the volcanic emissions. Thus a direct comparison of the "volcanic sulphate" levels of GISP2 between Laki and Mt. St. Helens (to some extent also Katmai) will lead to wrong results. With both "upwind" eruptions falling in the 20th century, the interpretation of the dominance of the "upwind" eruption is also not supported by this data.

The data and the figure have been changed. We have also substantially modified the discussion and do not delve in the comparison with eruptions from the 20th Century.

Reviewer #2 (Remarks to the Author):

My issue with the paper is that I think the authors have overstated the eruptive influences on cloud cover and precipitation on Figure 3. This is not a trivial matter since the simulations are intended to document volcanic influences for two modern analogs in a situation where increased glacial melting is what is being proposed. Here is the way I read Figure 3 vs. the paper text:

- 1. line 131-133 - The paper claims that over the FIS and Nordic Seas summer cloud cover for a high latitude eruption increased by 15% and precipitation by 30%. This seems to stretch what actually appears on Figure 3. Looking at the figure it would appear to me that in only one small area of the Baltic Sea does an increase in cloud cover approach 15% (orange and yellow colors signify less than 15%) and across much of the FIS and Nordic Seas precipitation increases by substantially less than 30%. It is not clear how the authors arrived at the 15% and 30% values that they portray as the average change for a high altitude eruption.**
- 2. line 133-135 - The paper claims that over the FIS and Nordic Seas summer cloud cover for a tropical eruption increased by 15%. But this looks much less convincing than for the high latitude eruption simulation. This claim very much stretches what appears to be the case on Figure 3. Looking at the figure it would appear to me that white to blue areas would indicate decreases or an unchanged situation and I think a claim of 15% increase is unwarranted. Issues outlined above should be addressed.**

Thank you for bringing this up. The model results presented in Figure 3 have now been more precisely quantified and provided in details (new lines 168-176). Specifically, we provide averaged values for southern Scandinavia. Even though the changes estimated with our simulations are relatively small, we still deem the figures significant, and particularly as far as the high-latitude eruptions are concerned. Moreover, since we now argue that – based on the lines of evidence provided in the new version of the manuscript – the signals recorded in GISP2 ice cores during the period of study are mainly related to high-latitude volcanoes, the model results associated with tropical eruptions are somewhat of secondary importance.

It should also be noted that, in the light of the new comparative analysis with Antarctic ice cores and the identification of Icelandic tephra, we have now emphasized the role of ash deposition on the FIS at the expenses of the role of other potential feedbacks, such as those highlighted in our simulations (i.e. cloud cover and precipitation changes).

Finally, as also discussed in the replies to Reviewer#1's comments, we would like to stress that the results from our simulations should be considered as a

mean to explore climate sensitivity to volcanic forcing rather than to quantify the actual climate response to changes in volcanic aerosol. This has now been clearly stated (new lines 164-167) and we hope that this stands out in the new version of the manuscript.

The capture for figure 4 should say Figure 4 and not Figure 3.

This has been changed.

Reviewer #3 (Remarks to the Author):

1. The fundamental results are presented in Fig. 2, but I do not understand how the correlation between volcanic eruptions and varve thickness is calculated. The figure shows two time series with spikes, but because they claim simultaneous annual spikes and 900 years are plotted, it is impossible to see whether the thick spikes plotted line up or not. There are circles that are unexplained. They need to provide a list of the actual year of each spike and the ones from each time series that they claim are connected. They also need a better statistical test of whether they correspond or not. The plots at the bottom obscure the data and do not show the actual agreements.

The specifics of the Monte Carlo tests applied here are detailed in the Methods. Figure 2 has been changed to improve (we hope) clarity and new supplementary figures are provided whereby additional statistical analyses are presented (Figure S2).

In Figure 2 there are now bars indicating the isochrones events, and the NGRIP acidity profile has been plotted further down to facilitate comparison with GISP2 records. The results from the two independent Monte Carlo tests used to estimate synchronicity between the varve and the GISP2 records are displayed on the bottom and a detailed explanation is provided in the Methods (new lines 332-339) and in the figure caption. The estimated agreement is also reported in the caption. Figure S2 shows in detail the age differences between the ETVs in our varve record and the potential sulfate peak isochrones in GISP2 ice cores. The total age uncertainties associated with each chronology is also indicated. Moreover, we provide an additional test to visualize the agreement between ETVs and sulphate peaks for different placements of the varve chronology on the GICC05 time scale (Fig. S2e). A new table is also provided (Table S1) whereby we report the ages of each ETV and the age of the corresponding sulfate peak in GISP2. Age uncertainties associated with both records are also reported for reference.

2. The authors do not understand feedbacks and forcing in the climate system. Their statements about them in the abstract and introduction are completely wrong. And those claims are not even discussed in the paper, and have no business being in the abstract.

We rephrased the incipit lines of the abstract.

3. The authors never explain what glacial varves are and what the physical mechanism would be linking volcanic eruptions and thick varves. Are they at the edge of the ice sheet? Do they represent progressive shrinkage of the ice sheet? Why are there only winter and summer layers and not for fall and spring? In lines 140-144 they explain how a volcanic eruption could affect the ice sheet, but what does this have to do with varves?

Thank you for bringing this up. We agree with Reviewer#3 that a clear explanation of the glacial varve was missing and we apologize for this shortcoming. We have now better introduced the proxy, described in details how varves are formed and their relationship to ice-sheet melting (new lines 56-71). We have also included a new figure (Figure 5), which illustrates the depositional environment and formation of glacial-clay varves, as well as the mechanisms that link volcanisms to enhanced subglacial sediment discharge (and therefore varve thickness). An additional detailed explanation of glacial-clay varve formation is also provided in the caption.

4. There are serious questions about the timing of the volcanic deposition vs. the varve record. The date of a volcanic sulfate layer is not the same as that of the eruption. It takes 1-2 years for the stratospheric sulfate to make it from the stratosphere to the ice for tropical eruptions. For high-latitude eruptions, the time is shorter, and the fraction of sulfate that ends up there is much larger, and not representative of the climate forcing.

We have discussed these issues in the main text (new lines 221-230). In particular, as we show and argue that the majority of the eruptions that occurred during the period investigated here originated from high-latitudes, it is likely that any potential lead/lag between tephra and sulphate deposition for any given eruption was not longer than one year (Abbott and Davies, 2012). First, assuming that the varve records capture a signal associated with darkening of the ice due to ash deposition, we could not find any systematic lead/lag with respect to the sulphate isochrones in GISP2 (Fig. S2). Second, as we now explain in the text (new lines 231-232), it is difficult here to assess the potential offsets between sulfate deposition in Greenland and ash deposition on the Scandinavian Ice Sheet (or initiation of the cloud feedback) due to resolution limitations with the GISP2 records.

5. The description of the climate model results (lines 131-135) are wrong. There are no significant responses over the FIS region for the tropical simulations, and only cloud but not precipitation responses for the high latitude eruption.

Please see replies to Reviewer#1 and #2. This issue has been addressed. We have provided more rigorous values for the simulated climate responses in southern Scandinavia (new lines 168-176). It is correct that there is no significant cloud and precipitation response to tropical eruptions. Given the new

results and evidence presented in the revised version of the manuscript (ice core comparison, tephra identification, etc.), we do not focus our attention on the tropical volcanism. Finally our model simulations only aim at providing a qualitative insight on potential causes that could play together with the albedo feedback in enhancing FIS melting.

6. The claim that most of the volcanic eruptions during this period were from high-altitude eruptions (I'm not sure what that even means - do they mean high latitude?) has no evidence to support it. Ice cores do not tell you where the eruption was.

We apologize for this. It was a mistake due to the auto spelling checking. This has now been corrected (“high-latitude”). We have also provided further evidence showing that the volcanic eruptions were largely originated from high-latitude.

7. There are two Figure 3s.

This has now been changed.

8. The second Fig. 3 (Fig. 4?) shows only one ice core record, yet it is well known that any individual core produces a very noisy volcanic signal. Why not use a record from multiple ice cores, such as Gao et al., Crowley and Unterman, or the new Sigl record, all of which produce a much better Greenland record, as they account for the different deposition patterns and average out the noise?

Thank you for this comment. We agree with Reviewer#3 and now present in Figure 4, as suggested, volcanic sulfate flux data from Sigl et al. (2015) and global stratospheric volcanic sulfate injections from Gao et al. (2008).

9. The caption for the second Fig. 3 (Fig. 4?) mixes up SO₂ and SO₄, and does not give the units for SO₄. Furthermore, the estimates of SO₂ emissions from the different eruptions are out-of-date. Use the most recent ice core data, from Sigl et al. In any case, the numbers plotted on the figure do not agree with those in Textor et al., Table 3.

The figure has been changed according to the new suggested data.

10. The altitude of the Tambora simulations was too low. The tropopause is at about 18 km in the tropics. The emissions should have been above 20 km. Otherwise, most of the SO₂ would have been removed in the troposphere.

We agree that the Tambora eruption is set too low compared to reality. The reason is because the model has few layers in the stratosphere and therefore, injecting SO₂ well above 20 km would make the SO₄ residence time unrealistic. We had chosen the starting altitude of 18 km, using the Pinatubo eruption as testing case. We aimed at reproducing a similar SO₄ peak concentration and e-folding time compared to observation. By injecting the SO₂ mostly between 15

and 21 km, the model shows a SO₄ peak of ~22 Tg and an e-folding time of ~17 months (Figure S3-S5 – now also included in the Supplementary Material). Observational evidence indicates that between 21 and 40 Tg of sulfate aerosol was produced (Russell et al., 1996) and the e-folding time was between 12-14 months (Baran and Foot, 1994; Barnes and Hofmann, 1997; Lambert et al., 1993). Therefore, our model output fall in the lower bound of the observations for the SO₄ peak and slightly underestimate the SO₄ removal. The simulated global cooling is around 0.4°C, in agreement with other modeling studies and observations, i.e. 0.4-0.5°C (Kirchner et al., 1999). Therefore, we have adopted the same injection height for the Tambora experiment as the Pinatubo. This explanation has now been included in the Methods (new lines 360-371).

11. We know what the climate response to the 1783 Laki and 1815 Tambora eruptions was. The authors need to evaluate these simulations with all the data we have in terms of the regional patterns of temperature and precipitation, as well as global average temperature change so that we have a way of validating the simulations, before we are expected to accept that the regional results presented are accurate.

The performance of NorESM in simulating the Laki eruption has been extensively investigated in Pausata et al., 2015. Regarding the Tambora, we have now included in the Supplementary Material a validation of the model performance in reproducing the Pinatubo eruption as described above, because only few climate data were available and they were mostly confined to Europe and North America for the Tambora eruption. Overall, our model is able to reproduce the main feature of the global impacts associated to both high-latitude and tropical eruptions.

12. All the figures have issues, as indicated on the attached annotated manuscript, which also has a number of other comments, all of which need to be addressed.

These issues have been addressed.

References

- Abbott, P.M., Davies, S.M., 2012. Volcanism and the Greenland ice-cores: The tephra record. *Earth-Science Reviews*. doi:10.1016/j.earscirev.2012.09.001
- Baldini, J.U.L., Brown, R.J., McElwaine, J.N., 2015. Was millennial scale climate change during the Last Glacial triggered by explosive volcanism? *Scientific reports* 5, 17442. doi:10.1038/srep17442
- Baran, a. J., Foot, J.S., 1994. New application of the operational sounder HIRS in determining a climatology of sulphuric acid aerosol from the Pinatubo eruption. *Journal of Geophysical Research* 99, 25673. doi:10.1029/94JD02044
- Barnes, J.E., Hofmann, D.J., 1997. Lidar measurements of stratospheric aerosol over Mauna Loa Observatory. *Geophysical Research Letters* 24, 1923–1926.

doi:10.1029/97GL01943

- Bazin, L., Landais, A., Lemieux-Dudon, B., Toyé Mahamadou Kele, H., Veres, D., Parrenin, F., Martinerie, P., Ritz, C., Capron, E., Lipenkov, V., Loutre, M.F., Raynaud, D., Vinther, B., Svensson, A., Rasmussen, S.O., Severi, M., Blunier, T., Leuenberger, M., Fischer, H., Masson-Delmotte, V., Chappellaz, J., Wolff, E., 2013. An optimized multi-proxy, multi-site Antarctic ice and gas orbital chronology (AICC2012): 120-800 ka. *Climate of the Past* 9, 1715–1731. doi:10.5194/cp-9-1715-2013
- Fischer, H., Siggaard-Andersen, M.L., Ruth, U., Röthlisberger, R., Wolff, E.W., 2007. Glacial / Interglacial Changes in Mineral Dust and Sea-Salt Records in Polar Ice Cores : Sources , Transport , and Deposition. *Reviews of Geophysics* 45, 1–26. doi:10.1029/2005RG000192.1.INTRODUCTION
- Gao, C., Robock, A., Ammann, C., 2008. Volcanic forcing of climate over the past 1500 years: An improved ice core-based index for climate models. *Journal of Geophysical Research Atmospheres* 113. doi:10.1029/2008JD010239
- Highwood, E.J., Stevenson, D.S., 2003. Atmospheric impact of the 1783–1784 Laki Eruption: Part II Climatic effect of sulphate aerosol. *Atmospheric Chemistry and Physics Discussions* 3, 1599–1629. doi:10.5194/acpd-3-1599-2003
- Kirchner, I., Stenchikov, G.L., Graf, H.-F., Robock, A., Antuña, J.C., 1999. Climate model simulation of winter warming and summer cooling following the 1991 Mount Pinatubo volcanic eruption. *Journal of Geophysical Research* 104, 19039. doi:10.1029/1999JD900213
- Kravitz, B., Robock, A., 2011. Climate effects of high-latitude volcanic eruptions: Role of the time of year. *Journal of Geophysical Research Atmospheres* 116. doi:10.1029/2010JD014448
- Lambert, A., Grainger, R., Remedios, J., Rodgers, C., Corney, M., Taylor, F., 1993. Measurements of the Evolution of the Mount Pinatubo Aerosol Cloud by ISAMS. *Geophys. Res. Lett.* 20, 1287–1290.
- Löfverström, M., Caballero, R., Nilsson, J., Kleman, J., 2014. Evolution of the large-scale atmospheric circulation in response to changing ice sheets over the last glacial cycle. *Clim. Past* 10, 1453–1471. doi:10.5194/cp-10-1453-2014
- Löfverström, M., Caballero, R., Nilsson, J., Messori, G., 2016. Stationary Wave Reflection as a Mechanism for Zonalizing the Atlantic Winter Jet at the LGM. *Journal of the Atmospheric Sciences* 73, 3329–3342.
- Macleod, a., Brunnberg, L., Wastegård, S., Hang, T., Matthews, I.P., 2014. Lateglacial cryptotephra detected within clay varves in Östergötland, south-east Sweden. *Journal of Quaternary Science* 29, 605–609. doi:10.1002/jqs.2738
- Mortensen, A.K., Bigler, M., Grönvold, K., Steffensen, J.P., Johnsen, S.J., 2005. Volcanic ash layers from the last glacial termination in the NGRIP ice core. *Journal of Quaternary Science* 20, 209–219. doi:10.1002/jqs.908
- Muschitiello, F., Pausata, F.S.R., Watson, J.E., Smittenberg, R.H., Salih, A.A.M., Brooks, S.J., Whitehouse, N.J., Karlatou-Charalampopoulou, A., Wohlfarth, B., 2015. Fennoscandian freshwater control on Greenland hydroclimate shifts at the onset of the Younger Dryas. *Nature Communications* 6, 1–8. doi:10.1038/ncomms9939
- Muschitiello, F., Wohlfarth, B., 2015. Time-transgressive environmental shifts across Northern Europe at the onset of the Younger Dryas. *Quaternary*

- Science Reviews 109, 49–56. doi:10.1016/j.quascirev.2014.11.015
- Oman, L., Robock, A., Stenchikov, G., Schmidt, G.A., Ruedy, R., 2005. Climatic response to high-latitude volcanic eruptions. *Journal of Geophysical Research Atmospheres* 110. doi:10.1029/2004JD005487
- Oman, L., Robock, A., Stenchikov, G.L., Thordarson, T., 2006a. High-latitude eruptions cast shadow over the African monsoon and the flow of the Nile. *Geophysical Research Letters* 33. doi:10.1029/2006GL027665
- Oman, L., Robock, A., Stenchikov, G.L., Thordarson, T., Koch, D., Shindell, D.T., Gao, C., 2006b. Modeling the distribution of the volcanic aerosol cloud from the 1783-1784 Laki eruption. *Journal of Geophysical Research Atmospheres* 111. doi:10.1029/2005JD006899
- Pausata, F.S.R., Chafik, L., Caballero, R., Battisti, D.S., 2015a. Impacts of high-latitude volcanic eruptions on ENSO and AMOC. *Proceedings of the National Academy of Sciences* 112, 201509153. doi:10.1073/pnas.1509153112
- Pausata, F.S.R., Grini, A., Caballero, R., Hannachi, A., Seland, Ø., 2015b. High-latitude volcanic eruptions in the Norwegian Earth System Model: the effect of different initial conditions and of the ensemble size. *Tellus B* 67. doi:10.3402/tellusb.v67.26728
- Pausata, F.S.R., Li, C., Wettstein, J., Kageyama, M., Nisancioglu, K.H., 2011. The key role of topography in altering North Atlantic atmospheric circulation during the last glacial period. *Past climate variability: model analysis and proxy intercomparison* 7, 1089–1101.
- Pausata, F.S.R., Li, C., Wettstein, J.J., Nisancioglu, K.H., Battisti, D.S., 2009. Changes in atmospheric variability in a glacial climate and the impacts on proxy data: a model intercomparison. *Climate of the Past* 5, 489–502.
- Praetorius, S., Mix, A., Jensen, B., Froese, D., Milne, G., Wolhowe, M., Addison, J., Prah, F., 2016. Interaction between climate, volcanism, and isostatic rebound in Southeast Alaska during the last deglaciation. *Earth and Planetary Science Letters* 452, 79–89.
- Rasmussen, S.O., Andersen, K.K., Svensson, A.M., Steffensen, J.P., Vinther, B.M., Clausen, H.B., Siggaard-Andersen, M.L., Johnsen, S.J., Larsen, L.B., Dahl-Jensen, D., Bigler, M., Röthlisberger, R., Fischer, H., Goto-Azuma, K., Hansson, M.E., Ruth, U., 2006. A new Greenland ice core chronology for the last glacial termination. *Journal of Geophysical Research: Atmospheres* 111. doi:10.1029/2005JD006079
- Russell, P.B., Livingston, J.M., Pueschel, R.F., Bauman, J.J., Pollack, J.B., Brooks, S.L., Hamill, P., Thomason, L.W., Stowe, L.L., Deshler, T., 1996. Global to microscale evolution of the Pinatubo volcanic aerosol derived from diverse measurements and analyses. *Journal of Geophysical Research: Atmospheres* 101, 18745–18763.
- Ruth, U., Wagenbach, D., Steffensen, J.P., Bigler, M., 2003. Continuous record of microparticle concentration and size distribution in the central Greenland NGRIP ice core during the last glacial period. *Journal of Geophysical Research* 108, 1–12. doi:10.1029/2002JD002376
- Schmidt, A., Thordarson, T., Oman, L.D., Robock, A., Self, S., 2012. Climatic impact of the long-lasting 1783 Laki eruption: inapplicability of mass-independent sulfur isotopic composition measurements. *Journal of Geophysical Research: Atmospheres* 117.
- Seierstad, I.K., Abbott, P.M., Bigler, M., Blunier, T., Bourne, A.J., Brook, E.,

- Buchardt, S.L., Buizert, C., Clausen, H.B., Cook, E., Dahl-Jensen, D., Davies, S.M., Guillevic, M., Johnsen, S.J., Pedersen, D.S., Popp, T.J., Rasmussen, S.O., Severinghaus, J.P., Svensson, A., Vinther, B.M., 2014. Consistently dated records from the Greenland GRIP, GISP2 and NGRIP ice cores for the past 104ka reveal regional millennial-scale $\delta^{18}O$ gradients with possible Heinrich event imprint. *Quaternary Science Reviews* 106, 29–46. doi:10.1016/j.quascirev.2014.10.032
- Sigl, M., Winstrup, M., McConnell, J.R., Welten, K.C., Plunkett, G., Ludlow, F., Büntgen, U., Caffee, M., Chellman, N., Dahl-Jensen, D., Fischer, H., Kipfstuhl, S., Kostick, C., Maselli, O.J., Mekhaldi, F., Mulvaney, R., Muscheler, R., Pasteris, D.R., Pilcher, J.R., Salzer, M., Schüpbach, S., Steffensen, J.P., Vinther, B.M., Woodruff, T.E., 2015. Timing and climate forcing of volcanic eruptions for the past 2,500 years. *Nature* 523, 543–549. doi:10.1038/nature14565
- Solomina, O.N., Bradley, R.S., Hodgson, D.A., Ivy-Ochs, S., Jomelli, V., Mackintosh, A.N., Nesje, A., Owen, L.A., Wanner, H., Wiles, G.C., Young, N.E., 2015. Holocene glacier fluctuations. *Quaternary Science Reviews*. doi:10.1016/j.quascirev.2014.11.018
- Veres, D., Bazin, L., Landais, A., Toyé Mahamadou Kele, H., Lemieux-Dudon, B., Parrenin, F., Martinerie, P., Blayo, E., Blunier, T., Capron, E., Chappellaz, J., Rasmussen, S.O., Severi, M., Svensson, A., Vinther, B., Wolff, E.W., 2013. The Antarctic ice core chronology (AICC2012): An optimized multi-parameter and multi-site dating approach for the last 120 thousand years. *Climate of the Past* 9, 1733–1748. doi:10.5194/cp-9-1733-2013
- Wolff, E.W., Cook, E., Barnes, P.R.F., Mulvaney, R., 2005. Signal variability in replicate ice cores. *Journal of Glaciology* 51, 462–468. doi:10.3189/172756505781829197

Reviewers' comments:

Reviewer #1 (Remarks to the Author):

I reviewed the revised manuscript and read the comments of all reviewers and the authors' responses. In the revised version, the authors have addressed most of the raised concerns. Overall, they provided convincingly evidence that rapid melting events indicated by ETVs in the foreland of the margin of the Fennoscandian ice sheet were somehow associated with sub-aerial volcanism, with little chance that this relation is caused by pure chance. This is, to my knowledge, the first time that two very different proxy records during the last glacial have been compared to each other at an annual resolution, thus taking full advantage of high dating precision of these records. Using in addition published tephra studies from Greenland ice cores and ice core records from Antarctica (which I comment on in more details below) the authors now regard especially high-latitude eruptions (most likely from Iceland) as the main contributors to increased volcanic activity, which I agree on. Regarding the potential mechanisms involved, the authors suggest two possibilities how volcanism could trigger enhanced ice-sheet melting: 1) Changes of cloud-cover caused by sulphate aerosol injection and 2) increased ice-albedo due to deposition and accumulation of light absorbing dark volcanic ash. While it is difficult to assess and quantify the relative importance and magnitudes of these proposed effects, the authors do present modern analogues and preliminary climate model results that suggest that these effects appear to be at least plausible in explaining the observed association. More studies will certainly be necessary to further investigate such linkages in the future (including potentially more dedicated modeling studies), but on the basis of the exceptional strong associations drawn from the high-res. observational records at hand at the moment, the authors have, in my view, done the best possible analysis.

Below I added a few specific comments:

L. 27: According to your Fig. 3, the area of increased cloud cover following Laki type eruptions is also covering Scandinavia, not only the Atlantic.

L. 104: You may want to include that ECM record has much higher resolution than the GISP2 record, otherwise people may (falsely) believe the GISP2 record is an exceptionally good ice-core record. It is not, at least compared to present day standards. It is just the only record available for Greenland.

L. 193: Please provide the reference for the "50x more frequent Icelandic eruptions". I believe it is 41 not 40. If a factor 50 was indeed real, than I assume the majority of this increase must come from small magnitude eruptions as such a large increase is not supported by the GISP2 ice core record (see my earlier comment on your original submission).

L. 202: Better weaken down the statement such as "...no clear signs of a number of large-magnitude tropical volcanic eruptions over most of the time" or a "general paucity of strong and frequent activity in the Southern Hemisphere". The volcanic synchronization performed previously (Veres et al., 2013) was stopped at a GICC05 age of 11,518 yr BP. Below that, the stratigraphic matching is based on CH4 with a relative age uncertainty of ± 300 years. This means that AICC2012 and GICC05 are NOT synchronized on annual/decadal timescales that would allow you to attribute volcanic signals from Antarctica to the Greenland record (your grey bars). It is also very unlikely that over 900 years there would have been no tropical eruptions at all, when about 80 occurred over the past 2,500 years (Sigl et al., 2015). Still, you may be able to argue for an absence of many large tropical volcanoes on the basis of the absolute number of events over the entire 900 year window for Greenland vs. Antarctica, respectively. Severi et al. (2007) identified only three common events in EDML and EDC, whereas Rasmussen et al. (2013) identified 17 common events between NGRIP2 and NEEM. But I would not go down the path to claim that the three large signals in Antarctica did not have a Greenland counterpart.

L. 220-221: The really widespread dispersal of crypto-tephra for so many events is a relatively new observation that also benefited from new approaches how ice-cores are now analyzed. If you had space you could for example move the review paper (reference 38) up to the traditional approach (line 221) and add newer references for crypto-tephra that were detected with using the

new sampling approaches (Jensen et al., 2014; Ponomareva et al., 2015; Sun et al., 2014). As you already mentioned in your reply regarding Siwan Davies personal communication and your on manuscript on North American tephra in Scandinavia such events are much more common than is currently anticipated. These published papers will also provide evidence that tephra dispersion can be widespread even with moderate SO₂ injections.

L 249/251: Why would the frequency of eruptions impact the magnitude of response for a single event? To you suggest a sustained impact from multiple eruptions occurring closely in time? Or do you suggest that also the magnitude of last glacial eruptions may have been higher. For both there is some evidence in the proxy records. I realized you downplayed the potential role that eruptions were larger during the late Glacial partly in response to my earlier critique. I did not intend to deny this aspect completely, but mostly wanted to criticize that some of the conclusions were drawn by comparing the ice-core concentrations directly with modern events. You could write "...overall increase in frequencies and magnitudes (Maclennan et al., 2002; Thordarson and Hoskuldsson, 2008)" as the geological records do indicate some very large volcanic eruptions.

L. 265: higher rates of tephra deposition

L. 365: typo: eruption

References: Correct redundant citations!

Fig. S1:

I would remove the grey vertical bars and use the higher abundances of Greenland volcanic events relative to Antarctica as an argument in support of high volcanic activity in NH high-latitudes. On which ice core is the number of annual layers plus uncertainty based? EDML? To my knowledge the annual-layer dating of EDML has not (yet?) been published in a peer-reviewed journal, and the absolute age of the CH₄ based tie-point is rather poorly constrained.

Readability and figure captions for all figures have strongly improved.

References

- Jensen, B. J. L., Pyne-O'Donnell, S., Plunkett, G., Froese, D. G., Hughes, P. D. M., Sigl, M., McConnell, J. R., Amesbury, M. J., Blackwell, P. G., van den Bogaard, C., Buck, C. E., Charman, D. J., Clague, J. J., Hall, V. A., Koch, J., Mackay, H., Mallon, G., McColl, L., and Pilcher, J. R.: Transatlantic distribution of the Alaskan White River Ash, *Geology*, 42, 875-878, 2014.
- Maclennan, J., Jull, M., McKenzie, D., Slater, L., and Gronvold, K.: The link between volcanism and deglaciation in Iceland, *Geochem Geophys Geosy*, 3, 2002.
- Ponomareva, V., Portnyagin, M., and Davies, S. M.: Tephra without Borders: Far-Reaching Clues into Past Explosive Eruptions, *Frontiers in Earth Science*, 3, 2015.
- Rasmussen, S. O., Abbott, P. M., Blunier, T., Bourne, A. J., Brook, E., Buchardt, S. L., Buizert, C., Chappellaz, J., Clausen, H. B., Cook, E., Dahl-Jensen, D., Davies, S. M., Guillevic, M., Kipfstuhl, S., Laepple, T., Seierstad, I. K., Severinghaus, J. P., Steffensen, J. P., Stowasser, C., Svensson, A., Vallenga, P., Vinther, B. M., Wilhelms, F., and Winstrup, M.: A first chronology for the North Greenland Eemian Ice Drilling (NEEM) ice core, *Clim Past*, 9, 2713-2730, 2013.
- Severi, M., Becagli, S., Castellano, E., Morganti, A., Traversi, R., Udasti, R., Ruth, U., Fischer, H., Huybrechts, P., Wolff, E., Parrenin, F., Kaufmann, P., Lambert, F., and Steffensen, J. P.: Synchronisation of the EDML and EDC ice cores for the last 52 kyr by volcanic signature matching, *Clim Past*, 3, 367-374, 2007.
- Sigl, M., Winstrup, M., McConnell, J. R., Welten, K. C., Plunkett, G., Ludlow, F., Buntgen, U., Caffee, M., Chellman, N., Dahl-Jensen, D., Fischer, H., Kipfstuhl, S., Kostick, C., Maselli, O. J., Mekhaldi, F., Mulvaney, R., Muscheler, R., Pasteris, D. R., Pilcher, J. R., Salzer, M., Schupbach, S., Steffensen, J. P., Vinther, B. M., and Woodruff, T. E.: Timing and climate forcing of volcanic eruptions for the past 2,500 years, *Nature*, 523, 543-+, 2015.
- Sun, C. Q., Plunkett, G., Liu, J. Q., Zhao, H. L., Sigl, M., McConnell, J. R., Pilcher, J. R., Vinther, B., Steffensen, J. P., and Hall, V.: Ash from Changbaishan Millennium eruption recorded in Greenland ice: Implications for determining the eruption's timing and impact, *Geophys Res Lett*, 41, 694-701, 2014.
- Thordarson, T. and Hoskuldsson, A.: Postglacial volcanism in Iceland, *Jokull*, 58, 197-228, 2008.
- Veres, D., Bazin, L., Landais, A., Kele, H. T. M., Lemieux-Dudon, B., Parrenin, F., Martinerie, P.,

Blayo, E., Blunier, T., Capron, E., Chappellaz, J., Rasmussen, S. O., Severi, M., Svensson, A., Vinther, B., and Wolff, E. W.: The Antarctic ice core chronology (AICC2012): an optimized multi-parameter and multi-site dating approach for the last 120 thousand years, *Clim Past*, 9, 1733-1748, 2013.

Reviewer #2 (Remarks to the Author):

My previous concerns about Figure 3 have been addressed in the revised text. I read the full paper again and found it to be better supported and also much easier to read with new data and references. I did find a few typographical errors that I list below.

I strongly recommend publishing this paper after seeing the revised version.

Line 47 – “annual glacial varves” is redundant. All varves are annual sediment layers by definition and there is no such thing as a non-annual varve. You could say just “glacial varves”.

Line 183 – The citation of Fig. 4 here seems unnecessary since it is given in line 182.

Line 202 – “On the contrary” starts a line that is not contrary to the previous line in the text and seems to support it if anything.

Line 234 – “mechanism” should be plural – “mechanisms”

Line 270 – “northern should be capitalized as “Northern”

Line 352 – the reference numbers 41 and 42 are incorrect and should be 56 and 57. Shouldn't the author references be removed and replaced with the citation numbers?

Line 602 – “ions” should be “ion”

Lines 609 and 610 – the references to (d) and (e) should be (e) and (f).

Lines 614 to 620 – Starting with “Grey bars indicate” to the end on line 620 should be associated with the entry for part d of the figure and not that for e-f.

Line 619 – I suggest that “displayed” should be changed to “labeled” to make it clear that you are not indicating a different form of data but only labels on the graph.

Reviewer #3 (Remarks to the Author):

Review of Muschitiello et al. revised

I still recommend that this paper be rejected. The hypothesis that large high-latitude volcanic eruptions cause excessive melting of the Fennoscandian ice sheet is not supported by the evidence.

1. Because of the natural variability of sulfate deposition on ice cores, as acknowledged by the authors in the caption for Fig. 2, volcanic time series produced from the past 1000 years typically use 20 or more ice cores, to account for this noise. Any one individual core can completely miss large volcanic eruptions, or have amplified deposition because of weather.

2. One climate model is inadequate for simulating cloud response to volcanic eruptions. Cloud responses are very noisy, and each model is different. And the model use has not been validated adequately with recent volcanic eruptions. Fig. 4 shows increased cloudiness in the region of the ice sheet, but what type of clouds? Typically high clouds warm and low clouds cool, but in summer there is a lot more cooling because of enhanced insolation. And what is the diurnal cycle of the cloudiness? At night and in the winter, clouds warm, but during the day and in the summer they cool. It would be easy to extract this information from the climate model runs and plot the actual cloud radiative forcing changes rather than an averaged cloud amount.

In any case, the model used here simulates a 3.5°C cooling over the FIS from high-latitude

eruptions. How, then, could there have been enhanced melting? In any case, in the summer more clouds should cause cooling and have contributed to the cooling.

3. Is there really a large volcanic ash albedo effect? Ash tends to be rather high albedo. Calculate the radiative forcing, including under clouds.

4. If ash is important, it should be in the Swedish cores. Were the layers with thicker varves enhanced in volcanic ash?

5. How much volcanic ash was actually transported to Sweden? Particles are large and are not transported large distances. How large would eruptions have to be to have a substantial transport? What evidence is there from recent eruptions? Did the two largest high latitude eruptions, 1783 Laki, a huge eruption, or 1912 Katmai, produce any ash in Sweden? For ash to get to Sweden, the wind would have had to have been in a very specific pattern from the volcano to the point of deposition, unlikely for a distant volcano.

6. The authors argue that enhanced precipitation after volcanic eruptions enhances ice melt, but Fig. 3 shows no enhanced precipitation over the FIS.

7. Which summer is shown in Fig. 3? Is it the same year as the eruption or the next year? For a high latitude eruption, all the aerosol would have been removed by the next year, so they need to use THE SAME YEAR as the eruption, but for a tropical one, it would not have been until THE YEAR AFTER the eruption that high latitude climate response would be seen. The authors need to make this clear and adjust their analysis if they used different summers.

8. Which leaves us with the correspondence between ETVs and volcanic deposition in Figs. 2e and 2f. Since the period studied is about 500 years and the width of the volcanic samples is 3-6 years (please say this more precisely), that only leaves about 100 independent signals, and it seems like, from Fig. 2c that most of the years have volcanic signals. Thus it is not surprising that ETVs would line up with some volcanic signal. I suggest that this is just by chance.

The physical mechanisms have certainly not been proven – only suggested without robust evidence. Please quantify the radiative forcing for the volcanic ash under clear and cloudy conditions. Please quantify the cloud radiative forcing from the model. Since the model gives so much cooling, I just don't understand how these suggested radiative effects can overwhelm this. At least, this forcing has to be calculated.

In addition, there are other minor issues:

The authors did not correct my previous point about "feedbacks." Similarly, for "coupling." A feedback or coupling requires a response to a forcing, that in turn affects the initial climate response. In all instances, they are talking about forcings only.

Line 59 says 57 varves, but line 600 says 56. Which is it?

Line 118: What is a "potential isochrone?"

Line 199: What is a "volcanic isochrone?"

Why are the time axes plotted backwards? Time should increase from left to right.

There are a few other issues in the attached annotated manuscript.

We thank the referees for their time and energy to review our manuscript once again. Please find below our responses to the points and comments raised by the reviewers.

Reviewer #1 (Remarks to the Author):

I reviewed the revised manuscript and read the comments of all reviewers and the authors' responses. In the revised version, the authors have addressed most of the raised concerns. Overall, they provided convincingly evidence that rapid melting events indicated by ETVs in the foreland of the margin of the Fennoscandian ice sheet were somehow associated with sub-aerial volcanism, with little chance that this relation is caused by pure chance. This is, to my knowledge, the first time that two very different proxy records during the last glacial have been compared to each other at an annual resolution, thus taking full advantage of high dating precision of these records. Using in addition published tephra studies from Greenland ice cores and ice core records from Antarctica (which I comment on in more details below) the authors now regard especially high-latitude eruptions (most likely from Iceland) as the main contributors to increased volcanic activity, which I agree on.

We are glad that Reviewer#1 agrees with us on the value of our proxy record. We now provide further results from runoff/melt model experiments to substantiate the mechanisms that we originally suggested.

Regarding the potential mechanisms involved, the authors suggest two possibilities how volcanism could trigger enhanced ice-sheet melting: 1) Changes of cloud-cover caused by sulphate aerosol injection and 2) increased ice-albedo due to deposition and accumulation of light absorbing dark volcanic ash. While it is difficult to assess and quantify the relative importance and magnitudes of these proposed effects, the authors do present modern analogues and preliminary climate model results that suggest that these effects appear to be at least plausible in explaining the observed association. More studies will certainly be necessary to further investigate such linkages in the future (including potentially more dedicated modeling studies), but on the basis of the exceptional strong associations drawn from the high-res. observational records at hand at the moment, the authors have, in my view, done the best possible analysis.

Below I added a few specific comments:

L. 27: According to your Fig. 3, the area of increased cloud cover following Laki type eruptions is also covering Scandinavia, not only the Atlantic.

The abstract has been changed.

L. 104: You may want to include that ECM record has much higher resolution than the GISP2 record, otherwise people may (falsely) believe the GISP2 record is an exceptionally good ice-core record. It is not, at least compared to present day standards. It is just the only record available for Greenland.

This information has now been included.

L. 193: Please provide the reference for the “50x more frequent Icelandic eruptions”. I believe it is 41 not 40. If a factor 50 was indeed real, than I assume the majority of this increase must come from small magnitude eruptions as such a large increase is not supported by the GISP2 ice core record (see my earlier comment on your original submission).

This reference has been changed.

L. 202: Better weaken down the statement such as “...no clear signs of a number of large-magnitude tropical volcanic eruptions over most of the time” or a “general paucity of strong and frequent activity in the Southern Hemisphere”. The volcanic synchronization performed previously (Veres et al., 2013) was stopped at a GICC05 age of 11,518 yr BP. Below that, the stratigraphic matching is based on CH₄ with a relative age uncertainty of ± 300 years. This means that AICC2012 and GICC05 are NOT synchronized on annual/decadal timescales that would allow you to attribute volcanic signals from Antarctica to the Greenland record (your grey bars). It is also very unlikely that over 900 years there would have been no tropical eruptions at all, when about 80 occurred over the past 2,500 years (Sigl et al., 2015). Still, you may be able to argue for an absence of many large tropical volcanoes on the basis of the absolute number of events over the entire 900 year window for Greenland vs. Antarctica, respectively. Severi et al. (2007) identified only three common events in EDML and EDC, whereas Rasmussen et al. (2013) identified 17 common events between NGRIP2 and NEEM. But I would not go down the path to claim that the three large signals in Antarctica did not have a Greenland counterpart.

The line has been rephrased and the statement toned down. It now reads: “Furthermore, there is a paucity of many large tropical events over the time window $\sim 13,200$ -12,300 years BP and generally a higher frequency of volcanic events recorded in Greenland as compared to Antarctica.”

L. 220-221: The really widespread dispersal of crypto-tephra for so many events is a relatively new observation that also benefited from new approaches how ice-cores are now analyzed. If you had space you could for example move the review paper (reference 38) up to the traditional approach (line 221) and add newer references for crypto-tephra that were detected with using the new sampling approaches (Jensen et al., 2014; Ponomareva et al., 2015; Sun et al., 2014). As you already mentioned in your reply regarding Siwan Davies personal communication and your on manuscript on North American tephra in Scandinavia such events are much more common than is currently anticipated. These published papers will also provide evidence that tephra dispersion can be widespread even with moderate SO₂ injections.

Thank you for this observation. The suggested references have now been included.

L 249/251: Why would the frequency of eruptions impact the magnitude of response for a single event? To you suggest a sustained impact from multiple eruptions occurring closely in time? Or do you suggest that also the magnitude of last glacial eruptions may have been higher. For both there is some evidence in the proxy records. I realized you downplayed the potential role that eruptions were larger during the late Glacial partly in response to my earlier critique. I did not intend to deny this aspect completely, but mostly wanted to criticize that some of the conclusions were drawn by comparing the ice-core concentrations directly with modern events. You could write "...overall increase in frequencies and magnitudes (Maclennan et al., 2002; Thordarson and Hoskuldsson, 2008)" as the geological records do indicate some very large volcanic eruptions.

The text has been changed accordingly and the new references have been included.

L. 265: higher rates of tephra deposition

This has been changed.

L. 365: typo: eruption

After major revisions, this part of the text has been removed.

References: Correct redundant citations!

This has been corrected.

Fig. S1:

I would remove the grey vertical bars and use the higher abundances of Greenland volcanic events relative to Antarctica as an argument in support of high volcanic activity in NH high-latitudes. On which ice core is the number of annual layers plus uncertainty based? EDML? To my knowledge the annual-layer dating of EDML has not (yet?) been published in a peer-reviewed journal, and the absolute age of the CH₄ based tie-point is rather poorly constrained. Readability and figure captions for all figures have strongly improved.

We agree with Reviewer#1 that the synchronization between Greenland and Antarctic ice cores around this interval is still not too well constrained and that the absolute age of the CH₄ tie point is rather loose. However, we would still prefer to show the vertical bars for reference and we leave to the editor to deciding. In the meanwhile, we have mentioned in the main text that there is a general higher frequency of volcanic events recorded in Greenland relative to Antarctica as suggested by Reviewer#1.

The cumulative age uncertainty of Antarctic ice cores presented in the figure is based on the Antarctic ice core chronology 2012 (AICC2012) (Bazin et al., 2013; Veres et al., 2013). This is now specified in the caption of Figure S1.

Bazin, L. *et al.* An optimized multi-proxy, multi-site Antarctic ice and gas orbital

chronology (AICC2012): 120-800 ka. *Climate of the Past* **9**, 1715–1731 (2013).

Veres, D. *et al.* The Antarctic ice core chronology (AICC2012): An optimized multi-parameter and multi-site dating approach for the last 120 thousand years. *Climate of the Past* **9**, 1733–1748 (2013).

Reviewer #2 (Remarks to the Author):

My previous concerns about Figure 3 have been addressed in the revised text. I read the full paper again and found it to be better supported and also much easier to read with new data and references. I did find a few typographical errors that I list below.

I strongly recommend publishing this paper after seeing the revised version.

Line 47 – “annual glacial varves” is redundant. All varves are annual sediment layers by definition and there is no such thing as a non-annual varve. You could say just “glacial varves”.

This is generally correct. However, some varve records can have sub-annual/seasonal resolution (as our records for instance). We therefore deem it is important to underscore that here we present data at annual resolution.

Line 183 – The citation of Fig. 4 here seems unnecessary since it is given in line 182.

This has been changed.

Line 202 – “On the contrary” starts a line that is not contrary to the previous line in the text and seems to support it if anything.

This has been changed.

Line 234 – “mechanism” should be plural – “mechanisms”

This has been changed.

Line 270 – “northern should be capitalized as “Northern”

This has been changed.

Line 352 – the reference numbers 41 and 42 are incorrect and should be 56 and 57. Shouldn't the author references be removed and replaced with the citation numbers?

This has been changed.

Line 602 – “ions” should be “ion”

This has been changed.

Lines 609 and 610 – the references to (d) and (e) should be (e) and (f).

This has been changed.

Lines 614 to 620 – Starting with “Grey bars indicate” to the end on line 620 should be associated with the entry for part d of the figure and not that for e-f.

This has been changed.

Line 619 – I suggest that “displayed” should be changed to “labeled” to make it clear that you are not indicating a different form of data but only labels on the graph.

This has been changed.

We take this opportunity to thank Reviewer#2 for spotting all these inaccuracies.

Reviewer #3 (Remarks to the Author):

Review of Muschitiello et al. revised

I still recommend that this paper be rejected. The hypothesis that large high-latitude volcanic eruptions cause excessive melting of the Fennoscandian ice sheet is not supported by the evidence.

1. Because of the natural variability of sulfate deposition on ice cores, as acknowledged by the authors in the caption for Fig. 2, volcanic time series produced from the past 1000 years typically use 20 or more ice cores, to account for this noise. Any one individual core can completely miss large volcanic eruptions, or have amplified deposition because of weather.

We generally agree with Reviewer#3 that more ice core records are needed to improve the signal to noise ratio. Unfortunately we don't avail of multiple Greenlandic volcanic records at the moment and analysing volcanic sulphates in several different ice cores is beyond the scope of this study. Note also that we acknowledged the effect of dust levels on suppressing the acidity signal in ice cores rather than downplaying the value of the GISP2 sulfate record. Furthermore, we think there is no reason to undermine the overall quality of the GISP2 record, which captures reasonably well all the major historical eruptions when compared to the combined Greenlandic volcanic record of Sigl et al. (2015) (please compare Figure 3 of first submission with Figure 4 second submission).

Sigl, M. *et al.* Timing and climate forcing of volcanic eruptions for the past 2,500 years. *Nature* **523**, 543–549 (2015).

2. One climate model is inadequate for simulating cloud response to volcanic eruptions. Cloud responses are very noisy, and each model is different. And the model use has not been validated adequately with recent volcanic eruptions. Fig. 4 shows increased cloudiness in the region of the ice sheet, but what type of clouds? Typically high clouds warm and low clouds cool, but in summer there is a lot more cooling because of enhanced insolation. And what is the diurnal cycle of the cloudiness? At night and in the winter, clouds warm, but during the day and in the summer they cool. It would be easy to extract this information from the climate model runs and plot the actual cloud radiative forcing changes rather than an averaged cloud amount.

In any case, the model used here simulates a 3.5°C cooling over the FIS from high-latitude eruptions. How, then, could there have been enhanced melting? In any case, in the summer more clouds should cause cooling and have contributed to the cooling.

Based on the criticism raised by Reviewer#3, we now provide a comprehensive quantitative analysis of the ice-sheet runoff response to the climatic change induced by high-latitude volcanic eruptions (starting both in summer and winter) using a model capable of fully account for density, temperature and albedo changes of the snow/ice column (Morris et al., 2014).

Several experiments were run using a field-validated one-dimensional energy and mass balance model of melt, refreezing and runoff processes that occur within a given snowpack/ice column (Morris et al., 2014). Idealised simulations were conducted for both volcanic and non-volcanically forced conditions, as determined by NorESM1-M climate model output and prescribed snow/ice albedo changes due to ash fall. Specifically, snow/ice model simulations were undertaken using temperature, shortwave radiation and precipitation conditions equivalent to those of the Younger Dryas (no-volcano control) and those altered by a Laki-type eruption (volcanically forced) occurring in summer, and a second ensemble of simulations for a similar eruption beginning in winter. Sensitivity to initial snowpack/ice column conditions was fully explored through systematic evaluation of the modelled runoff response for given combinations of initial snow and firn thickness. The relative impact of the non-volcanic/volcanic forcing scenarios on runoff was also evaluated for each combination of snow/firn thickness and prescribed albedo change for different ice sheet elevations (new Table 1, Table 2, Figure S3).

We now show in the new melt model experiments that the cloud radiative effect is present and has indeed an impact on the ice-sheet runoff balance (new Fig. S4b). Even though the volcanic cooling offsets the radiative effect due to cloud cover in this particular experiment, the increase in cloudiness is able to promote increased ice runoff (see Fig. S4). We therefore stand by our original interpretation and argue that changes in cloud cover do have a feedback in enhancing the melting due to albedo changes associated with ash deposition

(this also agrees with contemporary observations from Greenland [Van Tricht et al., 2016]).

We also show that when changes in cloudiness are coupled with modest changes in albedo due to ash fall (for both summer and winter eruptions), these are more than sufficient to offset the reductions in runoff due to temperature (new Table 1; Fig. S3). Where runoff simulations have been conducted with all climate forcings included (summer and winter plots), these show significant increases in runoff are possible despite a drop in temperature where there is a relatively modest change in albedo ($\alpha_{adj} = \alpha_{orig} * 0.85$). The snowpack model itself is able to account for the various feedbacks between environmental forcings and melt, refreezing, percolation, albedo change (due to the evolution from fresh snow to firn), density change and temperature change that ultimately contribute towards runoff. The model is therefore physically realistic, which has also been previously demonstrated by its accurate simulation of the evolution of temperature and density change along a transect of Devon Ice Cap, Canada (implying accurate calculation of melt and refreezing; Morris et al., 2014). This provides confidence that the results generated by the snowpack model represent realistic runoff values for given conditions.

All these results are now presented in the Discussion.

Morris, R. M. *et al.* Field-calibrated model of melt, refreezing, and runoff for polar ice caps: Application to Devon Ice Cap. *Journal of Geophysical Research: Earth Surface* **119**, 1995–2012 (2014).

Van Tricht, K. *et al.* Clouds enhance Greenland ice sheet meltwater runoff. *Nature Communications* **7**, 10266 (2016)

3. Is there really a large volcanic ash albedo effect? Ash tends to be rather high albedo. Calculate the radiative forcing, including under clouds.

The statement “Ash tends to be rather high albedo” comes quite as a surprise. It is well-established that volcanic ash falls have a significant impact on ice albedo even when present only in traces and even when rhyolitic (i.e. grey) (e.g. Brock et al., 2007; Dadic et al., 2013; Kirkbride and Dugmore, 2003; Möller et al., 2014; Nield et al., 2013; Wiscombe and Warren, 1980). In addition, reconstructions (Thordarson and Hoskuldsson, 2008) indicate that most of the postglacial eruptions originated from Iceland, such as those discussed in our study, were likely associated with explosive mafic events that produce very dark tephra. Even sulphates (which has a relatively high scattering coefficient) have been shown to decrease the snow albedo (e.g. Garrett and Zhao, 2006).

Furthermore, we demonstrate within the snowpack model that even modest changes in albedo can result in significant changes in runoff (see reply above). We would also argue that even rhyolitic volcanic ash is of relatively low albedo compared to that of fresh snow (~0.8), while it is arguable that basaltic (i.e. brown/black) ash fall over Fennoscandia would be more likely during the late glacial as ice still covered most of Iceland (Geirsdóttir et al., 2009), making

phreatomagmatic processes more common (i.e. magma interactions with meltwater causing explosive eruptions). We therefore suggest that the albedo changes that we have simulated in the snowpack model may even be conservative estimates of potential albedo change. The albedo change will therefore be dependent on how much ash falls on the ice surface, and the albedo of the erupted material.

Indeed, as discussed above the albedo effect seems to be a major factor driving changes in ice-sheet runoff. The new results have been obtained by driving the runoff model using the NorESM1-M climate model output, including the simulated cloud cover response to high-latitude volcanism.

All these issues are now outlined in the Results and the Discussion.

Brock, B., Rivera, A., Casassa, G., Bown, F., Acuña, C., 2007. The surface energy balance of an active ice-covered volcano: Villarrica Volcano, southern Chile. *Annals of Glaciology* 45, 104–114.

Dadic, R., Mullen, P.C., Schneebeli, M., Brandt, R.E., Warren, S.G., 2013. Effects of bubbles, cracks, and volcanic tephra on the spectral albedo of bare ice near the Transantarctic Mountains: Implications for sea glaciers on Snowball Earth. *Journal of Geophysical Research: Earth Surface* 118, 1658–1676.

Garrett, T.J., Zhao, C., 2006. Increased Arctic cloud longwave emissivity associated with pollution from mid-latitudes. *Nature* 440, 787–789. doi:10.1038/nature04636

Geirsdóttir, Á., Miller, G.H., Axford, Y., Ólafsdóttir, S., 2009. Holocene and latest Pleistocene climate and glacier fluctuations in Iceland. *Quaternary Science Reviews* 28, 2107–2118.

Kirkbride, M.P., Dugmore, A.J., 2003. Glaciological response to distal tephra fallout from the 1947 eruption of Hekla, south Iceland. *Journal of Glaciology* 49, 420–428. doi:10.3189/172756503781830575

Möller, R., Möller, M., Björnsson, H., Gudmundsson, S., Pálsson, F., Oddsson, B., Kukla, P. a., Schneider, C., 2014. MODIS-derived albedo changes of Vatnajökull (Iceland) due to tephra deposition from the 2004 Grímsvötn eruption. *International Journal of Applied Earth Observation and Geoinformation* 26, 256–269. doi:10.1016/j.jag.2013.08.005

Nield, J.M., Chiverrell, R.C., Darby, S.E., Leyland, J., Virca, L.H., Jacobs, B., 2013. Complex spatial feedbacks of tephra redistribution, ice melt and surface roughness modulate ablation on tephra covered glaciers. *Earth Surface Processes and Landforms* 38, 95–102. doi:10.1002/esp.3352

Thordarson, T., Hoskuldsson, A., 2008. Postglacial volcanism in Iceland. *Jokull* 58, 197–228.

Wiscombe, W.J., Warren, S.G., 1980. A model for the spectral albedo of snow. I: Pure snow. *Journal of Atmospheric Sciences*. doi:10.1175/1520-0469(1980)037<2734:AMFTSA>2.0.CO;2

4. If ash is important, it should be in the Swedish cores. Were the layers with thicker varves enhanced in volcanic ash?

Regretfully, the original material used to generate the varve chronology is not available anymore to directly test this. However, we observe that volcanic ash tends to occur in correspondence with thick varves: the Vedde Ash in first instance (e.g. Wohlfarth et al., 1995; MacLeod et al., 2013) as also mentioned in the text (new line 270-272). This is common in postglacial varves from southern Sweden where volcanic glacial glass shards have been identified in relatively darker and thicker varve layers (L. Brunnberg, pers. comms).

Finally, it should be born in mind that even volumetrically small Icelandic eruptions, such as that of Eyjafjallajökull (Apr-May 2010), are capable to spread ash particles over a vast area, and ash can easily be transported to Scandinavia (Fig. R1). Critically, a new study (Watson et al., 2017) also shows that during the last 1,000 years ash clouds have been more common over Northern Europe than previously thought (mean return interval of volcanic ash cloud over the region of 44 ± 7 years) from mid-sized eruptions that were both sourced from Icelandic and North American volcanoes.

Figure R1: Composite map of the position of the Icelandinc volcanic ash cloud over several days (14-25 April 2010), based on maps available at http://www.metoffice.gov.uk/aviation/vaac/vaacuk_vag.html

Macleod, a., Brunberg, L., Wastegård, S., Hang, T., Matthews, I.P., 2014. Lateglacial cryptotephra detected within clay varves in Östergötland, south-east Sweden. *Journal of Quaternary Science* 29, 605–609. doi:10.1002/jqs.2738

Watson, E.J., Swindles, G.T., Savov, I.P., Lawson, I.T., Connor, C.B., Wilson, J.A., 2017. Estimating the frequency of volcanic ash clouds over northern Europe. *Earth and Planetary Science Letters* 460, 41–49.

Wohlfarth, B., Bjorck, S., Possnert, G., 1995. The Swedish time scale; a potential calibration tool for the radiocarbon time scale during the late Weichselian. *Radiocarbon* 37, 347–359.

5. How much volcanic ash was actually transported to Sweden? Particles are large and are not transported large distances. How large would eruptions have to be to have a substantial transport? What evidence is there from recent eruptions? Did the two largest high latitude eruptions, 1783 Laki, a huge eruption, or 1912 Katmai, produce any ash in Sweden? For ash to get to Sweden, the wind would have had to have been in a very specific pattern from the volcano to the point of deposition, unlikely for a distant volcano.

Based on the body of studies showing that ash from Alaskan and Icelandic medium-sized volcanic eruptions can very easily reach as far as Europe and deliver a significant amount of tephra (e.g. Gudmundsson et al., 2012; Jensen et al., 2014), it is likely that ash transport to the FIS was non-negligible (see also Fig. R1).

Furthermore, as also stressed above, we argue that phreatomagmatism (explosive eruptions) due to interaction with sub-glacial meltwater helped the dispersal of tephra (new lines 233-236). Lastly, we should bear in mind that based on our new runoff results, there is not necessarily the need for large transport of ash to cause a substantial increase in runoff of the ice sheet.

Gudmundsson, M.T., Thordarson, T., Höskuldsson, Á., Larsen, G., Björnsson, H., Prata, F.J., Oddsson, B., Magnússon, E., Högnadóttir, T., Petersen, G.N., Hayward, C.L., Stevenson, J. a., Jónsdóttir, I., 2012. Ash generation and distribution from the April-May 2010 eruption of Eyjafjallajökull, Iceland. *Scientific Reports* 2, 1–12. doi:10.1038/srep00572

Jensen, B.J.L., Pyne-O'Donnell, S., Plunkett, G., Froese, D.G., Hughes, P.D.M., Sigl, M., McConnell, J.R., Amesbury, M.J., Blackwell, P.G., van den Bogaard, C., Buck, C.E., Charman, D.J., Clague, J.J., Hall, V.A., Koch, J., Mackay, H., Mallon, G., McColl, L., Pilcher, J.R., 2014. Transatlantic distribution of the Alaskan White River Ash.

6. The authors argue that enhanced precipitation after volcanic eruptions enhances ice melt, but Fig. 3 shows no enhanced precipitation over the FIS.

We are no longer discussing the role of changes in precipitation and this section has been removed. However, any changes in runoff due to precipitation are fully incorporated within the snowpack simulations.

7. Which summer is shown in Fig. 3? Is it the same year as the eruption or the next year? For a high latitude eruption, all the aerosol would have been removed by the next year, so they need to use THE SAME YEAR as the eruption, but for a tropical one, it would not have been until THE YEAR AFTER the eruption that high latitude climate response would be seen. The authors need to make this clear and adjust their analysis if they used different summers.

Figure 3 and the results presented therein are no longer present/discussed in the new version of the manuscript.

8. Which leaves us with the correspondence between ETVs and volcanic deposition in Figs. 2e and 2f. Since the period studied is about 500 years and the width of the volcanic samples is 3-6 years (please say this more precisely) (this was already stated in the Results; new line 94), that only leaves about 100 independent signals, and it seems like, from Fig. 2c that most of the years have volcanic signals. Thus it is not surprising that ETVs would line up with some volcanic signal. I suggest that this is just by chance.

We understand the reviewer's concerns on the significance of the correlation between varve and volcanic events, and we want to stress that we took the significance testing very seriously, which is the reason we adopted a double test to begin with.

Frankly, we are surprised that Reviewer#3 does not: 1) try to substantiate their assertion through critique of the significance tests applied; 2) acknowledge the fact that the records are precisely aligned throughout the interval under consideration as discussed in detail in our manuscript; 3) acknowledge that some of the major volcanic events have correlative major varve horizons. We have also re-addressed some of these issue in the previous revision where we presented additional statistical analysis showing the significance of the correlation for different placements of the varve chronology on the ice core time scale to further highlight that the correspondence between thick varves and volcanic events cannot be explained by a random pattern (Fig. S2e). Along the lines of this criticism, we now provide a third Monte Carlo test of significance where the correlation between thick varve years and volcanic events is inferred by comparing the GISP2 data to 1,000 independent Gaussian white noise time series. The third test confirms –once again– the robustness and significance of the correlation (new Fig. 2g).

The physical mechanisms have certainly not been proven – only suggested without robust evidence. Please quantify the radiative forcing for the volcanic ash under clear and cloudy conditions. Please quantify the cloud radiative forcing from the model. Since the model gives so much cooling, I just don't understand how these suggested radiative effects can overwhelm this. At least, this forcing has to be calculated.

We have now provided an extensive quantitative analysis of the ash and cloud radiative effects on runoff under volcanically- and non-volcanically forced climate conditions (see previous replies).

In addition, there are other minor issues:

The authors did not correct my previous point about “feedbacks.” Similarly, for “coupling.” A feedback or coupling requires a response to a forcing, that in turn affects the initial climate response. In all instances, they are talking about forcings only.

This has been changed.

Line 59 says 57 varves, but line 600 says 56. Which is it?

In new line 84 we further specify that we only focus on one specific portion of the chronology, which comprises 56 out of the total 57 varve diagrams.

Line 118: What is a “potential isochrone?”

This has been rephrased and it now reads: “We observe that the number of annual layers between the identified isochrones is consistent with [...]”.

Line 199: What is a “volcanic isochrone?”

This has been rephrased and it now reads: “Furthermore, there is a paucity of many large tropical events over the time window ~13,200-12,300 years BP [...]”.

Why are the time axes plotted backwards? Time should increase from left to right.

It is very common in paleoclimate studies to plot time from right to left. We do not think this should be changed but we leave to editor deciding.

Line 219-222: I don't understand this claim. If only two tephra deposits were found in Greenland, why do you think there were so many over the FIS?

As detailed in the text and also discussed by Reviewer#2, the number of tephra identified in ice cores does not reflect the actual number of tephra horizons present in the ice. Crypto-tephra analysis is a relatively new approach that is drastically increasing the number of tephra occurrences identified in ice cores. More importantly, these new techniques are revealing that even far-field

eruptions associated with moderate SO₂ injections can reach Greenland and Europe more easily than anticipated.

Jensen, B. J. L. *et al.* Transatlantic distribution of the Alaskan White River Ash. *Geology* **42**, 875–878 (2014).

Ponomareva, V., Portnyagin, M. & Davies, S. M. Tephra without Borders: Far-Reaching Clues into Past Explosive Eruptions. *Frontiers in Earth Science* **3**, 1–16 (2015).

Sun, C. *et al.* Ash from Changbaishan Millennium eruption recorded in Greenland ice: Implications for determining the eruption's timing and impact. *Geophysical Research Letters* **41**, 694–701 (2014).

Watson, E.J., Swindles, G.T., Savov, I.P., Lawson, I.T., Connor, C.B., Wilson, J.A., 2017. Estimating the frequency of volcanic ash clouds over northern Europe. *Earth and Planetary Science Letters* **460**, 41–49.

There are a few other issues in the attached annotated manuscript.

All these issues have been addressed.

Reviewers' comments:

Reviewer #1 (Remarks to the Author):

Corresponding Author: Francesco Muschitiello

Title: Impact of volcanic aerosols on Fennoscandian Ice Sheet melting during the last deglaciation

I have read now this 3rd version of the original manuscript, and also their responses to the three reviewers' comments. Overall, I find my comments have been well addressed and the manuscript has been improved. The major changes in comparison to the previous version is that the reviewers performed additional simulations aiming to test if simulated changes in cloud cover and snow albedo are capable of increasing the run-off from ice sheets. Without claiming expertise in snowpack modeling, I am, nevertheless, surprised that the effects of a simulated albedo reduction from enhanced deposition of light-absorbing ash particles on the ice-sheet is so much more pronounced during a winter-eruption case compared to a summer eruption. I would have expected the largest effects of albedo changes in the Arctic cryosphere during spring when an excess of dark particles on top of the bright winter snow results in the biggest relative changes of local albedo, in agreement with simulations of industrial black-carbon induced impact on the Arctic cryosphere (Flanner et al., 2009). Maybe I am lacking some information about how these simulations have been conducted. I take it from the method part that the injection of particles was over a 4-month period, so that most of the particles for the winter eruption scenario would have been deposited on the ice-sheet during the darker winter months (Dec-Mar) with reduced short wave radiation. What am I missing?

I am in the following commenting specifically on trends, magnitudes and sources of volcanic activity, and associated sulfate injection and ash generation during the deglaciation since reviewer 3 in his review is questioning some of the underlying records and conclusions.

The notion that large eruptions may be missed or largely exaggerated in a single ice-core is probably motivated by a recent study performed for one of the driest places in Antarctica (Gautier et al., 2016) where volcanic sulfate in the uppermost few centimeter is exposed to wind. Even for comparable low accumulation ice-core sites, the complete absence of large signals is scarcely, if ever, observed in a multitude of ice cores (see pdf attached for the example of Antarctica; note that GISP2 and WDC are comparable in terms of mean accumulation rate). It appears Gautier et al., were just unlucky in their choice of the drilling sites. For high-accumulation sites, such as GISP2, in which the volcanic fallout is contained in the upper 40-120 cm of the snowpack (depending in the duration of deposition) post-depositional changes of concentrations is very limited. So there is little evidence that the GISP2 sulfate record (despite its coarse resolution) is biased in any way that would adversely affect the interpretation as performed in this study.

All observational records, point to an increase of volcanic activity during the last deglaciation specifically in formerly glaciated areas (Brown et al., 2014; Huybers and Langmuir, 2009; Kutterolf et al., 2013). Sulfate levels are enhanced in Greenland ice cores and more and more volcanic ash layers are continuously discovered in proxy archives (Abbott et al., 2016; Bourne et al., 2016; Davies, 2015). The ice-core records from Greenland are full of tephra layers even for the past 2,000 years from far distant sources many of which have only recently been discovered due to analytical improvements in detection of tephra in ice cores (Jensen et al., 2014; Sigl et al., 2015; Sun et al., 2014). On the basis of these new results, it is not unlikely that volcanic areas such as Iceland and Alaska were during the deglaciation a frequent source of widespread ash dispersion in the NH high-latitudes atmosphere that would have been transported with the westerlies and deposited on the Fennoscandian Ice Sheet. Of course a direct proof remains difficult, with this ice-sheet now gone.

If accepting that the co-occurrence of ETVs and volcanic eruptions was not by chance (as is

supported by their analyses) increased cloud cover and decreased snow albedo appear as a very plausible mechanisms that could enhance short-term melting extremes despite short-term cooling. I could also think of short-term ozone depletion events following volcanic eruptions from the release of large amounts of halogens into the stratosphere (Cadoux et al., 2015; Calvo et al., 2015), a mechanism which may be equally difficult to proof. Of course it is desirable to repeat the tests for synchronicity between eruptions and ETVs with higher resolved state-of-the-art ice-core records should those become available in the future.

References

- Abbott, P. M., Bourne, A. J., Purcell, C. S., Davies, S. M., Scourse, J. D., and Pearce, N. J. G.: Last glacial period cryptotephra deposits in an eastern North Atlantic marine sequence: Exploring linkages to the Greenland ice-cores, *Quat Geochronol*, 31, 62-76, 2016.
- Bourne, A. J., Abbott, P. M., Albert, P. G., Cook, E., Pearce, N. J. G., Ponomareva, V., Svensson, A., and Davies, S. M.: Underestimated risks of recurrent long-range ash dispersal from northern Pacific Arc volcanoes, *Sci Rep-Uk*, 6, 2016.
- Brown, S. K., Croswell, H. S., Sparks, R. S. J., Cottrell, E., Deligne, N. I., Guerrero, N. O., Hobbs, L., Kiyosugi, K., Loughlin, S. C., Siebert, L., and Takarada, S.: Characterisation of the Quaternary eruption record: analysis of the Large Magnitude Explosive Volcanic Eruptions (LaMEVE) database, *Journal of Applied Volcanology*, 3, 5, 2014.
- Cadoux, A., Scaillet, B., Bekki, S., Oppenheimer, C., and Druitt, T. H.: Stratospheric Ozone destruction by the Bronze-Age Minoan eruption (Santorini Volcano, Greece), *Sci Rep-Uk*, 5, 2015.
- Calvo, N., Polvani, L. M., and Solomon, S.: On the surface impact of Arctic stratospheric ozone extremes, *Environ Res Lett*, 10, 2015.
- Davies, S. M.: Cryptotephra: the revolution in correlation and precision dating, *J Quaternary Sci*, 30, 114-130, 2015.
- Flanner, M. G., Zender, C. S., Hess, P. G., Mahowald, N. M., Painter, T. H., Ramanathan, V., and Rasch, P. J.: Springtime warming and reduced snow cover from carbonaceous particles, *Atmos Chem Phys*, 9, 2481-2497, 2009.
- Gautier, E., Savarino, J., Erbland, J., Lanciki, A., and Possenti, P.: Variability of sulfate signal in ice core records based on five replicate cores, *Clim Past*, 12, 103-113, 2016.
- Huybers, P. and Langmuir, C.: Feedback between deglaciation, volcanism, and atmospheric CO₂, *Earth Planet Sc Lett*, 286, 479-491, 2009.
- Jensen, B. J. L., Pyne-O'Donnell, S., Plunkett, G., Froese, D. G., Hughes, P. D. M., Sigl, M., McConnell, J. R., Amesbury, M. J., Blackwell, P. G., van den Bogaard, C., Buck, C. E., Charman, D. J., Clague, J. J., Hall, V. A., Koch, J., Mackay, H., Mallon, G., McColl, L., and Pilcher, J. R.: Transatlantic distribution of the Alaskan White River Ash, *Geology*, 42, 875-878, 2014.
- Kutterolf, S., Jegen, M., Mitrovica, J. X., Kwasnitschka, T., Freundt, A., and Huybers, P. J.: A detection of Milankovitch frequencies in global volcanic activity, *Geology*, 41, 227-230, 2013.
- Sigl, M., Winstrup, M., McConnell, J. R., Welten, K. C., Plunkett, G., Ludlow, F., Büntgen, U., Caffee, M., Chellman, N., Dahl-Jensen, D., Fischer, H., Kipfstuhl, S., Kostick, C., Maselli, O. J., Mekhaldi, F., Mulvaney, R., Muscheler, R., Pasteris, D. R., Pilcher, J. R., Salzer, M., Schüpbach, S., Steffensen, J. P., Vinther, B. M., and Woodruff, T. E.: Timing and climate forcing of volcanic eruptions for the past 2,500 years, *Nature*, 523, 543-549, 2015.

Sun, C. Q., Plunkett, G., Liu, J. Q., Zhao, H. L., Sigl, M., McConnell, J. R., Pilcher, J. R., Vinther, B., Steffensen, J. P., and Hall, V.: Ash from Changbaishan Millennium eruption recorded in Greenland ice: Implications for determining the eruption's timing and impact, *Geophys Res Lett*, 41, 694-701, 2014.

Reviewer #3 (Remarks to the Author):

I still don't accept the fundamental premise of the paper. Of the 6 largest volcanic eruptions in Fig. 2b and 2c, only 2 correspond to thick varves. Of the 11 largest acidity peaks in Fig. 2d, only 1 corresponds to a thick varve. All the rest looks like noise. Again, because the sulfate signals are multi-year, it would be hard for one not to correspond to a thick varve, no matter how many fancy statistical tests are used. The synchronicity of ETVs with very small volcanic sulfate layers is not a surprise, and not physically important. Without proof that small volcanic eruptions could have the claimed effect, I do not accept the statistical tests.

The presentation of new climate model simulations in supplemental material is not acceptable. The entire experiment needs to be explained in detail. The model used needs to be validated. Showing anecdotal results that support the premise, without any details is not acceptable. For example, Fig. S3 describes purple values as positive, but the purple plots (top left) only show negative values, so I am very confused. Furthermore, the entire description of the climate modeling is "Difference in annual runoff (cm w.e.) between simulations driven by non-volcanically forced and volcanically (Laki) forced climate conditions." What is "w.e.?" In lines 182-183 in the supplemental, it says that monthly cloud values are smoothed to hourly values, but this is not correct. Clouds have large short-term variations, and using monthly mean values will give the wrong answer. Similarly for precipitation, relative humidity, and SWRF. Fig. S6 gives anecdotal results from one simulation, but does not say how the simulation was done, and whether the cloud results are correct for volcanic eruptions for which we have data, such as 1912 Katmai. If you are going to base the fundamental conclusion of your paper on an off-line ice model designed for a different purpose driven by climate model simulations, you have to save the output from the climate model at the time resolutions demanded by the ice model. You have to explain how many ensemble members were used for each set of climate simulations. You have to show that this climate model does a good job for the present climate. You have to validate the Laki simulations with reconstructions of European climate that show what actually happened. There are far too many details left out to evaluate the accuracy of the results presented here. For example, see the Methods section of Tricht et al. (2016) to see how it is done.

There are 10 additional comments in the attached annotated manuscript.

We thank once again the referees for their time to review our study. Please find below our responses to the points and comments raised by the reviewers.

Reviewer #1 (Remarks to the Author):

I have read now this 3rd version of the original manuscript, and also their responses to the three reviewers' comments. Overall, I find my comments have been well addressed and the manuscript has been improved. The major changes in comparison to the previous version is that the reviewers performed additional simulations aiming to test if simulated changes in cloud cover and snow albedo are capable of increasing the run-off from ice sheets. Without claiming expertise in snowpack modeling, I am, nevertheless, surprised that the effects of a simulated albedo reduction from enhanced deposition of light-absorbing ash particles on the ice-sheet is so much more pronounced during a winter-eruption case compared to a summer eruption. I would have expected the largest effects of albedo changes in the Arctic cryosphere during spring when an excess of dark particles on top of the bright winter snow results in the biggest relative changes of local albedo, in agreement with simulations of industrial black-carbon induced impact on the Arctic cryosphere (Flanner et al., 2009). Maybe I am lacking some information about how these simulations have been conducted. I take it from the method part that the injection of particles was over a 4-month period, so that most of the particles for the winter eruption scenario would have been deposited on the ice-sheet during the darker winter months (Dec-Mar) with reduced short wave radiation. What am I missing?

The winter eruption scenario has the eruption initiating during winter and lasting the full duration of the melt season. This is mentioned in L191-194 and L258-261 of the previous version of the manuscript, however we have now made it more explicit by adding L185-186 and L415 in this version of the manuscript.

We do not attempt to conduct a 4 month winter eruption simulation as the washout processes of tephra in the ablation zones are likely to be spatially heterogenous and are currently poorly constrained for contemporary ice sheets, let alone for paleo ice sheets. Consequently the purpose of the experiment is to gain insight into the difference in the magnitude of runoff volumes between eruptions initiating in winter and summer, and the roles that a combination of timing, duration and conditions antecedent to an eruption play in conditioning the snowpack/ice column for producing runoff.

As to the reason why the ash-induced albedo effect is much more pronounced during winter than in summer, this is due to the lack of any significant cooling in the winter simulation. This is discussed in L189-190.

I am in the following commenting specifically on trends, magnitudes and sources of volcanic activity, and associated sulfate injection and ash generation during the deglaciation since reviewer 3 in his review is questioning some of the underlying records and conclusions.

The notion that large eruptions may be missed or largely exaggerated in a single ice-core is probably motivated by a recent study performed for one of the driest places in Antarctica (Gautier et al., 2016) where volcanic sulfate in the uppermost few centimeter is exposed to wind. Even for comparable low accumulation ice-core sites, the complete absence of large signals is scarcely, if ever, observed in a multitude of ice cores (see pdf attached for the example of Antarctica; note that GISP2 and WDC are comparable in terms of mean accumulation rate). It appears Gautier et al., were just unlucky in their choice of the drilling sites. For high-accumulation sites, such as GISP2, in which the volcanic fallout is contained in the upper 40-120 cm of the snowpack (depending in the duration of deposition) post-depositional changes of concentrations is very limited. So there is little evidence that the GISP2 sulfate record (despite its coarse resolution) is biased in any way that would adversely affect the interpretation as performed in this study.

All observational records, point to an increase of volcanic activity during the last deglaciation specifically in formerly glaciated areas (Brown et al., 2014; Huybers and Langmuir, 2009; Kutterolf et al., 2013). Sulfate levels are enhanced in Greenland ice cores and more and more volcanic ash layers are continuously discovered in proxy archives (Abbott et al., 2016; Bourne et al., 2016; Davies, 2015). The ice-core records from Greenland are full of tephra layers even for the past 2,000 years from far distant sources many of which have only recently been discovered due to analytical improvements in detection of tephra in ice cores (Jensen et al., 2014; Sigl et al., 2015; Sun et al., 2014). On the basis of these new results, it is not unlikely that volcanic areas such as Iceland and Alaska were during the deglaciation a frequent source of widespread ash dispersion in the NH high-latitudes atmosphere that would have been transported with the westerlies and deposited on the Fennoscandian Ice Sheet. Of course a direct proof remains difficult, with this ice-sheet now gone.

If accepting that the co-occurrence of ETVs and volcanic eruptions was not by chance (as is supported by their analyses) increased cloud cover and decreased snow albedo appear as a very plausible mechanisms that could enhance short-term melting extremes despite short-term cooling. I could also think of short-term ozone depletion events following volcanic eruptions from the release of large amounts of halogens into the stratosphere (Cadoux et al., 2015; Calvo et al., 2015), a mechanism which may be equally difficult to proof. Of course it is desirable to repeat the tests for synchronicity between eruptions and ETVs with higher resolved state-of-the-art ice-core records should those become available in the future.

We appreciate R1's inputs in support of our original claims. We are also thankful for pointing out a number of studies we were not aware of. We have now included some of these citations in the main text (i.e., Brown et al., 2014 – L211; Kutterolf et al., 2013 – L211; Bourne et al., 2016 – L225; Davies, 2015 – L278).

Reviewer #3 (Remarks to the Author):

I still don't accept the fundamental premise of the paper. Of the 6 largest volcanic eruptions in Fig. 2b and 2c, only 2 correspond to thick varves. Of the 11 largest acidity peaks in Fig. 2d, only 1 corresponds to a thick varve. All the rest looks like noise. Again, because the sulfate signals are multi-year, it would be hard for one not to correspond to a thick varve, no matter how many fancy statistical tests are used. The synchronicity of ETVs with very small volcanic sulfate layers is not a surprise, and not physically important. Without proof that small volcanic eruptions could have the claimed effect, I do not accept the statistical tests.

We understand R3's point of view but we beg to disagree and kindly ask R3 to provide some clarifications on what specifically about our statistical tests that he/she doesn't agree with. We think that R3 should give a clear rationale for not believing the statistical tests rather than just stating it without any critique of the methods we employ. In our opinion, we have done all that we could to demonstrate that the correspondence between thick varves and volcanic events is not a coincidence (as also acknowledged by R1 and R2), through a systematic quality assessment of our chronology and three independent Monte Carlo significance tests of the correlation between the records investigated. We also believe we have made all possible efforts to properly interpret the available ice-core and our varve records, and with the Fennoscandian Ice Sheet gone it is impossible for us to directly measure the relationship between volcanic eruptions and melting events other than inferring such relationship from the paleoclimate records.

The presentation of new climate model simulations in supplemental material is not acceptable. The entire experiment needs to be explained in detail. The model used needs to be validated.

We agree with R3 and with the editor that the presentation of the model experiments should be transferred to the main text. We have now moved all this information to the Methods section.

The runoff model is field validated for Devon Ice Cap (Morris et al., 2014). This was stated in the previous version of the manuscript in L425-428, and was alluded to in L118-122 of the Supplementary Information. This is now restated in L160-161 and L434-446 in this version of the manuscript. We are afraid we are unsure what extra details R3 requires, though we consider our experiments to be replicable given the information we have provided in the manuscript and SI.

The results in Morris et al. (2014) demonstrate that the model performs exceptionally well in recreating the surface mass balance conditions and observed snowpack evolution of an altitudinal transect on Devon Ice Cap similar to that simulated in this study (albeit under more idealized conditions). The idealized conditions for this study are necessary given that (as R1 points out) direct proof of initial snowpack conditions and surface mass balance response is difficult given that the Fennoscandian Ice Sheet no longer exists. Direct validation against observations of the FIS snowpack is therefore obviously impossible. However, in an attempt to circumvent the uncertainties associated with this, we explore an extremely wide

range of parameter space relating to initial snowpack conditions (shown in Figure S3, and already explained in the Supplementary Information – now in L538-547 in the main text).

Given our use of contemporary field validated input parameter values for the runoff model and comprehensive exploration of snowpack/ice column density uncertainty, we are confident that our results reflect the most realistic and honest representation of runoff that it is possible to achieve for a paleo ice sheet.

Morris, R. M. *et al.* Field-calibrated model of melt, refreezing, and runoff for polar ice caps: Application to Devon Ice Cap. *Journal of Geophysical Research: Earth Surface* **119**, 1995–2012 (2014).

Showing anecdotal results that support the premise, without any details is not acceptable. For example, Fig. S3 describes purple values as positive, but the purple plots (top left) only show negative values, so I am very confused.

We thank R3 for pointing this out in Fig S3. The mention of purples as positive values arose due to the extreme of both ends of the color bar appearing purple to one of the authors (JL) when the positive value is in fact red. The figure caption has been changed to reflect this (SI - L78).

Following the current revisions we now consider the information provided in the manuscript and SI sufficient to allow our experiments to be replicated.

Furthermore, the entire description of the climate modeling is “Difference in annual runoff (cm w.e.) between simulations driven by non-volcanically forced and volcanically (Laki) forced climate conditions.” What is “w.e.?”

The abbreviation w.e. stands for water equivalent, and is standard terminology in surface mass balance modelling (an explanation of this abbreviation was provided in the captions for Tables 1 and 2 in the previous version of the manuscript). It accounts for the fact that the density of snow/firn/ice is variable by expressing the amount of runoff in terms of the equivalent amount of liquid water.

Nevertheless, we agree that given the readership of this article is likely to be much broader than that audience, so an explanation of the abbreviation is useful to have in both the main manuscript (L807 and L811) and the SI. Consequently, we now also explain the abbreviation in the caption S3 (SI L76-77).

In lines 182-183 in the supplemental, it says that monthly cloud values are smoothed to hourly values, but this is not correct. Clouds have large short-term variations, and using monthly mean values will give the wrong answer. Similarly for precipitation, relative humidity, and SWRF.

We thank R3 for pointing this out. This is a good point that requires consideration.

Unfortunately it is not possible to rerun the climate model to get daily or hourly data. It would be an enormous amount of work that would also require a significant computer data storage capacity. While it is true that clouds have large short-term variations, the nature of the experiment is highly idealised and models in general have large biases in reproducing correctly cloud cover. Furthermore, we focus on a very large region and we would like to stress that the runoff model only considers a mean value of total cloud cover over the entire region. In addition, small changes in total cloud cover in our climate model do not lead to significant differences. Therefore, re-performing all model simulations with all ensemble members to extract hourly cloud cover values is not feasible and would not lead to any appreciable difference. On the other hand, precipitation and relative humidity values are provided at daily intervals (L480 and L487 in the main text).

Nonetheless, to address potential concerns on the significance of the variability of cloudiness in controlling runoff, we have conducted a new ensemble of melt/runoff simulations where we add random noise (at a daily timescale) to the cloudiness data. This aims to evaluate the impact of short-term changes in this input variable on the overall trends and magnitudes of the runoff results generated by the snowpack model. The results of these simulations are now presented in the Supplementary Information and Supplementary Methods (Fig. S9) as the difference between the simulations where the noise has been added to the cloudiness data (Figure S8) and the original simulations (i.e. those presented in the manuscript – Fig. S3). The original cloudiness values were obtained by interpolating monthly mean cloudiness values from climate model output (Fig. S7), while the maximum magnitude of the noise added is ± 0.3 from these original values.

The same pattern of noise was added to both the volcanically and non-volcanically driven simulations. This ensures that both simulations are consistent, and only the impact of cloud cover variability on the overall results is evaluated. The monthly means of the cloudiness data with noise added are consistent with those of the original simulations.

In addition, adding noise to the cloudiness values will also change the shortwave radiation fluxes compared to the original simulations. Consequently for the new simulations the shortwave radiation flux is recalculated following the same method described in the Supplementary Information.

All remaining data used to drive the new simulations are consistent with those of the original simulations. Figure S7 shows results for the simulations with cloudiness data added in a format that is directly comparable to Figure S3. The absolute differences in runoff between Figures S3 and Figure S8 are also plotted on Figure S9.

In response to R3's comment that "using monthly mean [cloud] values will give the wrong answer" in the runoff simulations, the results show that introducing noise to the cloudiness inputs leads to minimal differences in runoff compared to the original simulations (Figure S9). This shows that our original approach of using monthly mean cloudiness data to drive our runoff simulations (as we have done in our manuscript)

is suitable. Also, where relatively larger differences do occur these are mainly at lower elevations, where differences in runoff are likely to have a smaller proportionate impact on the transport of sediment that could be related to the formation of exceptionally thick varves (ETVs; see also L250-267 in manuscript). It should be noted that the general trends and magnitude of results shown by both Figures S8 and S3 are also consistent, and does not lead the authors to doubt the veracity of the results provided by the runoff simulations or the conclusions drawn from them.

Fig. S6 gives anecdotal results from one simulation, but does not say how the simulation was done [...]

The caption for S6 from the previous manuscript did explain how inputs to the snowpack/ice model were treated, though we have now amended the caption to hopefully clarify this (SI L107-112).

[...] and whether the cloud results are correct for volcanic eruptions for which we have data, such as 1912 Katmai.

We believe that comparing the simulation of cloud cover in our Laki-type experiment to the observed changes in cloud cover after the Katmai's eruption should not be performed for the following reasons:

- 1) First and above all, one cannot separate the internal natural variability of cloud cover from the observed changes in cloud cover; therefore any attempt to validate the model in this way is fundamentally wrong and should be discouraged.
- 2) We have performed a Laki-type eruption, which was multi-stage, less explosive and long lasting eruption; on the other hand, Katmai was an explosive eruption (aerosol reaching 24 km) and lasted only a couple of days. Therefore, one cannot expect to get a comparable effect on clouds.
- 3) The aim of using a runoff model is to get an estimate of the potential impact of changing albedo, cloud cover and temperature following such type of eruption on ice-sheet runoff. The nature of these simulations is idealized and changes in cloud cover are of secondary importance compared to the temperature and albedo change as detailed in our manuscript.
- 4) Finally, in the previous version of the manuscript we did compare our study to the only other modeling study for Laki eruption (Oman et al., 2006) which shows an increase in cloud cover over Scandinavia after the Laki eruption (L303-306 in the main text).

Oman, L., Robock, A., Stenchikov, G. L. & Thordarson, T. High-latitude eruptions cast shadow over the African monsoon and the flow of the Nile. *Geophysical Research Letters* **33**, (2006).

If you are going to base the fundamental conclusion of your paper on an off-line ice model designed for a different purpose driven by climate model simulations, you

have to save the output from the climate model at the time resolutions demanded by the ice model.

We hope we have demonstrated, as shown by our new runoff experiments detailed in the previous reply, that our approach of using monthly mean cloudiness data to drive the runoff simulations does not affect our conclusions.

You have to explain how many ensemble members were used for each set of climate simulations. You have to show that this climate model does a good job for the present climate. You have to validate is Laki simulations with reconstructions of European climate that show what actually happened. There are far too many details left out to evaluate the accuracy of the results presented here. For example, see the Methods section of Tricht et al. (2016) to see how it is done.

We agree with R3 that it is important to provide such information, and we would like to point out that we already indicated this in the previous version of our manuscript in L399-401. NorESM's performances have been evaluated in several papers from a basic evaluation of the physical climate (Bentsen et al., 2013) to the climate response to future scenarios (Iversen et al., 2013), to aerosol-climate interactions (Kirkevåg et al, 2013) as well as to high-latitude Laki-type volcanic eruptions (Pausata et al., 2015). This is now restated and better phrased in L400-405 in this version of the manuscript.

Bentsen, M. *et al.* The Norwegian Earth System Model, NorESM1-M – Part 1: Description and basic evaluation of the physical climate. *Geoscientific Model Development* **6**, 687–720 (2013)

Kirkevåg, A. *et al.* Aerosol–climate interactions in the Norwegian Earth System Model–NorESM1-M. *Geoscientific Model Development* **6**, 207–244 (2013).

Iversen, T. *et al.* The Norwegian Earth System Model, NorESM1-M – Part 2: Climate response and scenario projections. *Geoscientific Model Development* **6**, 389–415 (2013).

Pausata, F. S. R., Grini, A., Caballero, R., Hannachi, A. & Seland, Ø. High-latitude volcanic eruptions in the Norwegian Earth System Model: the effect of different initial conditions and of the ensemble size. *Tellus B* **67**, (2015).

There are 10 additional comments in the attached annotated manuscript.

All the typos have been amended and comments have been addressed in the manuscript. In particular, we now emphasize the speculative nature of our claim in regard to the causative relationship between a volcanic eruption and the drainage of the Baltic Ice Lake (new L331 in the main text).

Reviewers' comments:

Reviewer #1 (Remarks to the Author):

I have read the revised version and appreciate the clarifications.

Overall, I believe that the authors presented convincing evidence of a potential causal relationship of strong volcanic activity in the North Atlantic region and rapid melting events on the Fennoscandian Ice Sheet during a part of the past deglaciation, in a study possible due to the extremely precisely synchronized proxy records which is critical for such short-lived events. The authors performed model experiments to test if this observed association can be plausibly explained by involving indirect aerosol effects on cloud cover and surface albedo changes due to ash deposition on the ice sheets. The observational records presented and discussed do support 1) anomalous volcanic activity during the deglaciation in general, 2) often producing and distributing light-absorbing ashes, 3) some of which are synchronous with extreme ice-sheet melt events as shown by the authors. Aerosol indirect effects on cloud information as well as the role of light-absorbing aerosols in the cryosphere are understood to be important in present days. The volcanic activity (frequency, duration, intensity, strength) of volcanic activity in the North Atlantic realm during the deglaciation most likely was way outside the range for which we have observations or even have performed simulations (including Laki, as argued by the authors). Given that minute amounts of industrial soot deposited on ice-sheets nowadays are frequently associated with increased melting in present day climate, it would not be surprising that vast ash blankets potentially deposited on ice-sheets in the past accelerated meltdown of the ice-sheet as proposed here. I also cannot imagine any other hypothesis to link these events sufficiently supported by other evidence. This will certainly require deeper investigations in the future.

Without claiming expertise in the modeling part of their study the simulated effects they presented appear in general to lend support to their hypothesis.

I suggest the authors to be more specific with the usage of the term "volcanic forcing" throughout the text. When do you mean aerosol direct radiative forcing, when do you mean aerosol indirect effects (change of cloud cover), when do you mean surface albedo affects (change of albedo of the ice-sheet)?

Minor points:

L. 191-192: Are there any field based measurements of albedo changes of ice-sheets following recent ash-producing eruptions available?

L. 234: Change "events" to "eruptions" to not confuse it with ash deposition

Reviewer #4 (Remarks to the Author):

This paper explains the melt of the FIS by decreasing surface albedo resulting from volcanic eruption, likely enhanced by an increase of cloud cover resulting from eruptions. While the authors have significantly improved their manuscript in response to the reviewer remarks from the first round, I am not convinced by the main story of this manuscript and large uncertainties remain in the modelling effort.

First of all, the recent lowest surface melt rates over the Greenland ice sheet were observed in 1983 and 1992 after the El Chichon (1982) and Mount Pinatubo (1991) eruption. (See for example Fettweis et al., 2007, The cryosphere). Both of these eruptions induced significant infrared radiation (LWD) negative anomalies in summer due to the temporal cooling of the Earth climate

but also to solar radiation negative anomalies due to the volcanic aerosols. The decrease of albedo due to the deposition of dark volcanic ash was fully counterbalanced by the decrease of incoming shortwave radiation (SWD). These recent observations are in total disagreement with the theory suggested in this paper.

Secondly, the author suggest that the increase of cloudiness enhanced the melt (eg. line 862-865). It is true that clouds enhance melt in the accumulation zone (where the albedo is high and where the melt is driven by LWD) but not in the low albedo zones (in the ablation zone) where the surface albedo is generally lower than the cloud albedo where the melt is rather driven by the SWD. Here, the author explain the FIS melt by decreasing albedo which is more sensitive to SWD anomalies than LWD anomalies. Therefore, the increase of cloudiness likely dampened the melt albedo positive feedback but not enhanced it. This issue should be discussed in the text.

Thirdly, the author use a robust snow model but with monthly forcing. Even if these outputs are daily downscaled, large uncertainties remains in the daily clouds and SWD variability. Why does the snowmodel model forced by cloudiness and not by LWD?

Finally, it seems that the impact of increasing atmospheric volcanic aerosols on SWD is not taken into here while the melt albedo feedback is fully driven by SWD. Therefore, the numbers given in table 2 are likely a bit hazardous. Therefore, only significant albedo reduction (a lot of larger than the -15% tested here) could induce a significant FIS melting as both cloudiness increase (as shown in this study) and atmospheric volcanic aerosols should damp the melt albedo feedback.

What are the anomalies of LWD/SWD of the "volcanic" simulation in respect to the control simulation ? What are the anomalies of accumulation ? What are the differences to explain that the recent impact of Volcano are in contradiction with the theory suggested here ?

It is clear that volcanic eruption decrease ice sheet albedo but albedo is not the only factor driving the melt and for me, the uncertainties remain too high to claim here that the decrease of albedo was the mean driving factor of the FIS melt.

We would like to thank once again the referees for their time and their constructive criticism
that has helped improving our manuscript. Please find below our responses to the points
and comments raised by the reviewers.

Reviewer #1 (Remarks to the Author):

I have read the revised version and appreciate the clarifications.

Overall, I believe that the authors presented convincing evidence of a potential causal
relationship of strong volcanic activity in the North Atlantic region and rapid melting events
on the Fennoscandian Ice Sheet during a part of the past deglaciation, in a study possible
due to the extremely precisely synchronized proxy records which is critical for such short-
lived events. The authors performed model experiments to test if this observed association
can be plausibly explained by involving indirect aerosol effects on cloud cover and surface
albedo changes due to ash deposition on the ice sheets. The observational records
presented and discussed do support 1) anomalous volcanic activity during the deglaciation in
general, 2) often producing and distributing light-absorbing ashes, 3) some of which are
synchronous with extreme ice-sheet melt events as shown by the authors. Aerosol indirect
effects on cloud information as well as the role of light-absorbing aerosols in the cryosphere
are understood to be important in present days. The volcanic activity (frequency, duration,
intensity, strength) of volcanic activity in the North Atlantic realm during the deglaciation
most likely was way outside the range for which we have observations or even have
performed simulations (including Laki, as argued by the authors). Given that minute
amounts of industrial soot deposited on ice-sheets nowadays are frequently associated with
increased melting in present day climate, it would not be surprising that vast ash blankets
potentially deposited on ice-sheets in the past accelerated meltdown of the ice-sheet as
proposed here. I also cannot imagine any other hypothesis to link these events sufficiently
supported by other evidence. This will certainly require deeper investigations in the future.

Without claiming expertise in the modeling part of their study the simulated effects they
presented appear in general to lend support to their hypothesis.

I suggest the authors to be more specific with the usage of the term “volcanic forcing”
throughout the text. When do you mean aerosol direct radiative forcing, when do you mean
aerosol indirect effects (change of cloud cover), when do you mean surface albedo affects
(change of albedo of the ice-sheet)?

This has been changed in the text where appropriate (lines 52, 314, 318, 341, 342, 345).

Minor points:

35 L. 191-192: Are there any field-based measurements of albedo changes of ice-sheets
following recent ash-producing eruptions available?

Indeed, there are a number of studies showing that tephra deposition on glaciers has a
critical influence on surface melting via albedo changes. These studies were cited in the
Introduction but we have now highlighted the related findings also in the Discussion (line
246-248).

41 L. 234: Change "events" to "eruptions" to not confuse it with ash deposition.

This has been changed accordingly (lines 96, 115, 116, 134, 236, 238, 254, 290, 391, 879,
886).

Reviewer #4 (Remarks to the Author):

This paper explains the melt of the FIS by decreasing surface albedo resulting from volcanic
eruption, likely enhanced by an increase of cloud cover resulting from eruptions. While the
authors have significantly improved their manuscript in response to the reviewer remarks
from the first round, I am not convinced by the main story of this manuscript and large
uncertainties remain in the modelling effort.

We thank R4 for giving us the opportunity to further improve our study and clarify some
issues regarding our modeling approach. The points raised have particularly highlighted
aspects that (we hope they will agree) have helped to explore some of the nuances of our
results and the runoff modelling approach we have taken. However, we would also like to
emphasize at the outset of this response that the results of the runoff model are used as a
tool to explore a hypothesis regarding the mechanism for high-latitude northern hemisphere
volcanism being linked to enhanced runoff from the Fennoscandian Ice Sheet. We suggest
that the analysis we provide in the manuscript indicates that the links between sulfur peaks
and exceptionally thick varves is beyond reasonable doubt, while the runoff modelling
experiments help to demonstrate that our suggested mechanism is (i) physically possible
and (ii) solidly based in current understanding of contemporary ice sheet mass balance and
hydrology. Of course, we remain open to the possibility that in the future evidence may
emerge that indicate other mechanisms were at play, though given current knowledge we
propose that we are able to demonstrate that the mechanism suggested in this manuscript
is the most physically reasonable.

We also feel that it is important to highlight that each eruption would not have had a large
global climate impact similar to Laki, though could easily have had a significant impact on
the FIS (this is now highlighted more clearly in the manuscript e.g. lines 204-207). To provide
a contemporary example, the Eyjafjallajokull eruption in terms of volume was a very small
Icelandic eruption, yet it resulted in ashfall in Europe without a major climate impact. In
large part this comes down to the nature of where the eruption occurred (i.e. in the case of
Eyjafjallajokull underneath a branch of Myrdalsjokull ice cap, where interaction with
meltwater resulted in the eruption being explosive rather than effusive, an causing an
extensive ash cloud). Given that Iceland would have had a significant ice cover during the
last deglaciation (Patton et al., 2017), one could get a relatively small eruption with a
modest sulfur injection into the troposphere/stratosphere (thus forming a sulfur peak in
Greenland ice cores) resulting in a minor/negligible climate impact but a strong melt
response from FIS due to significant ashfall. This is discussed in detail in lines 239-257.

Patton, H., Hubbard, A., Bradwell, T. and Schomacker, A., 2017. The configuration, sensitivity
and rapid retreat of the Late Weichselian Icelandic ice sheet. *Earth-Science Reviews*.

First of all, the recent lowest surface melt rates over the Greenland ice sheet were observed
in 1983 and 1992 after the El Chichon (1982) and Mount Pinatubo (1991) eruption. (See for
example Fettweis et al., 2007, The cryosphere). Both of these eruptions induced significant

infrared radiation (LWD) negative anomalies in summer due to the temporal cooling of the
Earth climate but also to solar radiation negative anomalies due to the volcanic aerosols. The
decrease of albedo due to the deposition of dark volcanic ash was fully counterbalanced by
the decrease of incoming shortwave radiation (SWD). These recent observations are in total
disagreement with the theory suggested in this paper.

While equatorial eruptions such as El Chichon and Pinatubo were substantial and had a
global climate impact, the latitude of their eruption meant that there would have been very
little (if any) ash deposition on the Greenland Ice Sheet (this is now mentioned in the text:
lines 248-250). Our manuscript highlights the importance of high latitude northern
hemisphere eruptions to the runoff response of the FIS, while no evidence is found to link
ETVs to eruptions that have a probable equatorial origin similar to El Chichon/Pinatubo
(Figure S1; lines 111-118, lines 214-220, and lines 233-238). Rather than being at odds with
the theory we present, the low melt rates observed in Greenland following these large
equatorial eruptions that had a significant climate impact are entirely consistent with our
runoff model results (e.g Table 1; Figure S3; Figure S9b; see attached file:
LWD_SWD_anomalies.xlsx).

Using the runoff model, we establish that ashfall onto FIS following high latitude northern
hemisphere (i.e. Icelandic) eruptions would have had the potential to cause an increase in
melt if the eruption had no climate impact (Table 3; Figure S5), and even offset a volcanically
driven drop in air temperature at high elevations for modest albedo changes (e.g. Table 2;
Figure S4). Given a prevailing westerly air track across the Atlantic, it is also likely that FIS
would have been more susceptible to ashfall compared to Greenland given the decreased
likelihood of an ash cloud tracking towards Greenland.

Secondly, the author suggests that the increase of cloudiness enhanced the melt (eg. line
862-865). It is true that clouds enhance melt in the accumulation zone (where the albedo is
high and where the melt is driven by LWD) but not in the low albedo zones (in the ablation
zone) where the surface albedo is generally lower than the cloud albedo where the melt is
rather driven by the SWD.

Increase in runoff in the accumulation zone is consistent with the ETV mechanism described
in the manuscript (i.e. runoff from higher elevations causing a more extensive subglacial
hydrological system, which evacuates more subglacial sediment into the proglacial lake (see
lines 264-280). The SW energy that can be utilized for melt in the ablation zone following
ashfall would also increase due to the associated decrease in albedo. Our results are
therefore consistent with R4's comment.

Here, the authors explain the FIS melt by decreasing albedo which is more sensitive to SWD
anomalies than LWD anomalies. Therefore, the increase of cloudiness likely dampened the
melt albedo positive feedback but not enhanced it. This issue should be discussed in the
text.

We thank R4 for bringing this up as it has allowed us to investigate a previously unexplored
aspect of our mechanism for volcanic driven enhanced runoff. This has led us to change how
we deal with shortwave radiative forcing (SWRF) within the runoff model so we now explore
the effects of a much more "aggressive" volcanic forcing on SWRF. The results of these new
simulations remain consistent with our original mechanism, though further highlight the
importance of (i) the timing of an eruption, and (ii) the potential for an eruption to
significantly impact SWRF (i.e. sulfur volume emission into the atmosphere) (see below).

Runoff simulations in previous versions of this manuscript used SWRF for volcanic and non-
volcanic forced scenarios that were calculated outside of the climate model for a
topographically unshielded point at 60°N under a uniform sky, with corrections made
according to cloud cover data provided by the climate model – this approach ignored the
impact of sulfur emission on SWRF. Following R4's comments, we now incorporate this
directly into our runoff model using monthly SWRF output from the climate model to adjust
the hourly timescale SWRF that are calculated outside the model (see methods, lines 588-
609). This produces the volcanic/non-volcanic SWRFs shown in new Figure S6.

As R4 alluded to in a previous comment, we find that where there is a significant reduction
in SWRF due to a volcanic eruption (e.g. Pinatubo/El Chichon/Laki type sulfur emission
events) this acts to suppress melt rather than enhance it, which is not consistent with the
formation of an ETV (Figure S3). However, to negate our mechanism on this basis would be
assuming that every eruption that could have affected FIS is on the scale of Laki (and
occurred in summer), which is emphatically not the case (as discussed earlier in this
response to reviewers, i.e. 2nd paragraph of our 1st response to R4). Where the assumption is
made that an eruption does not significantly impact SWRF due to a lack of sulfur emission,
our results are consistent with those in previous versions of the manuscript (i.e. a volcanic
eruption can lead to enhanced runoff via decreases in albedo that is consistent with the
formation of an ETV (Figure S4 and Figure S5).

In response to these findings we have expanded our discussion on the conditions required
for Icelandic eruptions to drive the formation of an ETV (lines 187-211, lines 239-246). We
feel that this further contributes to the robustness of our mechanism, and provides
additional explanation as to why each sulfur peak (and their magnitude) does not necessarily
correspond to an ETV (see lines 135-155, lines 281-291, and lines 316-323).

Thirdly, the author use a robust snow model but with monthly forcing. Even if these outputs
are daily downscaled, large uncertainties remains in the daily clouds and SWD variability.
Why does the snowmodel model forced by cloudiness and not by LWD?

Unfortunately, by their nature palaeoclimatological simulations are beset by uncertainties
especially at finer timescales due to the limited/non-existent proxy data that would allow
validation of results at such temporal resolution. Therefore, we strongly think that monthly
mean forcing output is sufficient to get the broad picture that can be comparable to proxy
data. It should also be noted that our experiments are idealized sensitivity experiment
aimed at understanding the key drivers of FIS runoff during the deglaciation. Nevertheless,
in our runoff simulations we have sought to include realistic sub-daily forcing where it is
possible to constrain this (e.g. SWRF). We have also explored the impact of superimposing
daily timescale noise on cloudiness data on volcanic and non-volcanically driven scenarios to
evaluate its influence, with the outcome being that this is generally negligible (new Figure
S11). While we acknowledge that it is impossible to conduct runoff simulations for the
Younger Dryas with the confidence levels equivalent to conducting a study on contemporary
ice sheet runoff response (that benefit from direct meteorological station observations), we
are confident that we have taken all possible steps for ensuring that the data used to run the
model are as realistic as possible (given what can be expected), while we have also
comprehensively tested the model sensitivity to key uncertain variables.

The snow model is forced by cloudiness rather than LWD using equation 2 in Morris et al.
(2014) following work by Brutsaert (1975) on the emissivity of the atmosphere for varying
cloud conditions. The choice to drive the runoff model in this way was primarily for

convenience given that the climate model was able to output cloudiness data, while the
runoff model could be easily modified from its original form (Morris et al., 2014) to
incorporate this variable.

Brutsaert, W., 1975. On a derivable formula for long-wave radiation from clear skies. *Water*
*Resources Research*, 11(5), pp.742-744.

Finally, it seems that the impact of increasing atmospheric volcanic aerosols on SWD is not
taken into here while the melt albedo feedback is fully driven by SWD. Therefore, the
numbers given in table 2 are likely a bit hazardous. Therefore, only significant albedo
reduction (a lot of larger than the -15% tested here) could induce a significant FIS melting as
both cloudiness increase (as shown in this study) and atmospheric volcanic aerosols should
damp the melt albedo feedback.

Please see Response to Reviewers' comments lines 122-152.

What are the anomalies of LWD/SWD of the "volcanic" simulation in respect to the control
simulation ?

See attached file *LWD_SWD_anomalies.xlsx* for the anomalies associated with the results
shown in tables 1-4 in the main text.

What are the anomalies of accumulation?

See attached file *Accumulation_anomalies.xlsx* for the anomalies associated with the results
shown in tables 1-4 in the main text.

What are the differences to explain that the recent impact of Volcano are in contradiction
with the theory suggested here?

Please see Response to Reviewers' comments lines 90-106 and 137-152.

It is clear that volcanic eruption decrease ice sheet albedo but albedo is not the only factor
driving the melt and for me, the uncertainties remain too high to claim here that the
decrease of albedo was the mean driving factor of the FIS melt.

We hope that the above has gone some way towards explaining the uncertainties associated
with the runoff model and clarifying aspects of our modelling approach. To reiterate a point
mentioned earlier in our response, the runoff model results are intended to explore the
feasibility of our suggested mechanism for how Icelandic volcanic eruptions could have
resulted in enhanced FIS runoff. We acknowledge that uncertainties exist, though our
modelling approach has not shyed away from these, as evidenced by our extensive
sensitivity testing of nearly 5000 unique scenarios. This sensitivity testing also includes
evaluating the consistency of results where high temporal resolution data cannot
realistically be expected to be obtained for the time period under investigation (i.e. cloud
cover), and the inclusion of realistic sub-daily data where possible to improve the
simulations themselves (i.e. calculation of hourly SWRF variability outside of the climate
model). We therefore consider our runoff model results to provide the absolute best insight
possible into the potential runoff responses of FIS.

Summarizing our argument briefly, the manuscript demonstrates that high latitude northern
hemispheric eruptions are linked beyond doubt to the formation of ETVs, and we consider
our runoff model results to be consistent with the mechanism we suggest.

REVIEWERS' COMMENTS:

Reviewer #4 (Remarks to the Author):

The author's responses to my concerns are convincing and I am fine with the present version of the manuscript.

Reviewer #4 (Remarks to the Author):

The author's responses to my concerns are convincing and I am fine with the present version of the manuscript.

We thank Reviewer #4 for his/her comments that have helped greatly in the improvement of our manuscript.